# Non-linear dimensionality reduction on extracellular waveforms reveals cell type diversity in premotor cortex

Eric Kenji Lee[1], Hymavathy Balasubramanian[2], Alexandra Tsolias[3], Stephanie Udochukwu Anakwe[4], Maria Medalla[3], Krishna V Shenoy[5,6,7,8,9,10], Chandramouli Chandrasekaran[1,3,11,12]*

[1]Psychological and Brain Sciences, Boston University, Boston, United States; [2]Bernstein Center for Computational Neuroscience, Bernstein Center for Computational Neuroscience, Berlin, Germany; [3]Department of Anatomy and Neurobiology, Boston University, Boston, United States; [4]Undergraduate Program in Neuroscience, Boston University, Boston, United States; [5]Department of Electrical Engineering, Stanford University, Stanford, United States; [6]Department of Bioengineering, Stanford University, Stanford, United States; [7]Department of Neurobiology, Stanford University, Stanford, United States; [8]Wu Tsai Neurosciences Institute, Stanford University, Stanford, United States; [9]Bio-X Institute, Stanford University, Stanford, United States; [10]Howard Hughes Medical Institute, Stanford University, Stanford, United States; [11]Center for Systems Neuroscience, Boston University, Boston, United States; [12]Department of Biomedical Engineering, Boston University, Boston, United States

*For correspondence:
cchandr1@bu.edu

Competing interests: The authors declare that no competing interests exist.

**Abstract** Cortical circuits are thought to contain a large number of cell types that coordinate to produce behavior. Current in vivo methods rely on clustering of specified features of extracellular waveforms to identify putative cell types, but these capture only a small amount of variation. Here, we develop a new method (*WaveMAP*) that combines non-linear dimensionality reduction with graph clustering to identify putative cell types. We apply *WaveMAP* to extracellular waveforms recorded from dorsal premotor cortex of macaque monkeys performing a decision-making task. Using *WaveMAP*, we robustly establish eight waveform clusters and show that these clusters recapitulate previously identified narrow- and broad-spiking types while revealing previously unknown diversity within these subtypes. The eight clusters exhibited distinct laminar distributions, characteristic firing rate patterns, and decision-related dynamics. Such insights were weaker when using feature-based approaches. *WaveMAP* therefore provides a more nuanced understanding of the dynamics of cell types in cortical circuits.

## Introduction

The processes involved in decision-making, such as deliberation on sensory evidence and the preparation and execution of motor actions, are thought to emerge from the coordinated dynamics within and between cortical layers (*Chandrasekaran et al., 2017*; *Finn et al., 2019*), cell types (*Pinto and Dan, 2015*; *Estebanez et al., 2017*; *Lui et al., 2021*; *Kvitsiani et al., 2013*), and brain areas (*Gold and Shadlen, 2007*; *Cisek, 2012*). A large body of research has described differences in decision-related dynamics across brain areas (*Ding and Gold, 2012*; *Thura and Cisek, 2014*; *Roitman and Shadlen, 2002*; *Hanks et al., 2015*) and a smaller set of studies has provided insight into layer-dependent dynamics during decision-making (*Chandrasekaran et al., 2017*; *Finn et al.,*

*2019*; *Chandrasekaran et al., 2019*; *Bastos et al., 2018*). However, we currently do not understand how decision-related dynamics emerge across putative cell types. Here, we address this open question by developing a new method, *WaveMAP*, that combines non-linear dimensionality reduction and graph-based clustering. We apply *WaveMAP* to extracellular waveforms to identify putative cell classes and examine their physiological, functional, and laminar distribution properties.

In mice, and to some extent in rats, transgenic tools allow the in vivo detection of particular cell types (*Pinto and Dan, 2015*; *Lui et al., 2021*), whereas in vivo studies in primates are largely restricted to using features of the extracellular action potential (EAP) such as trough to peak duration, spike width, and cell firing rate (FR). Early in vivo monkey work (*Mountcastle et al., 1969*) introduced the importance of EAP features, such as spike duration and action potential (AP) width, in identifying cell types. These experiments introduced the concept of broad- and narrow-spiking neurons. Later experiments in the guinea pig (*McCormick et al., 1985*), cat (*Azouz et al., 1997*), and the rat (*Simons, 1978*; *Barthó et al., 2004*) then helped establish the idea that these broad- and narrow-spiking extracellular waveform shapes mostly corresponded to excitatory and inhibitory cells, respectively. These results have been used as the basis for identifying cell types in primate recordings (*Johnston et al., 2009*; *Merchant et al., 2008*; *Merchant et al., 2012*). This method of identifying cell types in mammalian cortex in vivo is widely used in neuroscience but it is insufficient to capture the known structural and transcriptomic diversity of cell types in the monkey and the mouse (*Hodge et al., 2019*; *Krienen et al., 2020*). Furthermore, recent observations in the monkey defy this simple classification of broad- and narrow-spiking cells as corresponding to excitatory and inhibitory cells, respectively. Three such examples in the primate that have resisted this principle are narrow-spiking pyramidal tract neurons in deep layers of M1 (Betz cells, *Vigneswaran et al., 2011*; *Soares et al., 2017*), narrow and broad spike widths among excitatory pyramidal tract neurons of premotor cortex (*Lemon et al., 2021*), and narrow-spiking excitatory cells in layer III of V1, V2, and MT (*Constantinople et al., 2009*; *Amatrudo et al., 2012*; *Onorato et al., 2020*; *Kelly et al., 2019*).

To capture a more representative diversity of cell types in vivo, more recent studies have incorporated additional features of EAPs (beyond AP width) such as trough to peak duration (*Ardid et al., 2015*), repolarization time (*Trainito et al., 2019*; *Banaie Boroujeni et al., 2021*), and triphasic waveform shape (*Barry, 2015*; *Robbins et al., 2013*). Although these user-specified methods are amenable to human intuition, they are insufficient to distinguish between previously identified cell types (*Krimer et al., 2005*; *Vigneswaran et al., 2011*; *Merchant et al., 2012*). It is also unclear how to choose these user-specified features in a principled manner (i.e. one set that maximizes explanatory power) as they are often highly correlated with one another. This results in different studies choosing between different sets of specified features each yielding different inferred cell classes (*Trainito et al., 2019*; *Viskontas et al., 2007*; *Katai et al., 2010*; *Sun et al., 2021*). Thus, it is difficult to compare putative cell types across literature. Some studies even conclude that there is no single set of specified features that is a reliable differentiator of type (*Weir et al., 2014*).

These issues led us to investigate techniques that do not require feature specification but are designed to find patterns in complex datasets through non-linear dimensionality reduction. Such methods have seen usage in diverse neuroscientific contexts such as single-cell transcriptomics (*Tasic et al., 2018*; *Becht et al., 2019*), in analyzing models of biological neural networks (*Maheswaranathan et al., 2019*; *Kleinman et al., 2019*), the identification of behavior (*Bala et al., 2020*; *Hsu and Yttri, 2020*; *Dolensek et al., 2020*), and in electrophysiology (*Jia et al., 2019*; *Gouwens et al., 2020*; *Klempíř et al., 2020*; *Markanday et al., 2020*; *Dimitriadis et al., 2018*).

Here, in a novel technique that we term *WaveMAP*, we combine a non-linear dimensionality reduction method (Universal Manifold Approximation and Projection [UMAP], *McInnes et al., 2018*) with graph community detection (Louvain community detection, (*Blondel et al., 2008*); we colloquially call 'clustering') to understand the physiological properties, decision-related dynamics, and laminar distribution of candidate cell types during decision-making. We applied *WaveMAP* to extracellular waveforms collected from neurons in macaque dorsal premotor cortex (PMd) in a decision-making task using laminar multi-channel probes (16 electrode 'U-probes'). We found that *WaveMAP* significantly outperformed current approaches without need for user-specification of waveform features like trough to peak duration. This data-driven approach exposed more diversity in extracellular waveform shape than any constructed spike features in isolation or in combination. Using interpretable machine learning, we also show that *WaveMAP* picks up on nuanced and meaningful biological variability in waveform shape.

*WaveMAP* revealed three broad-spiking and five narrow-spiking waveform types that differed significantly in shape, physiological, functional, and laminar distribution properties. Although most narrow-spiking cells had the high maximum firing rates typically associated with inhibitory neurons, some had firing rates similar to broad-spiking neurons which are typically considered to be excitatory. The time at which choice selectivity ('discrimination time') emerged for many narrow-spiking cell classes was earlier than broad-spiking neuron classes—except for the narrow-spiking cells that had broad-spiking like maximum firing rates. Finally, many clusters had distinct laminar distributions that appear layer-dependent in a manner matching certain anatomical cell types. This clustering explains variability in discrimination time over and above previously reported laminar differences (*Chandrasekaran et al., 2017*). Together, this constellation of results reveals previously undocumented relationships between waveform shape, physiological, functional, and laminar distribution properties that are missed by traditional approaches. Our results provide powerful new insights into how candidate cell classes can be better identified and how these types coordinate with specific timing, across layers, to shape decision-related dynamics.

## Results

### Task and behavior

Two male rhesus macaques (T and O) were trained to perform a red-green reaction time decision-making task (*Figure 1A*). The task was to discriminate the dominant color of a central static red-green checkerboard cue and to report their decision with an arm movement towards one of two targets (red or green) on the left or right (*Figure 1A*).

The timeline of the task is as follows: a trial began when the monkey touched the center target and fixated on a cross above it. After a short randomized period, two targets red and green appeared on the either side of the center target (see *Figure 1B*, top). The target configuration was randomized: sometimes the left target was red and the right target was green or vice versa. After another short randomized target viewing period, a red-green checkerboard appeared in the center of the screen with a variable mixture of red and green squares.

We parameterized the variability of the checkerboard by its signed color coherence and color coherence. The signed color coherence (SC) provides an estimate of whether there are more red or green squares in the checkerboard. Positive SC indicates the presence of more red squares, whereas negative SC indicates more green squares. SC close to zero (positive or negative) indicates an almost even number of red or green squares (*Figure 1B*, bottom). The coherence (C) provides an estimate of the difficulty of a stimulus. Higher coherence indicates that there is more of one color than the other (an easy trial) whereas a lower coherence indicates that the two colors are more equal in number (a difficult trial).

Our monkeys demonstrated the range of behaviors typically observed in decision-making tasks: monkeys made more errors and were slower for lower coherence checkerboards compared to higher coherence checkerboards (*Figure 1C,D*). We used coherence, choice, and reaction times (RT) to analyze the structure of decision-related neural activity.

### Recordings and single neuron identification

While monkeys performed this task, we recorded single neurons from the caudal aspect of dorsal premotor cortex (PMd; *Figure 1E*, top) using single tungsten (FHC electrodes) or linear multi-contact electrodes (Plexon U-Probes, 625 neurons, 490 U-probe waveforms; *Figure 1E*, right) and a Cerebus Acquisition System (Blackrock Microsystems). In this study, we analyzed the average EAP waveforms of these neurons. All waveforms were analyzed after being filtered by a fourth-order high-pass Butterworth filter (250 Hz). A 1.6 ms snippet of the waveform was recorded for each spike and used in these analyses, a duration longer than many studies of waveform shape (*Merchant et al., 2012*).

We restricted our analysis to well-isolated single neurons identified through a combination of careful online isolation combined with offline spike sorting (see Methods section: *Identification of single neurons during recordings*). Extracellular waveforms were isolated as single neurons by only accepting waveforms with minimal ISI violations (1.5% < 1.5 ms). This combination of online

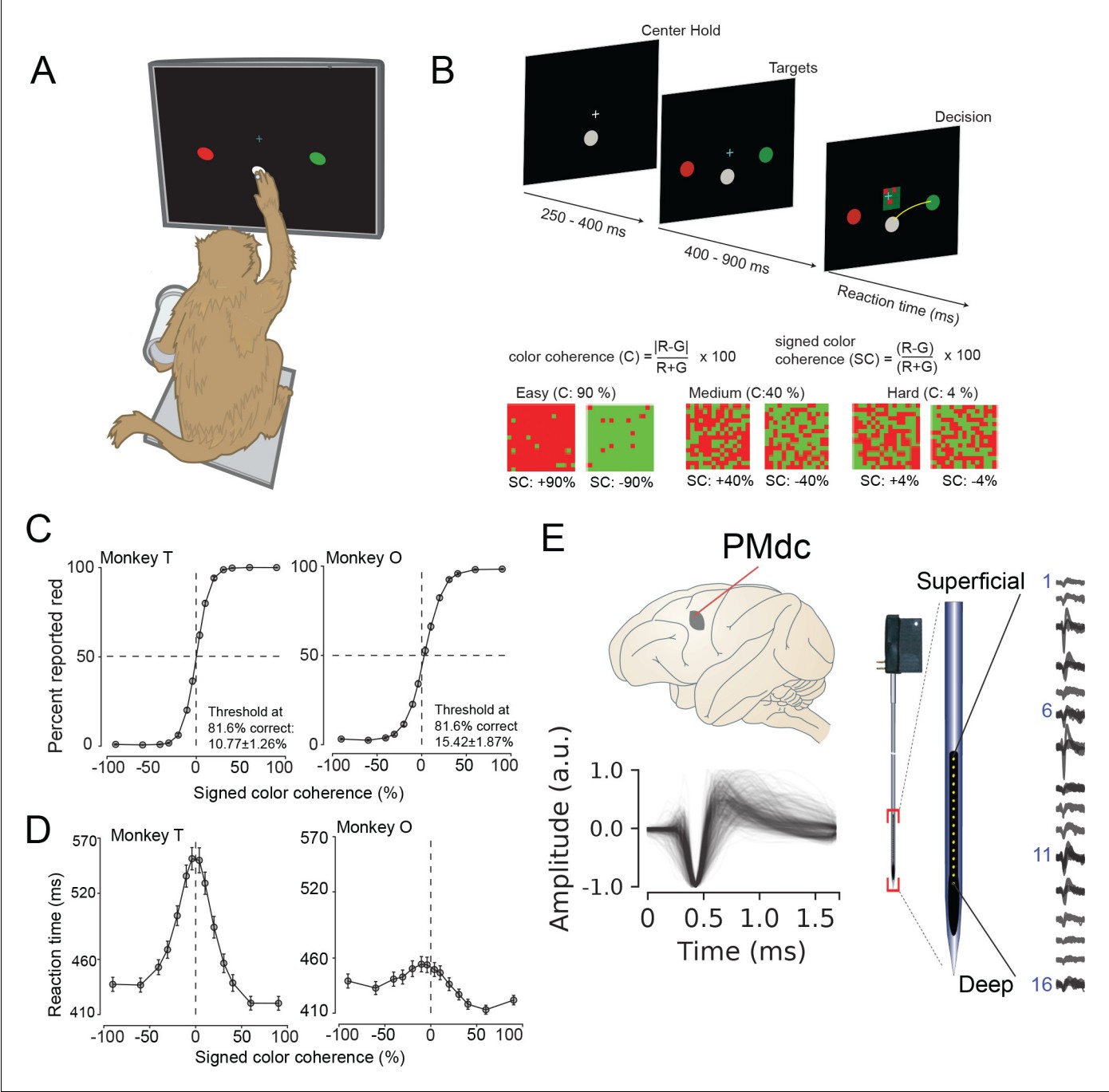

**Figure 1.** Recording locations, waveform shapes, techniques, task, and discrimination behavior. (A) An illustration of the behavioral setup in the discrimination task. The monkey was seated with one arm free and one arm gently restrained in a plastic tube via a cloth sling. An infrared-reflecting (IR) bead was taped to the forefinger of the free hand and was used in tracking arm movements. This gave us a readout of the hand's position and allowed us to mimic a touch screen. (B) A timeline of the decision-making task (top). At bottom is defined the parametrization of difficulty in the task in terms of color coherence and signed color coherence (SC). (C) Average discrimination performance and (D) Reaction time (RT) over sessions of the two monkeys as a function of the SC of the checkerboard cue. RT plotted here includes both correct and incorrect trials for each session and then averaged across sessions. Gray markers show measured data points along with 2 × S.E.M. estimated over sessions. For many data points in (C), the error bars lie within the marker. X-axes in both (C), (D) depict the SC in %. Y-axes depict the percent responded red in (C) and RT in (D). Also shown in the inset of (C) are discrimination thresholds (mean ± S.D. over sessions) estimated from a Weibull fit to the overall percent correct as a function of coherence. The discrimination threshold is the color coherence at which the monkey made 81.6% correct choices. Seventy-five sessions for monkey T (128,989 trials) and 66 sessions for monkey O (108,344 trials) went into the averages. (E) The recording location in caudal PMd (top); normalized and aligned isolated single-unit waveforms (n = 625, 1.6 ms each, bottom); and schematic of the 16-channel Plexon U-probe (right) used during the behavioral experiment.

vigilance, combined with offline analysis, provides us the confidence to label these waveforms as single neurons.

We used previously reported approaches to align, average, and normalize spikes (*Kaufman et al., 2013*; *Snyder et al., 2016*). Spikes were aligned in time via their depolarization trough and normalized between −1 and 1. 'Positive spiking' units with large positive amplitude prehyperpolarization spikes were dropped from the analysis due to their association with dendrites and axons (*Gold et al., 2009*; *Barry, 2015*; *Sun et al., 2021*). Recordings were pooled across monkeys to increase statistical power for *WaveMAP*.

## Non-linear dimensionality reduction with graph clustering reveals robust low-dimensional structure in extracellular waveform shape

In *WaveMAP* (*Figure 2*), we use a three-step strategy for the analysis of extracellular waveforms: We first passed the normalized and trough-aligned waveforms (*Figure 2A–i*) into UMAP to obtain a high-dimensional graph (*Figure 2A–ii*; *McInnes et al., 2018*). Second, we used this graph (*Figure 2B–iii*) and passed it into Louvain clustering (*Figure 2B–iv*, *Blondel et al., 2008*), to delineate high-dimensional clusters. Third, we used UMAP to project the high-dimensional graph into two dimensions (*Figure 2B–v*). We colored the data points in this projected space according to their Louvain cluster membership found in step two to arrive at our final *WaveMAP* clusters (*Figure 2B–vi*). We also analyzed the *WaveMAP* clusters using interpretable machine learning (*Figure 2B–vii*) and also an inverse transform of UMAP (*Figure 2B–viii*). A detailed explanation of the steps associated with *WaveMAP* is available in the methods, and further mathematical details of *WaveMAP* are available in the Supplementary Information.

*Figure 3A* shows how *WaveMAP* provides a clear organization without the need for prior specification of important features. For expository reasons, and to link to prior literature (*McCormick et al., 1985*; *Connors et al., 1982*), we use the trough to peak duration to loosely subdivide these eight clusters into 'narrow-spiking' and 'broad-spiking' cluster sets. The broad-spiking clusters had a trough to peak duration of 0.74 ± 0.24 ms (mean ± S.D.) and the narrow-spiking clusters had a trough to peak duration of 0.36 ± 0.07 ms (mean ± S.D.). The narrow-spiking neurons are shown in warm colors (including green) at right in *Figure 3A* and the broad-spiking neurons are shown in cool colors at left in the same figure. The narrow-spiking set was composed of five clusters with 'narrow-spiking' waveforms (clusters ①, ②, ③, ④, ⑤) and comprised ~12%, ~12%, ~18%, ~7%, and ~19% (n = 72, 78, 113, 43, and 116) respectively of the total waveforms, for ~68% of total waveforms. The broad-spiking set was composed of three 'broad-spiking' waveform clusters (⑥, ⑦, and ⑧) comprising ~13%, ~5%, and ~15% (n = 80, 29, and 94) respectively and collectively ~32% of total waveforms.

The number of clusters identified by *WaveMAP* is dependent on the resolution parameter for Louvain clustering. A principled way to choose this resolution parameter is to use the modularity score (a measure of how tightly interconnected the members of a cluster are) as the objective function to maximize. We chose a resolution parameter of 1.5 that maximized modularity score while ensuring that we did not overly fractionate the dataset (n < 20 within a cluster; *Figure 3A,B*, and columns of *Figure 3—figure supplement 1A*). Additional details are available in the 'Parameter Choice' section of the Supplementary Information.

Louvain clustering with this resolution parameter of 1.5 identified eight clusters in total (*Figure 3A*). Note, using a slightly higher resolution parameter (2.0), a suboptimal solution in terms of modularity, led to seven clusters (*Figure 3—figure supplement 1A*). The advantage of Louvain clustering is that it is hierarchical and choosing a slightly larger resolution parameter will only merge clusters rather than generating entirely new cluster solutions. Here, we found that the higher resolution parameter merged two of the broad-spiking clusters ⑥ and ⑦ while keeping the rest of the clusters largely intact and more importantly, did not lead to material changes in the conclusions of analyses of physiology, decision-related dynamics, or laminar distribution described below. Finally, an alternative ensembled version of the Louvain clustering algorithm (ensemble clustering for graphs [ECG] *Poulin and Théberge, 2018*), which requires setting no resolution parameter, produced a clustering almost exactly the same as our results (*Figure 3—figure supplement 1C*).

To validate that *WaveMAP* finds a 'real' representation of the data, we examined if a very different method could learn the same representation. We trained a gradient boosted decision tree classifier (with a softmax multi-class objective) on the exact same waveform data (vectors of 48 time

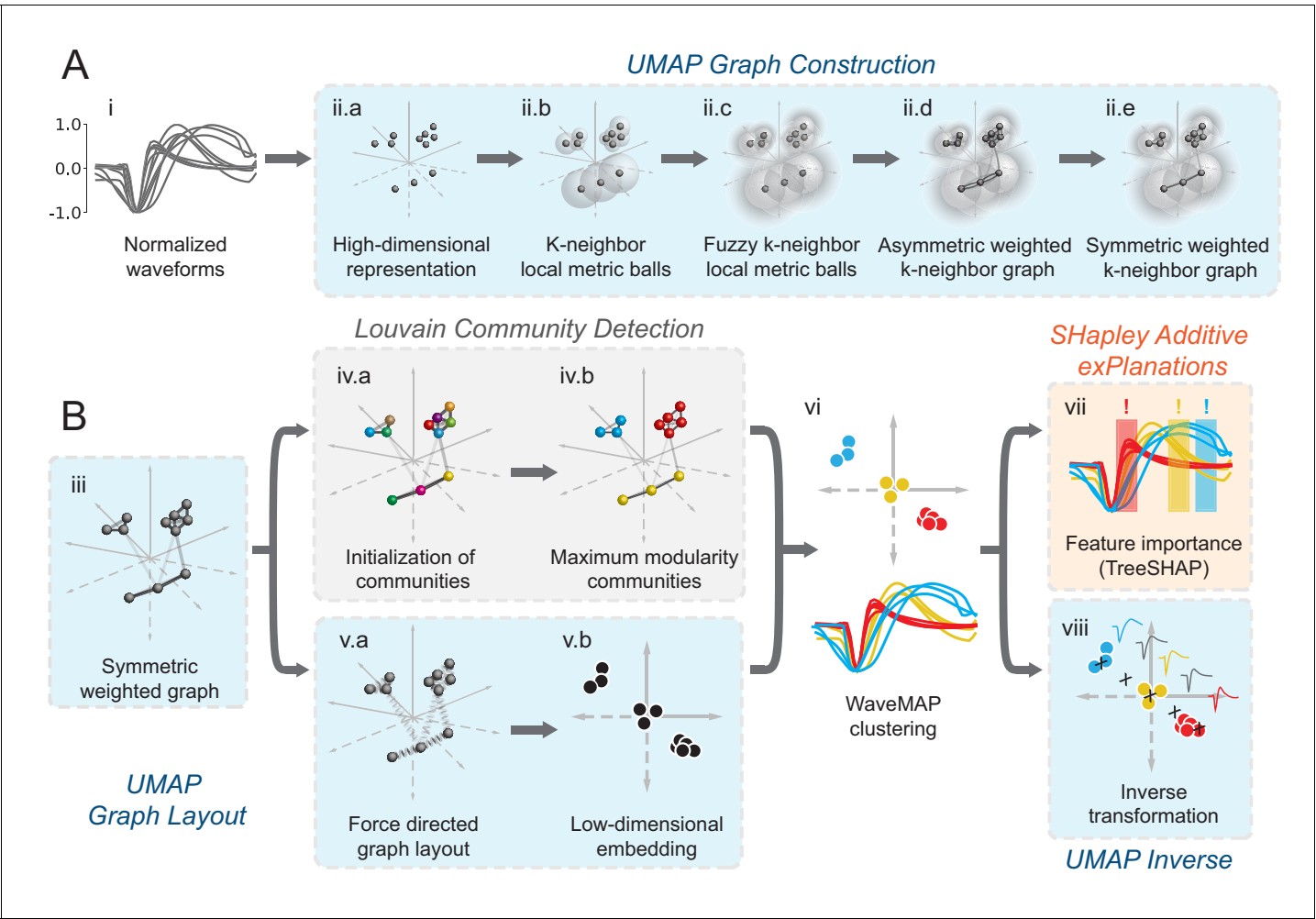

**Figure 2.** Schematic of *WaveMAP*. (**A**) *WaveMAP* begins with UMAP which projects high-dimensional data into lower dimension while preserving local and global relationships (see *Figure 2—figure supplement 1A* for an intuitive diagram). Normalized average waveforms from single units (i) are passed to UMAP (*McInnes et al., 2018*) which begins with the construction of a high-dimensional graph (ii). In the high-dimensional space (ii.a), UMAP constructs a distance metric local to each data point (ii.b). The unit ball (ball with radius of one) of each local metric stretches to the 1st-nearest neighbor. Beyond this unit ball, local distances decrease (ii.c) according to an exponential distribution that is scaled by the local density. This local metric is used to construct a weighted graph with asymmetric edges (ii.d). The 1-nearest neighbors are connected by en edge of weight 1.0. For the next $k-1$-nearest neighbors, this weight then falls off according to the exponential local distance metric (in this diagram k = 4 with some low weight connections omitted for clarity). These edges, $a$ and $b$, are made symmetric according to $a + b - a \cdot b$ (ii.e). (**B**) The high-dimensional graph (iii) captures latent structure in the high-dimensional space. We can use this graph in Louvain community detection (Louvain, iv) (*Blondel et al., 2008*) to find clusters (see *Figure 2—figure supplement 1B* for an intuitive diagram). In Louvain, each data point is first initialized as belonging to its own 'community' (iv.a, analogous to a cluster in a metric space). Then, in an iterative procedure, each data point joins neighboring communities until a measure called 'modularity' is maximized (iv.b, see Supplemental Information for a definition of modularity). Next, data points in the same final community are aggregated to a single node and the process repeats until the maximal modularity is found on this newly aggregated graph. This process then keeps repeating until the maximal modularity graph is found and the final community memberships are passed back to the original data points. We can also use this graph to find a low-dimensional representation through a graph layout procedure (**v**). The graph layout proceeds by finding a 'low energy' configuration that balances attractive (shown as springs in v.a) and repulsive (not shown) forces between pairs of points as a function of edge weight or lack thereof. This procedure iteratively minimizes the cross-entropy between the low-dimensional and high-dimensional graphs (v.b). The communities found through Louvain are then combined with the graph layout procedure to arrive at a set of clusters in a low-dimensional embedded space (vi). These clusters (vi, top) can be used to classify the original waveforms (vi, bottom). To investigate 'why' these data points became clusters, each cluster is examined for locally (within-cluster) important features (SHAP *Lundberg and Lee, 2017*), (vii) and globally important trends (UMAP inverse transform, viii). Not shown is the classifier SHAP values are calculated from. The diagrams for the graph construction and layout are based on UMAP documentation and the diagram for Louvain community detection is based on *Blondel et al., 2008*. *Figure 2—figure supplement 1*: An intuitive diagram of local and global distance preservation in UMAP and a schematic of the Louvain clustering process. The online version of this article includes the following figure supplement(s) for figure 2:

**Figure supplement 1.** Diagrams of UMAP and Louvain community detection.

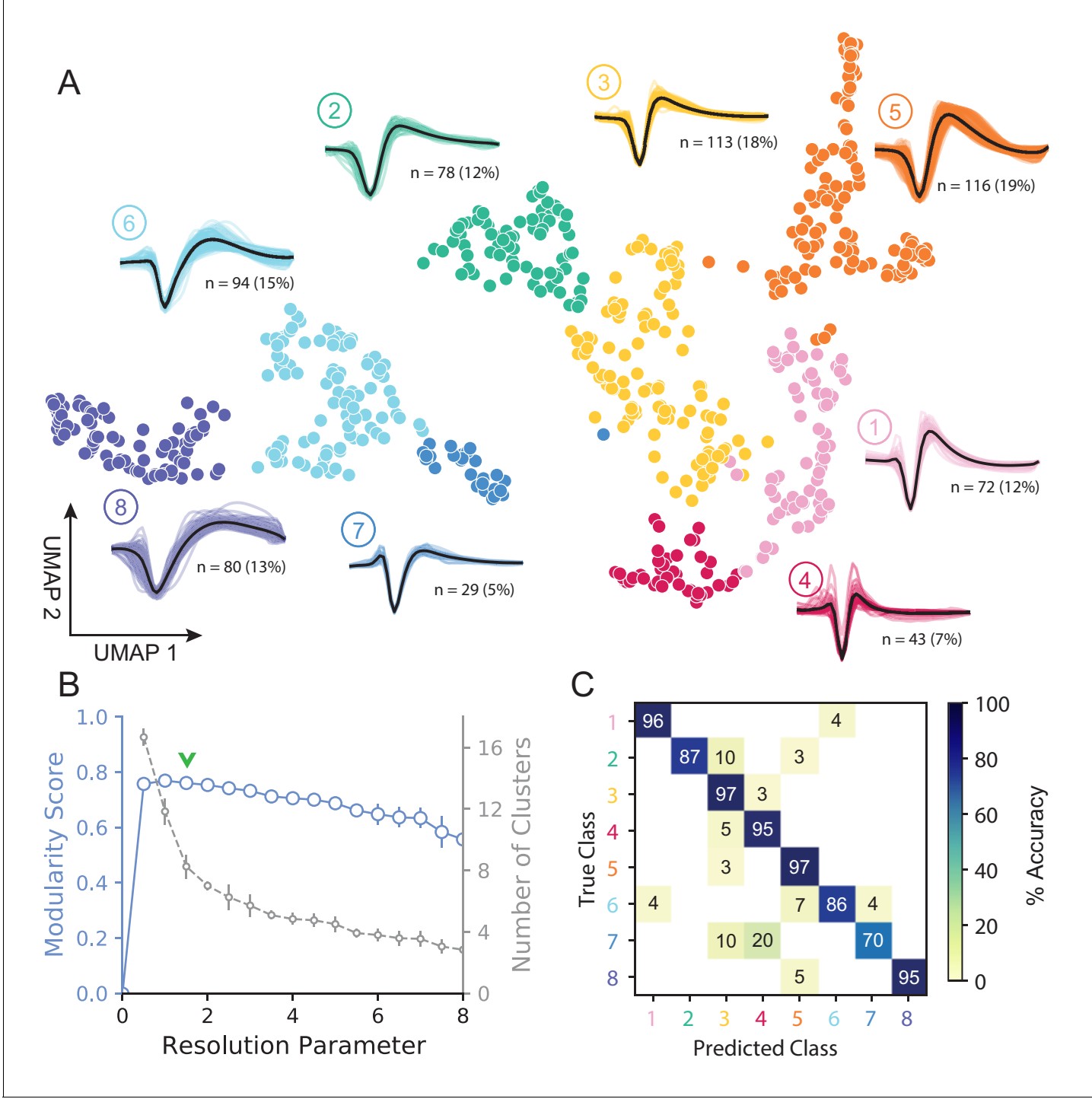

**Figure 3.** UMAP and Louvain clustering reveal a robust diversity of averaged single-unit waveform shapes. (**A**) Scatter plot of normalized EAP waveforms in UMAP space colored by Louvain cluster membership. Adjacent to each numbered cluster (① through ⑧) is shown all member waveforms and the average waveform shape (in black). Each waveform is 1.6 ms. Percentages do not add to 100% due to rounding. (**B**) Louvain clustering resolution parameter versus modularity score (in blue, axis at left) and the number of clusters (communities) found (in gray, axis at right). This was averaged over 25 runs for *WaveMAP* using 25 random samples and seeds of 80% of the full dataset at each resolution parameter from 0 to 8 in 0.5 unit increments (a subset of the data was used to obtain error bars). Each data point is the mean ± S.D. with many S.D. bars smaller than the marker size. Green chevrons indicate the resolution parameter of 1.5 chosen and its position along both curves. (**C**) The confusion matrix of a gradient boosted decision tree classifier with 5-fold cross-validation and hyperparameter optimization. The main diagonal shows the held-out classification accuracy for each cluster and the off-diagonals show the misclassification rates for each cluster to each other cluster. The average accuracy across all clusters was

*Figure 3 continued on next page*

*Figure 3 continued*

91%. ***Figure 3—figure supplement 1***: A stability analysis of *WaveMAP* clustering showing solutions are stable with respect to random seed, random data subset, and in an ensembled version of Louvain. ***Figure 3—figure supplement 2***: Different amplitude normalizations have similar effect but this processing is essential to *WaveMAP* extracting meaningful structure. ***Figure 3—figure supplement 3***: Pre-processing waveform data with principal component analysis does not alter *WaveMAP* results.

The online version of this article includes the following figure supplement(s) for figure 3:

**Figure supplement 1.** Stability analysis of *WaveMAP*: (**A**) *WaveMAP* instantiated with three different UMAP random seeds (each row is a different seed) and Louvain resolution parameters.

**Figure supplement 2.** Different amplitude normalizations have similar effect but are essential for meaningful *WaveMAP* structure.

**Figure supplement 3.** Pre-processing with PCA does not alter *WaveMAP* structure: the full dataset was pre-processed with principal component analysis (PCA) and projected into the space of the first three principal components.

points, 1.6 ms time length) passed to *WaveMAP* and used a test-train split with k-fold cross-validation applied to the training data. Hyperparameters were tuned with a 5-fold cross-validated grid search on the training data and final parameters shown in *Table 1*. After training, the classification was evaluated against the held-out test set (which was never seen in model training/tuning) and the accuracy, averaged over clusters, was 91%. *Figure 3C* shows the associated confusion matrix which contains accuracies for each class along the main diagonal and misclassification rates on the off-diagonals. Such successful classification at high levels of accuracy was only possible because there were 'generalizable' clusterings of similar waveform shapes in the high-dimensional space revealed by UMAP.

We find that cluster memberships found by *WaveMAP* are stable with respect to random seed when resolution parameter and n_neighbors parameter are fixed. This stability of *WaveMAP* clusters with respect to random seed is because much of the variability in UMAP layout is the result of the projection process (*Figure 2B–v.*a). Louvain clustering operates before this step on the high-dimensional graph generated by UMAP which is far less sensitive to the random seed. Thus, the actual layout of the projected clusters might differ subtly according to random seed, but the cluster memberships largely do not (see Supplementary Information and columns of *Figure 3—figure supplement 1A*). Here, we fix the random seed purely for visual reproducibility purposes in the figure. Thus, across different random seeds and constant resolution, the clusters found by *WaveMAP* did not change because the graph construction was consistent across random seed at least on our dataset (*Figure 3—figure supplement 1A*).

We also found that *WaveMAP* was robust to data subsetting (randomly sampled subsets of the full dataset, see Supplementary Information *Tibshirani and Walther, 2005*), unlike other clustering approaches (*Figure 3—figure supplement 1B*, green, *Figure 4—figure supplement 1*). We applied *WaveMAP* to 100 random subsets each from 10% to 90% of the full dataset and compared this to a 'reference' clustering produced by the procedure on the full dataset. WaveMAP was consistent in both cluster number (*Figure 3—figure supplement 1B*, red) and cluster membership (which waveforms were frequently 'co-members' of the same cluster; *Figure 3—figure supplement 1B*, green).

Finally, our results were also robust to another standard approach to normalizing spike waveforms: normalization to trough depth. This method exhibited the same stability in cluster number (*Figure 3—figure supplement 2C*), and also showed no differences in downstream analyses (*Figure 3—figure supplement 2D*). Without amplitude normalization, interesting structure was lost (*Figure 3—figure supplement 2E*) because UMAP likely attempts to explain both waveform amplitude and shape (shown as a smooth gradient in the trough to peak height difference *Figure 3—figure supplement 2F*). In addition, common recommendations to apply PCA before non-linear dimensionality reduction were not as important for our waveform dataset, which was fairly low-dimensional (first three PC's explained 94% variance). Projecting waveforms into a three-dimensional PC-space before *WaveMAP* produced a clustering very similar to data without this step (*Figure 3—figure supplement 3*).

## Traditional clustering methods with specified features sub-optimally capture waveform diversity

Our unsupervised approach (*Figure 3*) generates a stable clustering of waveforms. However, is our method better than the traditional approach of using specified features (*Snyder et al., 2016*;

*Trainito et al., 2019*; *Kaufman et al., 2010*; *Kaufman et al., 2013*; *Mitchell et al., 2007*; *Barthó et al., 2004*; *Merchant et al., 2008*; *Merchant et al., 2012*; *Song and McPeek, 2010*; *Simons, 1978*; *Johnston et al., 2009*)? To compare how *WaveMAP* performs relative to traditional clustering methods built on specified features, we applied a Gaussian mixture model (GMM) to the three-dimensional space produced by commonly used waveform features. In accordance with previous work, the features we chose (*Figure 4A*) were action potential (AP) width of the spike (width in milliseconds of the full-width half minimum of the depolarization trough *Vigneswaran et al., 2011*); the peak ratio the ratio of pre-hyperpolarization peak (A1) to the post-hyperpolarization peak (A2) *Barry, 2015*; and the trough to peak duration (time in ms from the depolarization trough to post-hyperpolarization peak) which is the most common feature used in analyses of extracellular recordings (*Snyder et al., 2016*; *Kaufman et al., 2013*; *Merchant et al., 2012*).

The result of the GMM applied to these three measures is shown in *Figure 4B*. This method identified four waveform clusters that roughly separated into broad-spiking (BS, ~33%, n = 208), narrow-spiking (NS, ~43%, n = 269), broad-spiking triphasic (BST, ~9%, n = 55), and narrow-spiking triphasic (NST, ~15%, n = 93) (*Figure 4B*). Triphasic waveforms, thought to be neurons with myelinated axons or neurites (*Barry, 2015*; *Robbins et al., 2013*; *Deligkaris et al., 2016*; *Bakkum et al., 2013*; *Sun et al., 2021*), contain an initial positive spike before the trough and can be identified by large peak ratios (*Figure 4A*). These GMM clusters are similar to those obtained from other clusterings of EAP's in macaque cortex (*Gur et al., 1999*; *Trainito et al., 2019*). We selected four clusters by examining the Bayesian information citerion (BIC) statistic as a function of the number of clusters and identified the cluster number at the elbow (green chevron in *Figure 4C*).

To compare the generalizability of this representation with the representation provided by UMAP, we trained the same decision tree classifier on the waveform data (after separate hyperparameter tuning, *Table 1*) but this time using the four GMM classes as target labels. After training, the accuracy across all four classes averaged ~78% with no classification accuracy over 95% and misclassifications between every class (*Figure 4D*). The classifier trained on specified features under-performed the classifier trained on the whole waveform found by *WaveMAP*. In *WaveMAP*, the individual classification accuracy of most classes exceeded 95% with few misclassifications between groups even though there were double the number of clusters. This result suggests that the clusters based on specified features are less differentiable than *WaveMAP* clusters even when a much lower cluster number is considered.

This deficit can be understood as an inability of the GMM to fully capture the latent structure of the data. If we examine the gray data point shadows (*Figure 4B*), no features contain clear clusters and neither do they contain Gaussian distributions which is an assumption of the GMM model. Examining the marginal distributions in *Figure 4B*, none of the features induce a clear separability between the clusters alone or in conjunction. Furthermore, the reproducible clusters found by *WaveMAP* are linearly inseparable in the feature space of the three GMM features (*Figure 4—figure supplement 2A*). Note, this is not an artifact of using a lower cluster number in the GMM as opposed to the eight found by *WaveMAP*. Even if the GMM is instantiated with eight clusters (*Figure 4—figure supplement 2B*), a classifier is still unable to generalize this clustering with even modest accuracy (average of 56% across clusters; *Figure 4—figure supplement 2C*) even if the waveforms shapes found by the GMM with eight clusters seem somewhat sensible (*Figure 4—figure supplement 2D*). In fact, across all cluster numbers (n_components from 2 to 16), a classifier tuned *for the GMM* performed more poorly on the GMM labels than a *WaveMAP* projection with the same number of clusters (*Figure 4—figure supplement 2E*, in red). Tuning *WaveMAP* parameters that induce different cluster numbers, whether n_neighbors (in dark blue) or resolution (in light blue), had little effect on classifier performance (*Figure 4—figure supplement 2E*, in blues). *WaveMAP* yielded mappings that were more generalizable than a GMM on features across every number of clusters and both parameters investigated. Thus, it is a deficit of the GMM on constructed feature-based approach to capture the full diversity of waveforms, especially at high cluster number, and not a peculiarity of the model parameters chosen or number of clusters induced.

We also investigated the representation of specified features in the projected UMAP space. We color coded the waveforms in UMAP, in *Figure 5—figure supplement 1*, according to each point's feature values using the same features as in *Figure 4* (*Figure 5—figure supplement 1A*): AP width (*Figure 5—figure supplement 1B*), trough to peak duration (*Figure 5—figure supplement 1C*), and peak ratio (*Figure 5—figure supplement 1D*). We find that *WaveMAP* implicitly captures each of

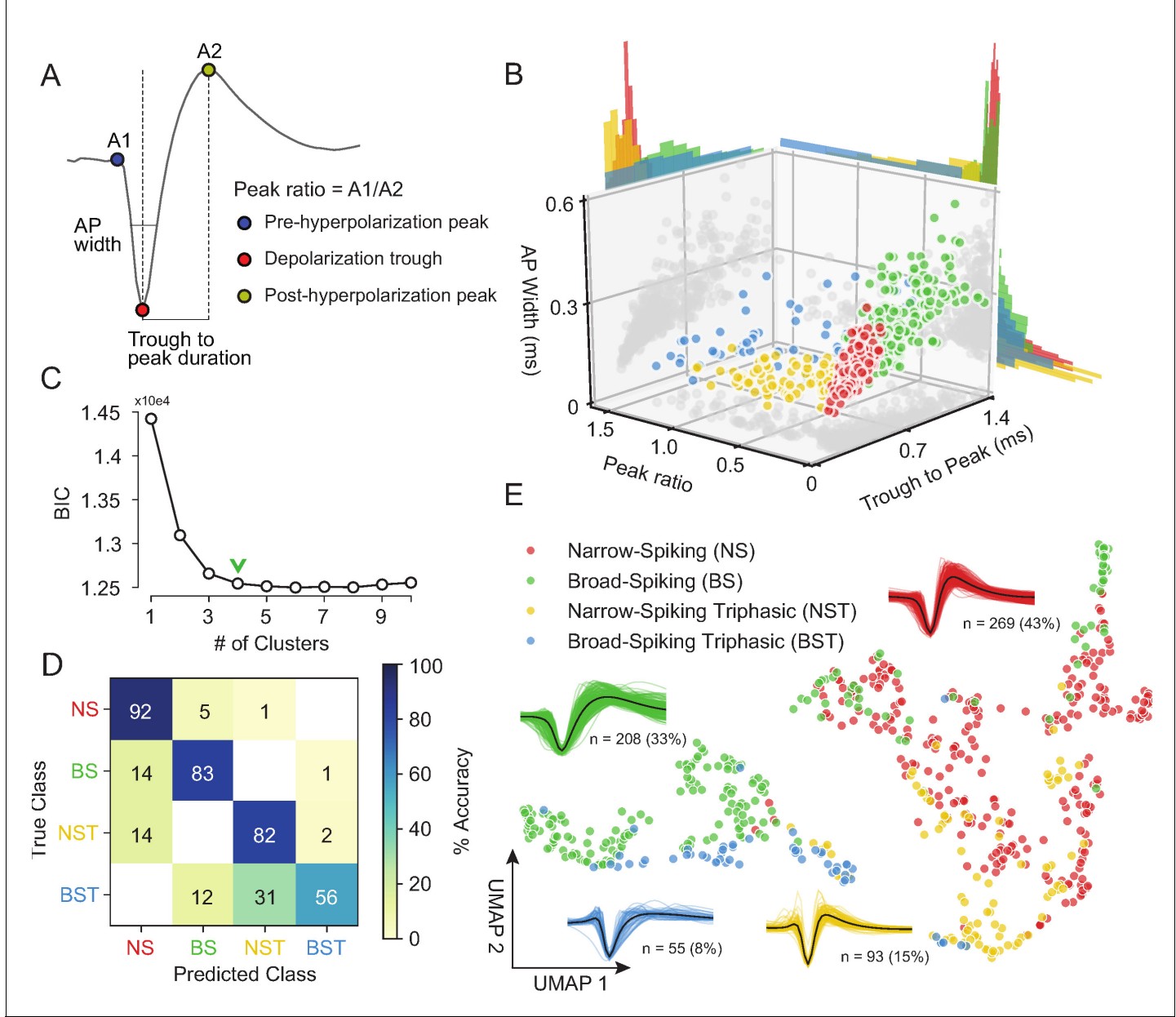

**Figure 4.** Gaussian mixture model clustering on specified features fails to capture the breadth of waveform diversity. (A) The three EAP waveform landmarks used to generate the specified features passed to the GMM on a sample waveform. ● is the pre-hyperpolarization peak (A1); ● is the depolarization trough; and ● is the post-hyperpolarization peak (A2). (B) A three-dimensional scatter plot with marginal distributions of waveforms and GMM classes on the three specified features in (A). Narrow-spiking (NS) are in red; broad-spiking (BS) in green; narrow-spiking triphasic (NST) in yellow; and broad-spiking triphasic (BST) types are in blue. Trough to peak was calculated as the time between ● and ●; peak ratio was determined as the ratio between the heights of ● and ● (A1/A2); and AP width was determined as the width of the depolarization trough ● using the MLIB toolbox (**Stuttgen, 2019**). (C) The optimal cluster number in the three-dimensional feature space in (B) was determined to be four clusters using the Bayesian information criterion (BIC) (**Trainito et al., 2019**). The number of clusters was chosen to be at the 'elbow' of the BIC curve (green chevron). (D) A confusion matrix for a gradient boosted decision tree classifier with 5-fold cross-validation with hyperparameter optimization. The main diagonal contains the classification accuracy percentages across the four GMM clusters and the off-diagonal contains the misclassification rates. The average accuracy across classes was 78%. (E) The same scatter plot of normalized EAP waveforms in UMAP space as in **Figure 3A** but now colored by GMM category. **Figure 4—figure supplement 1**: We show that WaveMAP clusterings are more consistent across random data subsets than either DBSCAN on t-SNE or a GMM on PCA. **Figure 4—figure supplement 2**: GMMs fail to full capture the latent structure in the waveforms.

The online version of this article includes the following figure supplement(s) for figure 4:

**Figure supplement 1.** WaveMAP clusterings are more consistent than either DBSCAN on t-SNE or a GMM on PCA.
**Figure supplement 2.** Comparison of GMM and UMAP in the constructed feature space.

these specified features shown as a smooth gradient of values. Our method also exposes the correlation between certain specified features: the gradient between trough to peak duration and AP width points point roughly in the same direction so thus both features are highly correlated. This correlation between features exposes their redundancy and is another reason why traditional approaches fail to capture the full diversity of waveform shapes.

To obtain a clearer picture of how *WaveMAP* captures latent structure missed by specified features, we color the points in UMAP space by their GMM cluster identity in *Figure 4E*. Here, *WaveMAP* is able to recapitulate the same structure observed by specified features as a gradient from broad- to narrow-spiking along the UMAP-1 direction. Our technique also captures the transition from triphasic to biphasic along the UMAP-2 direction. *WaveMAP* is also able to find clusters that occupy an intermediate identity between GMM classes. For instance, *WaveMAP* cluster ② (*Figure 3A*) is nearly equal parts broad- and narrow-spiking in the GMM clustering (*Figure 4E*). If a GMM were used, ② would be split between two classes despite it having a distinct waveform shape characterized by a small pre-hyperpolarization peak, a moderate post-hyperpolarization peak, and relatively constant repolarization slope.

## *WaveMAP* interpretably recapitulates and expands upon known waveform features

We have established that *WaveMAP* has the ability to discover extracellular waveform clusters, but a common contention with such non-linear methods is that they are uninterpretable. Here, using an interpretable machine learning approach, we show that *WaveMAP* produces sensible results (*Molnar, 2020*; *Azodi et al., 2020*). To identify the features our algorithm is paying attention to, we first computed the inverse mapping of the UMAP transform to probe the projected space in a systematic way. Second, we leverage the gradient boosted decision tree classifier in *Figure 3C* and used a decision tree implementation (path-dependent TreeSHAP *Lundberg et al., 2018*) of SHapley Additive exPlanations (SHAP values *Lundberg and Lee, 2017*; *Lundberg et al., 2020*) to reveal what waveform features are implicitly used to differentiate clusters.

To quantify the differences between Louvain clusters, we applied a grid of 'test points' to the UMAP projected space (*Figure 5A*, top) and inverted the transform at each location; each of these test points is a coordinate on a grid (black x's) and shows the waveform associated with every point in the projected space (*Figure 5A*, bottom). On the bottom of *Figure 5A* is shown the waveform that corresponds to each point in UMAP space color-coded to the nearest cluster or to gray if there were no nearby clusters. As UMAP-1 increases, there is a smooth transition in the sign of the inflection of the repolarization slope (the second derivative) from negative to positive (slow to fast repolarization rate). That is, the post-hyperpolarization peak becomes more sharp as we increase in the UMAP-1 direction. As UMAP-2 increases, we see a widening of the post-hyperpolarization slope distinct from the change in its inflection (UMAP-1). These two UMAP dimensions recapitulate the known importance of hyperpolarization properties in clustering waveforms. Both hyperpolarization rate (proportional to trough to peak width) and hyperpolarization slope inflection (proportional to repolarization time) are separate but highly informative properties (*Trainito et al., 2019*; *Ardid et al., 2015*). Furthermore, since repolarization rate and post-hyperpolarization width associate with different UMAP dimensions, this implies that these two processes are somewhat independent factors shaping the waveform. Repolarization rates are governed by potassium channel dynamics and may play an important in waveform shape (*Soares et al., 2017*). Thus, *WaveMAP* not only finds an interpretable and smoothly varying low-dimensional space it also offers biological insights; in this case, how cell types might differ according to channel protein expression and dynamics.

In *Figure 5B*, we made use of SHAP values to identify which aspects of waveform shape the gradient boosted decision tree classifier utilizes in assigning what waveform to which cluster (*Lundberg and Lee, 2017*; *Lundberg et al., 2020*). SHAP values build off of the game theoretic quantity of Shapley values (*Shapley, 1988*; *Štrumbelj and Kononenko, 2014*), which poses that each feature (point in time along the waveform) is of variable importance in influencing the classifier to decide whether the data point belongs to a specific class or not. Operationally, SHAP values are calculated by examining the change in classifier performance as each feature is obscured (the waveform's amplitude at each time point in this case), one-by-one (*Lundberg and Lee, 2017*). *Figure 5B* shows the top-10 time points in terms of mean absolute SHAP value (colloquially called 'SHAP

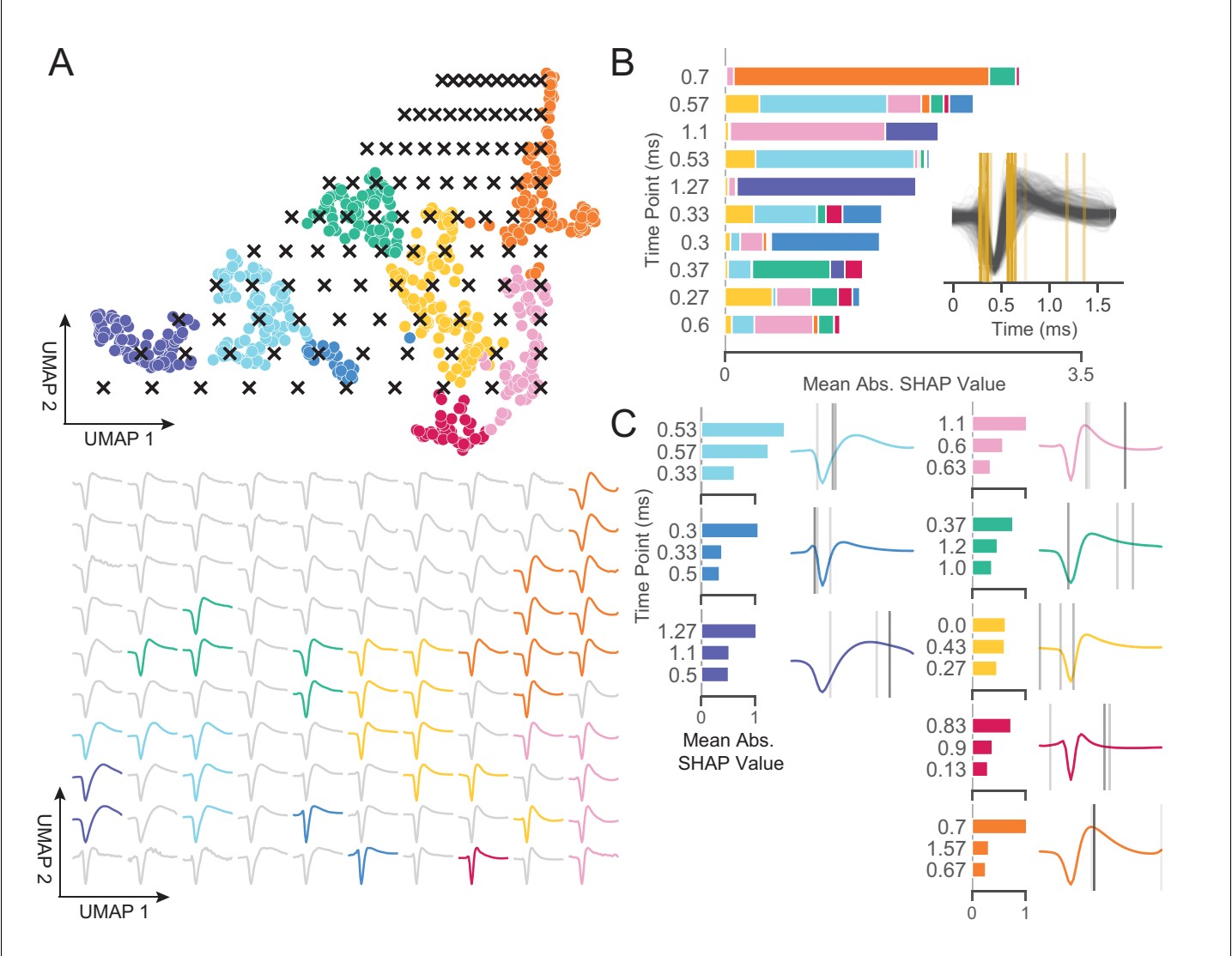

**Figure 5.** WaveMAP provides interpretable representations that both validate and extend known and unknown features importances. (**A**) *WaveMAP* applied to the EAP's as in *Figure 3A* but overlaid with a grid of test points (black x's, top) spanning the embedded space. At bottom, the inverse UMAP transform is used to show the predicted waveform at each test point. For each x above, the predicted waveform is shown, plotted, and assigned the color of the nearest cluster or in gray if no cluster is nearby. Note that there exists instability in the waveform shape (see waveforms at corners) as test points leave the learned embedded space. (**B**) The mean absolute SHAP values for 10 time points along all waveforms subdivided according to the SHAP values contributed by each *WaveMAP* cluster. These SHAP values were informed by applying path-dependent TreeSHAP to a gradient boosted decision tree classifier trained on the waveforms with the *WaveMAP* clusters as labels. In the inset, all waveforms are shown and in gold are shown the time points for which the SHAP values are shown on the left. Each vertical line is such that the most opaque line contains the greatest SHAP value across *WaveMAP* clusters; the least opaque, the smallest SHAP value. (**C**) Each averaged *WaveMAP* waveform cluster is shown with the three time points containing the greatest SHAP values for each cluster individually. As before, the SHAP value at each time point is proportional to the opacity of the gray vertical line also shown as a bar graph at left. *Figure 5—figure supplement 1*: *WaveMAP* implicitly captures waveform features (such as trough to peak or AP width) without the need for prior specification.

The online version of this article includes the following figure supplement(s) for figure 5:

**Figure supplement 1.** WaveMAP implicitly captures waveform features without the need for specification.

value') and their location. It is important to note that not every time point is equally informative for distinguishing every cluster individually and thus each bar is subdivided into the mean absolute SHAP value contribution of the eight constituent waveform classes. For instance, the 0.7 ms location is highly informative for cluster ⑤ and the 0.3 ms point is highly informative for cluster ⑦ (*Figure 5C*).

In the inset is shown all waveforms along with each of the top ten time points (in gold) with higher SHAP value shown with more opacity. The time points with highest SHAP value tend to cluster around two different locations giving us an intuition for which locations are most informative for telling apart the Louvain clusters. For instance, the 0.5 to 0.65 ms region contains high variability amongst waveforms and is important in separating out broad- from narrow-spiking clusters. This region roughly contains the post-hyperpolarization peak which is a feature of known importance and incorporated into nearly every study of EAP waveform shape (see *Table 1* in *Vigneswaran et al., 2011*). Similarly, SHAP values implicate the region around 0.3 ms to 0.4 ms as time points that are also of importance and these correspond to the pre-hyperpolarization peak which is notably able to partition out triphasic waveforms (*Barry, 2015*). Importance is also placed on the location at 0.6 ms corresponding to the inflection point which is similarly noted as being informative (*Trainito et al., 2019*; *Banaie Boroujeni et al., 2021*). These methods also implicate other regions of interest that have not been previously noted in the literature to the best of our knowledge: two other locations are highlighted farther along the waveform at 1.1 and 1.27 ms and are important for differentiating ⑧ and ① from the other waveforms. This result suggests that using only up to 1.0 ms or less of the waveform may obscure diversity.

In *Figure 5C*, we show the three locations that are most informative for delineating a specific cluster; these appear as gray lines with their opacity proportional to their SHAP importance. These individually informative features often do align with those identified as globally-informative but do so with cluster-specific weights. Put another way, not every time point is equally informative for identifying waveforms individually and these 'most informative' parts of each waveform do not always perfectly align with globally informative features. In summary, *WaveMAP* independently and sensibly arrived at a more nuanced incorporation of the very same features identified in previous work—and several novel ones—using a completely unsupervised framework which obviated the need to specify waveform features.

In the second half of the paper, we investigate whether these clusters have distinct physiological (in terms of firing rate), functional, and laminar distribution properties which could give credence that *WaveMAP* clusters connect to cell types.

## *WaveMAP* clusters have distinct physiological properties

A defining aspect of cell types is that they vary in their physiology and especially firing rate properties (*Mountcastle et al., 1969*; *McCormick et al., 1985*; *Connors and Gutnick, 1990*; *Nowak et al., 2003*; *Connors et al., 1982*; *Contreras, 2004*). However, these neuronal characterizations via waveform ex vivo are not always conserved when the same waveform types are observed in vivo during behavior (*Steriade, 2004*; *Steriade et al., 1998*). To connect our waveform clusters to physiological cell types in vivo, we identified each cluster's firing rate properties. We performed several analyses using the firing rate (FR) in spikes per second (spikes/s) for each cluster during the decision-making task described in *Figure 1*.

The trial-averaged FRs are aligned to stimulus onset (stim-aligned) and separated into preferred (PREF, solid trace) or non-preferred (NONPREF, dashed trace) reach direction trials. This is shown for both broad- (*Figure 6A*) and narrow-spiking (*Figure 6B*) clusters. A neuron's preferred direction (right or left) was determined as the reach direction in which it had a higher FR on average in the 100 ms time period before movement onset.

To further quantify the FR differences between clusters, we calculated three properties of the FR response to stimulus: baseline FR, max FR, and FR range.

### Baseline FR

Cell types are thought to demonstrate different baseline FRs. We estimated baseline FR (*Figure 6C*) as the median FR across the 200 ms time period before the appearance of the red-green checkerboard and during the hold period after targets appeared for the broad (*Figure 6A*), and narrow-spiking clusters (*Figure 6B*). The broad-spiking clusters showed significant differences in baseline FR when compared against the narrow-spiking clusters (p = 0.0028, Mann-Whitney *U* test). Similar patterns were observed in another study of narrow- vs. broad-spiking neurons in PMd during an instructed delay task (*Kaufman et al., 2010*). We also found that not all broad-spiking neurons had low baseline FR and not all narrow-spiking neurons had high baseline FR. The broad-spiking clusters

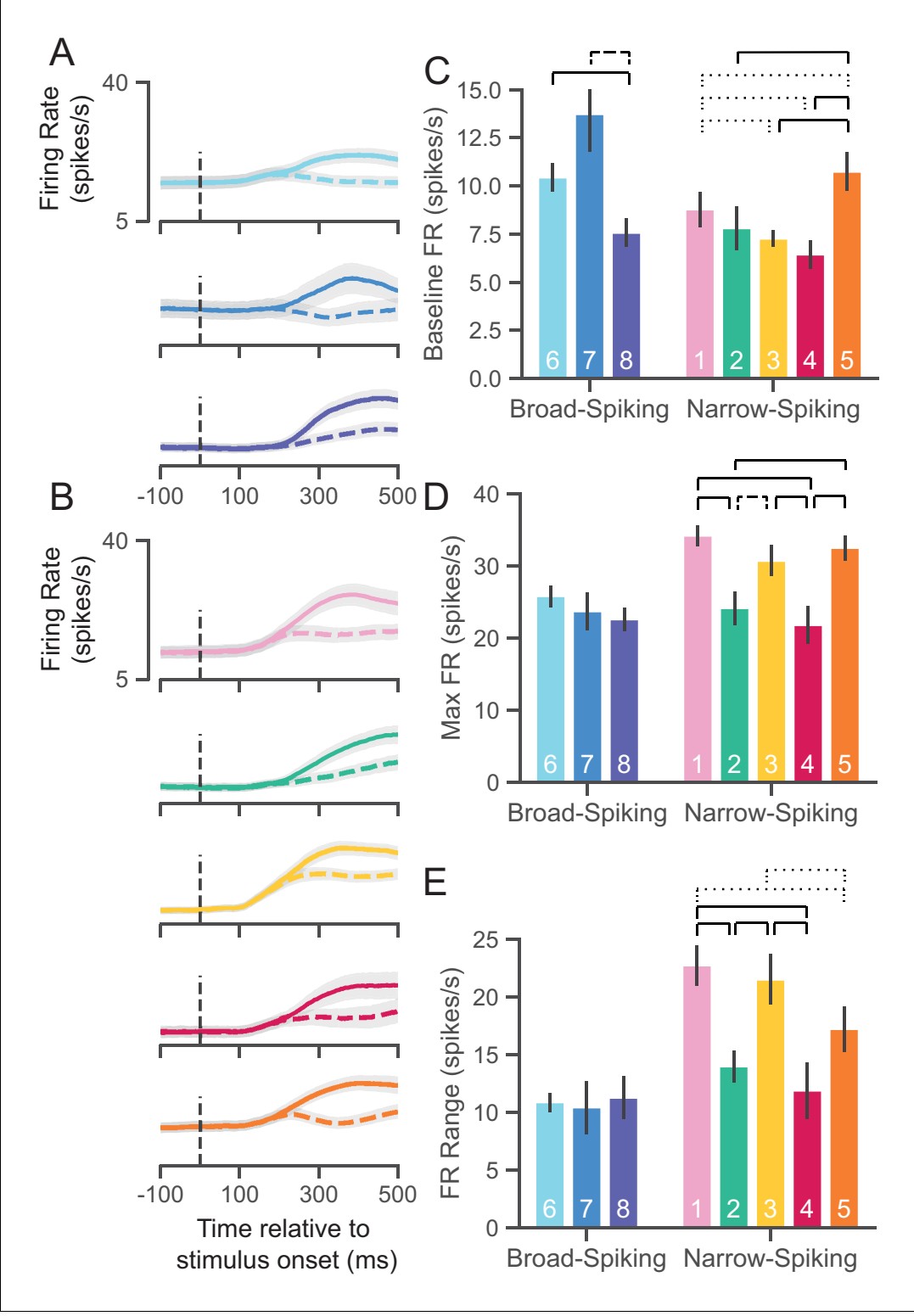

**Figure 6.** UMAP clusters exhibit distinct physiological properties. (A) Stimulus-aligned trial-averaged firing rate (FR; spikes/s) activity in PMd for broad-spiking *WaveMAP* clusters. The traces shown are separated into trials for PREF direction reaches (solid traces) and NONPREF direction reaches (dashed traces) and across the corresponding *WaveMAP* clusters. Shaded regions correspond to bootstrapped standard error of the mean. Dashed vertical line is stimulus-onset time. (B) The same plots as in (A) but for narrow-spiking *WaveMAP* clusters. (C) Baseline median FR ± S.E.M. for the neurons in the eight different classes. Baselines were taken as the average

*Figure 6 continued on next page*

*Figure 6 continued*

FR from 200 ms of recording before checkerboard stimulus onset. (D) Median maximum FR ± S.E.M. for the neurons in the eight different clusters. This was calculated by taking the median of the maximum FR for each neuron across the entire trial. (E) Median FR range ± S.E.M. calculated as the median difference, per neuron, between its baseline and max FR. —— p < 0.05; —— p < 0.01; —— p < 0.005; Mann-Whitney *U* test, FDR adjusted. *Figure 6—figure supplement 1*: GMM clusters are less physiologically distinguishable than *WaveMAP* clusters.

The online version of this article includes the following figure supplement(s) for figure 6:

**Figure supplement 1.** GMM clusters are less physiologically distinguishable than *WaveMAP* clusters.

---

⑥ and ⑦ were not significantly different but both differed significantly from ⑧ in that their baseline FR was much higher (10.3 ± 0.7 and 13.2 ± 1.9 spikes/s vs. 7.6 ± 0.75 spikes/s [median ± bootstrap S. E.]; p = 0.0052, p = 0.0029 respectively, Mann-Whitney *U* test, FDR adjusted). The narrow-spiking clusters (*Figure 6B*, right) ②, ③, and ④ had relatively low median baseline FRs (7.5 ± 1.1, 7.4 ± 0.4, 6.5 ± 0.7 spikes/s, median ± bootstrap S.E.) and were not significantly different from one another but all were significantly different from ① and ⑤ (p = 0.04, p = 2.8e-4, p = 2.8e-7, p = 4.9e-5, respectively, Mann-Whitney *U* test, FDR adjusted; see *Figure 6C*).

## Maximum FR

A second important property of cell types is their maximum FR (*Mountcastle et al., 1969*; *McCormick et al., 1985*; *Connors and Gutnick, 1990*). We estimated the maximum FR for a cluster as the median of the maximum FR of neurons in the cluster in a 1200 ms period aligned to movement onset (800 ms before and 400 ms after movement onset; *Figure 6D*). In addition to significant differences in baseline FR, broad- vs. narrow-spiking clusters showed a significant difference in max FR (p = 1.60e-5, Mann-Whitney *U* test). Broad-spiking clusters were fairly homogeneous with low median max FR (24.3 ± 1.0, median ± bootstrap S.E.) and no significant differences between distributions. In contrast, there was significant heterogeneity in the FR's of narrow-spiking neurons: three clusters (①, ③, and ⑤) had uniformly higher max FR (33.1 ± 1.1, median ± bootstrap S.E.) while two others (② and ④) were uniformly lower in max FR (23.0 ± 1.4, median ± bootstrap S.E.) and were comparable to the broad-spiking clusters. Nearly each of the higher max FR narrow-spiking clusters were significantly different than each of the lower max FR clusters (all pairwise relationships p < 0.001 except ③ to ④ which was p = 0.007, Mann-Whitney *U* test, FDR adjusted).

## FR range

Many neurons, especially inhibitory types, display a sharp increase in FR and also span a wide range during behavior (*Kaufman et al., 2010*; *Kaufman et al., 2013*; *Chandrasekaran et al., 2017*; *Johnston et al., 2009*; *Hussar and Pasternak, 2009*). To examine this change over the course of a trial, we took the median difference across trials between the max FR and baseline FR per neuron to calculate the FR range. We again found the group difference between broad- and narrow-spiking clusters to be significant (p = 0.0002, Mann-Whitney *U* test). Each broad-spiking cluster (⑥, ⑦, and ⑧) had a median increase of around 10.8 spikes/s (10.8 ± 0.8, 10.7 ± 2.3, and 10.9 ± 1.9 spikes/s respectively, median ± bootstrap S.E.) and each was nearly identical in FR range differing by less than 0.2 spikes/s. In contrast, the narrow-spiking clusters showed more variation in their FR range— similar to the pattern observed for max FR. ①, ③, and ⑤ had a large FR range (20.3 ± 1.1 spikes/s, median ± bootstrap S.E.) and the clusters ③ and ④ had a relatively smaller FR range (13.4 ± 1.3 spikes/s, median ± bootstrap S.E.). These results demonstrate that some narrow-spiking clusters, in addition to having high baseline FR, highly modulated their FR over the course of a behavioral trial.

Such physiological heterogeneity in narrow-spiking cells has been noted before (*Ardid et al., 2015*; *Banaie Boroujeni et al., 2021*; *Quirk et al., 2009*) and in some cases, attributed to different subclasses of a single inhibitory cell type (*Povysheva et al., 2013*; *Zaitsev et al., 2009*). Other work also strongly suggests that narrow-spiking cells contain excitatory neurons with distinct FR properties contributing to this diversity (*Vigneswaran et al., 2011*; *Onorato et al., 2020*).

Furthermore, if *WaveMAP* has truly arrived at a closer delineation of underlying cell types compared to previous methods, it should produce a 'better' clustering of physiological properties

beyond just a better clustering of waveform shape. To address this issue, we calculate the same firing rate traces and physiological properties as in *Figure 6* but with the GMM clusters (*Figure 6—figure supplement 1*). While the FR traces maintain the same trends (BS does not increase its FR prior to the split into PREF and NONPREF while NS does; compare to *WaveMAP* broad-spiking vs. narrow-spiking clusters respectively), much of the significant differences between clusters is lost across all physiological measures even though fewer groups are compared (*Figure 6—figure supplement 1B,C and D*). We also quantitatively estimate these differences by calculating the effect sizes (Cohen's $f^2$) across the *WaveMAP* and GMM clusterings with a one-way ANOVA. The effect size was larger for *WaveMAP* vs. GMM clustering respectively for every physiological property: baseline FR (0.070 vs. 0.013), maximum FR (0.035 vs. 0.011), and FR range (0.055 vs. 0.034).

## *WaveMAP* clusters have distinct decision-related dynamics

Our analysis in the previous section showed that there is considerable heterogeneity in their physiological properties. Are these putative cell types also functionally different? Prior literature argues that neuronal cell types have distinct functional roles during cortical computation with precise timing. For instance, studies of macaque premotor (*Song and McPeek, 2010*), inferotemporal (IT) (*Mruczek and Sheinberg, 2012*), and frontal eye field (FEF) (*Ding and Gold, 2012*) areas show differences in decision-related functional properties: between broad- and narrow-spiking neurons, narrow-spiking neurons exhibit choice-selectivity earlier than broad-spiking neurons. In the mouse, specific aspects of behavior are directly linked with inhibitory cell types (*Pinto and Dan, 2015*; *Estebanez et al., 2017*). Here, we examine the functional properties of each cluster based on two inferred statistics: choice-related dynamics and discrimination time.

### Choice-related dynamics

The first property we assessed for these *WaveMAP* clusters was the dynamics of the choice-selective signal. The neural prediction made by computational models of decision-making (for neurons that covary with an evolving decision) is the build-up of average neural activity in favor of a choice is faster for easier compared to harder color coherences (*Chandrasekaran et al., 2017*; *Ding and Gold, 2012*; *Roitman and Shadlen, 2002*). Build-up activity is measured by analyzing the rate of change of choice-selective activity vs. time. We therefore examined the differences in averaged stimulus-aligned choice-selectivity signals (defined as |left - right|) for different checkerboard color coherences for each cluster.

In *Figure 7A and B*, we show average choice-selectivity signals across the seven color coherence levels (*Figure 7A*, legend) for an example broad- (⑥) and narrow-spiking cluster (①). For ⑥ (*Figure 7A*), easier stimuli (higher coherence) only led to modest increases in the rate at which the choice selectivity signal increases. In contrast, ① (*Figure 7B*) shows faster rates for the choice-selective signal as a function of coherence. We summarized these effects by measuring the rate of change for the choice-selective signal between 175 and 325 ms for stimulus-aligned trials in each coherence condition (dashed lines in *Figure 7A,B*). This rate of rise for the choice-selective signal (spikes/s/s) vs. coherence is shown for broad- (*Figure 7C*) and narrow-spiking (*Figure 7D*) clusters. The broad-spiking clusters demonstrate fairly similar coherence-dependent changes with each cluster being somewhat indistinguishable and only demonstrating a modest increase with respect to coherence. In contrast, the narrow-spiking clusters show a diversity of responses with ① and ⑤ demonstrating a stronger dependence of choice-related dynamics on coherence compared to the other three narrow-spiking clusters which were more similar in response to broad-spiking neurons.

We further summarized these plots by measuring the dependence of the rate of rise of the choice-selective signal as a function of coherence measured as the slope of a linear regression performed on the rate of rise vs. color coherence for each cluster (*Figure 7E*). The coherence slope for broad-spiking clusters was moderate and similar to ②, ③, and ④ while the coherence slope for ① and ⑤ was steeper. Consistent with *Figure 7C and D*, the choice selective signal for ① and ⑤ showed the strongest dependence on stimulus coherence.

### Discrimination time

The second property that we calculated was the discrimination time for clusters which is defined as the first time in which the choice-selective signal (again defined as |left - right|) departed from the FR

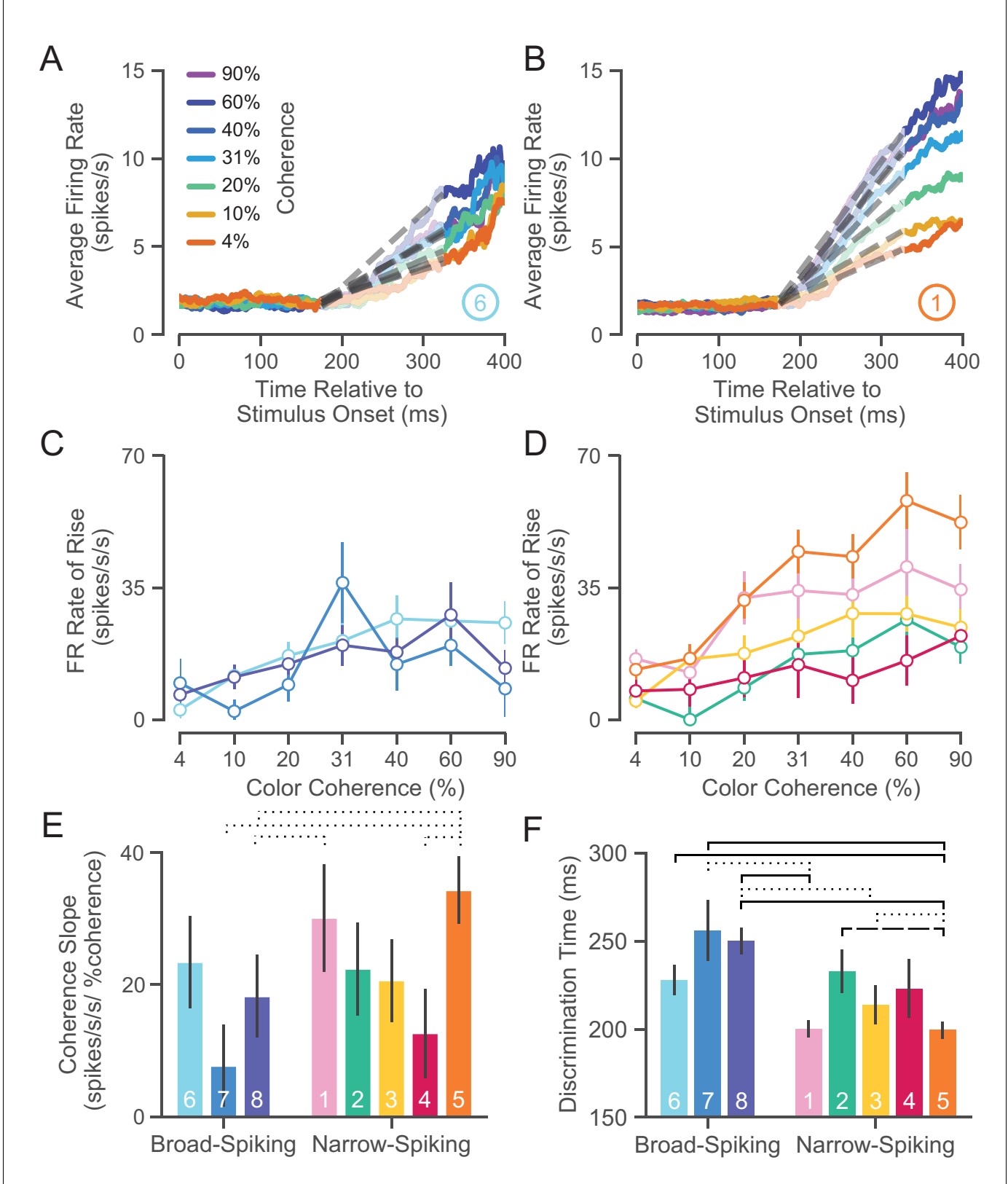

**Figure 7.** UMAP clusters exhibit distinct functional properties. (**A**) Average firing rate (FR) over time for ⑥ (used as a sample broad-spiking cluster) across trials of different color coherences. The gray-dashed lines indicate the linear regression lines used to calculate the FR rate of rise. (**B**) Average FR over time for ① (used as a sample narrow-spiking cluster) across different color coherences. (**C**) FR rate of rise vs. color coherence for broad- and (**D**)

*Figure 7 continued on next page*

*Figure 7 continued*

narrow-spiking clusters. Error bars correspond to standard error of the mean across trials. (E) Bootstrapped median color coherence slope is shown with the bootstrapped standard error of the median for each cluster on a per-neuron basis. Coherence slope is a linear regression of the cluster-specific lines in the previous plots C and D. (F) Median bootstrapped discrimination time for each cluster with error bars as the bootstrapped standard error of the median. Discrimination time was calculated as the the amount of time after checkerboard appearance at which the choice-selective signal could be differentiated from the baseline FR (*Chandrasekaran et al., 2017*). dotted line p < 0.05; dashed line p < 0.01; solid line p < 0.005; Mann-Whitney *U* test, FDR adjusted.

of the hold period. We calculated the discrimination time on a neuron-by-neuron basis by computing the first time point in which the difference in FR for the two choices was significantly different from baseline using a bootstrap test (at least 25 successive time points significantly different from baseline FR corrected for multiple comparisons *Chandrasekaran et al., 2017*). Discrimination time for broad-spiking clusters (255 ± 94 ms, median ± bootstrap S.E.) was significantly later than narrow-spiking clusters (224 ± 89 ms, p < 0.005, median ± bootstrap S.E., Mann-Whitney *U* test). Clusters ① and ⑤, with the highest max FRs (34.0 ± 1.4 and 33.0 ± 1.8 spikes/s, median ± S.E.) and most strongly modulated by coherence, had the fastest discrimination times as well (200.0 ± 4.9 and 198.5 ± 4.9 ms, median ± S.E.).

Together the analysis of choice-related dynamics and discrimination time showed that there is considerable heterogeneity in the properties of narrow-spiking neuron types. Not all narrow-spiking neurons are faster than broad-spiking neurons and choice-selectivity signals have similar dynamics for many broad-spiking and narrow-spiking neurons. ① and ⑤ have the fastest discrimination times and strongest choice dynamics. In contrast, the broad-spiking neurons have uniformly slower discrimination times and weaker choice-related dynamics.

## *WaveMAP* clusters contain distinct laminar distributions

In addition to having certain physiological properties and functional roles, numerous studies have shown that cell types across phylogeny, verified by single-cell transcriptomics, are defined by distinct patterns of laminar distribution in cortex (*Hodge et al., 2019*; *Tosches et al., 2018*). Here, we examined the laminar distributions of *WaveMAP* clusters and compared them to laminar distributions of GMM clusters. The number of waveforms from each cluster was counted at each of sixteen U-probe channels separately. These channels were equidistantly spaced every 0.15 mm between 0.0 and 2.4 mm. This spanned the entirety of PMd which is approximately 2.5 mm in depth from the pial surface to white matter (*Arikuni et al., 1988*). However, making absolute statements about layers is difficult with these measurements because of errors in aligning superficial electrodes with layer I across different days. This could lead to shifts in estimates of absolute depth; up to 0.15 mm (the distance between the first and second electrode) of variability is induced in the alignment process (see Materials and methods). However, relative comparisons are likely better preserved. Thus, we use relative comparisons to describe laminar differences between distributions and in comparison to anatomical counts in fixed tissue in later sections.

Above each column of *Figure 8A and B* are the laminar distributions for all waveforms in the associated set of clusters (in gray); below these are the laminar distributions for each cluster set's constituent clusters. On the right (*Figure 8C*), we show the distribution of all waveforms collected at top in gray with each GMM cluster's distribution shown individually below.

The overall narrow- and broad-spiking populations did not differ significantly according to their distribution (p = 0.24, Kolmogorov-Smirnov test). The broad-spiking cluster set of neurons (⑥ , ⑦ , and ⑧) are generally thought to contain cortical excitatory pyramidal neurons enriched in middle to deep layers (*Nandy et al., 2017*; *McCormick et al., 1985*). Consistent with this view, we found these broad-spiking clusters (*Figure 8A*) were generally centered around middle to deep layers with broad distributions and were not significantly distinguishable in laminarity (all comparisons p > 0.05, two-sample Kolmogorov-Smirnov test, FDR adjusted).

In contrast, narrow-spiking clusters (*Figure 8B*) were distinctly varied in their distribution such that almost every cluster had a unique laminar distribution. Cluster ① contained a broad distribution. It was significantly different in laminar distribution from clusters ② and ④ (p = 0.002 and p = 0.013, respectively, two-sample Kolmogorov-Smirnov, FDR adjusted).

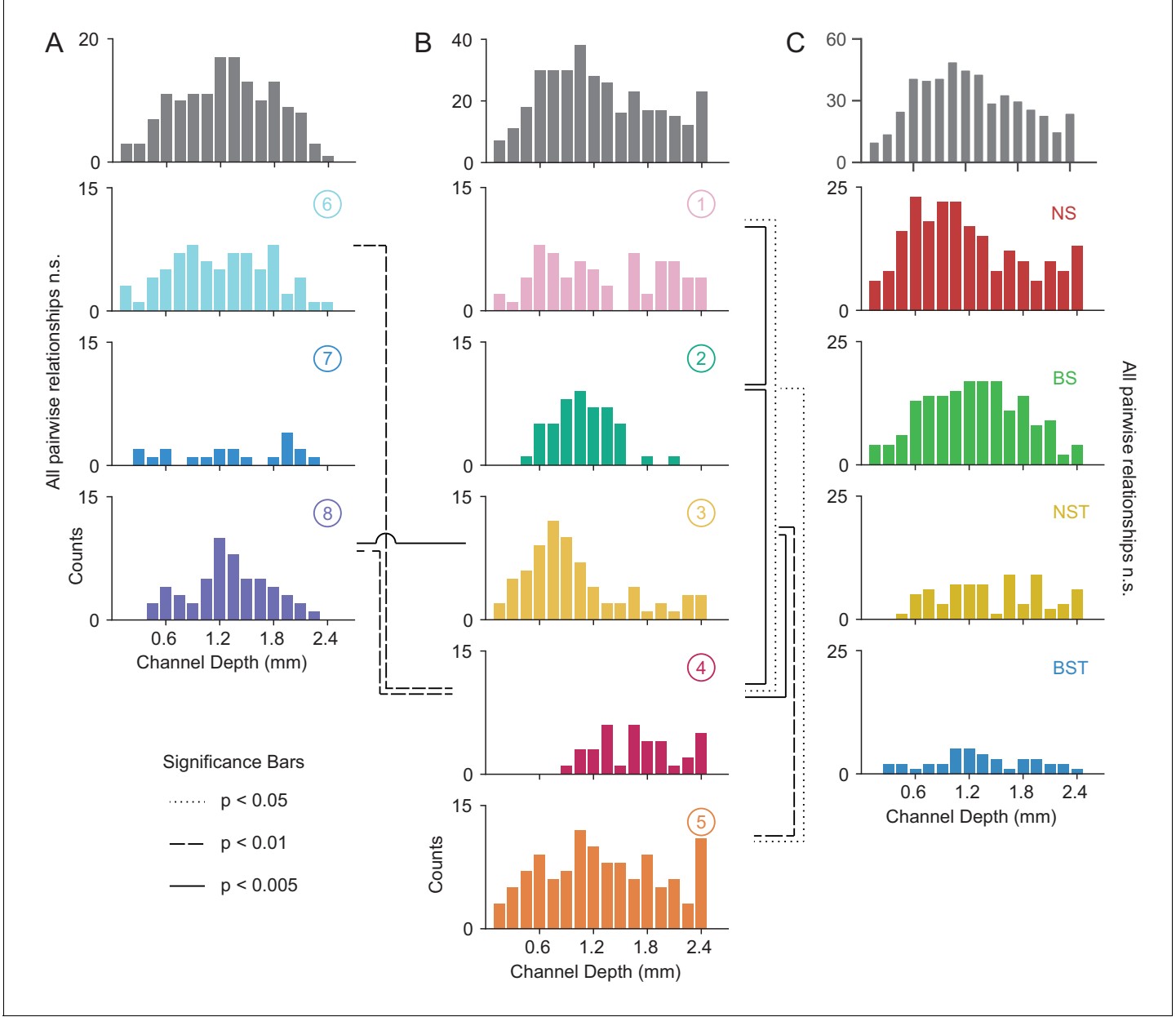

**Figure 8.** Laminar distribution of *WaveMAP* waveform clusters. (**A, B**) The overall histogram for the broad- and narrow-spiking waveform clusters are shown at top across cortical depths on the left and right respectively (in gray); below are shown histograms for their constituent *WaveMAP* clusters. These waveforms are shown sorted by the cortical depth at which they were recorded from the (0.0 mm [presumptive pial surface] to 2.4 mm in 0.15 mm increments). Broad-spiking clusters were generally centered around middle layers and were less distinct in their differences in laminar distribution. Narrow-spiking clusters are shown on the right and were varied in their distribution with almost every cluster significantly varying in laminar distribution from every other. (**C**) Depth histograms for all waveforms collected (top, in gray) and every GMM cluster (below). dotted line p < 0.05; dashed line p < 0.01; solid line p < 0.005; two-sample Kolmogorov-Smirnov Test, FDR adjusted. *Figure 8—figure supplement 1*: Composite figure showing each *WaveMAP* cluster with waveform, physiological, functional, and laminar distribution properties.

The online version of this article includes the following figure supplement(s) for figure 8:

**Figure supplement 1.** Detailed summary of each UMAP cluster and features.

Cluster ② showed a strongly localized concentration of neurons at a depth of 1.1 ± 0.33 mm (mean ± S.D.). It was significantly different from almost all other narrow-spiking clusters (p = 0.002, p = 1e-5, p = 0.010 for ①, ④, and ⑤ respectively; two-sample Kolmogorov-Smirnov test, FDR

adjusted). Similarly, cluster ③ also showed a strongly localized laminar distribution but was situated more superficially than ② with a heavier tail (1.0 ± 0.6 mm, mean ± S.D.).

Cluster ④ was uniquely deep in its cortical distribution (1.70 ± 0.44, mean ± S.D.). These neurons had a strongly triphasic waveform shape characterized by a large pre-hyperpolarization peak. These waveforms have been implicated as arising from myelinated excitatory pyramidal cells (*Barry, 2015*), which are especially dense in this caudal region of PMd (*Barbas and Pandya, 1987*).

The last cluster, ⑤, like ① was characterized by a broad distribution across cortical depths unique among narrow-spiking neurons and was centered around a depth of 1.3 ± 0.65 mm (mean ± S.D.) and present in all layers (*Arikuni et al., 1988*).

Such laminar differences were not observed when we used GMM clustering. Laminar distributions for BS, BST, NS, and NST did not significantly differ from each other (*Figure 8C*; BS vs. BST had p = 0.067, all other relationships p > 0.2; two-sample Kolmogorov-Smirnov test, FDR adjusted). Each GMM cluster also exhibited broad distributions across cortex which is at odds with our understanding of cell types using histology (discussed in the next section).

## Some narrow-spiking *WaveMAP* cluster laminar distributions align with inhibitory subtypes

We have shown that *WaveMAP* clusters have more distinct laminarity than GMM clusters. If *WaveMAP* clusters are consistent with cell type, we should expect their distributions to be relatively consistent with distributions from certain anatomical types visualized via immunohistochemistry (IHC). An especially well-studied set of non-overlapping anatomical inhibitory neuron types in the monkey are parvalbumin-, calretinin-, and calbindin-positive GABAergic interneurons ($PV^+$, $CR^+$, and $CB^+$ respectively) (*DeFelipe, 1997*). Using IHC, we examined tissue from macaque rostral PMd stained for each of these three interneuron types. We then conducted stereological counting of each type averaged across six exemplars to quantify cell type distribution across cortical layers (see *Figure 9A and B*, *Schmitz et al., 2014*) and compared it to the distributions in *Figure 8*.

Both $CB^+$ and $CR^+$ cells (*Figure 9C and D*, respectively) exhibited a similarly restricted superficial distribution most closely resembling ③. In addition, $CR^+$ and $CB^+$ cells are known to have very similar physiological properties and spike shape (*Zaitsev et al., 2005*). An alternative possibility is that one of $CR^+$ or $CB^+$ might correspond to ② and the other to ③ but this is less likely given their nearly identical histological distributions (*Figure 9C and D*) and similar physiology (*Zaitsev et al., 2005*).

In contrast, *WaveMAP* cluster ①, had laminar properties consistent with $PV^+$ neurons (*Figure 9B*): both were concentrated superficially but proliferated into middle layers (*Figure 9E*). In addition, there were striking physiological and functional similarities between ① and $PV^+$ cells. In particular, both ① and $PV^+$ cells have low baseline FR, early responses to stimuli and robust modulation of FR similar to $PV^+$ cells in mouse M1 (*Estebanez et al., 2017*). Cluster ⑤ also had similar properties to ① and could also correspond to $PV^+$ cells.

Together, these results from IHC suggest that the narrow-spiking clusters identified from *WaveMAP* potentially map on to different inhibitory types.

## Heterogeneity in decision-related activity emerges from both cell type and layer

Our final analysis examines whether these *WaveMAP* clusters can explain some of the heterogeneity observed in decision-making responses in PMd over and above previous methods (*Chandrasekaran et al., 2017*). Heterogeneity in decision-related activity can emerge from cortical depth, different cell types within each layer, or both. To quantify the relative contributions of *WaveMAP* clusters and cortical depth, we regressed discrimination time on both separately and together and examined the change in variance explained (adjusted $R^2$). We then compared this against the GMM clusters with cortical depth to show that *WaveMAP* better explains the heterogeneity of decision-related responses.

We previously showed that some of the variability in decision-related responses is explained by the layer from which the neurons are recorded (*Chandrasekaran et al., 2017*). Consistent with previous work, we found that cortical depth explains some variability in discrimination time (1.7%). We next examined if the *WaveMAP* clusters identified also explained variability in discrimination time: a categorical regression between *WaveMAP* clusters and discrimination time, explained a much larger

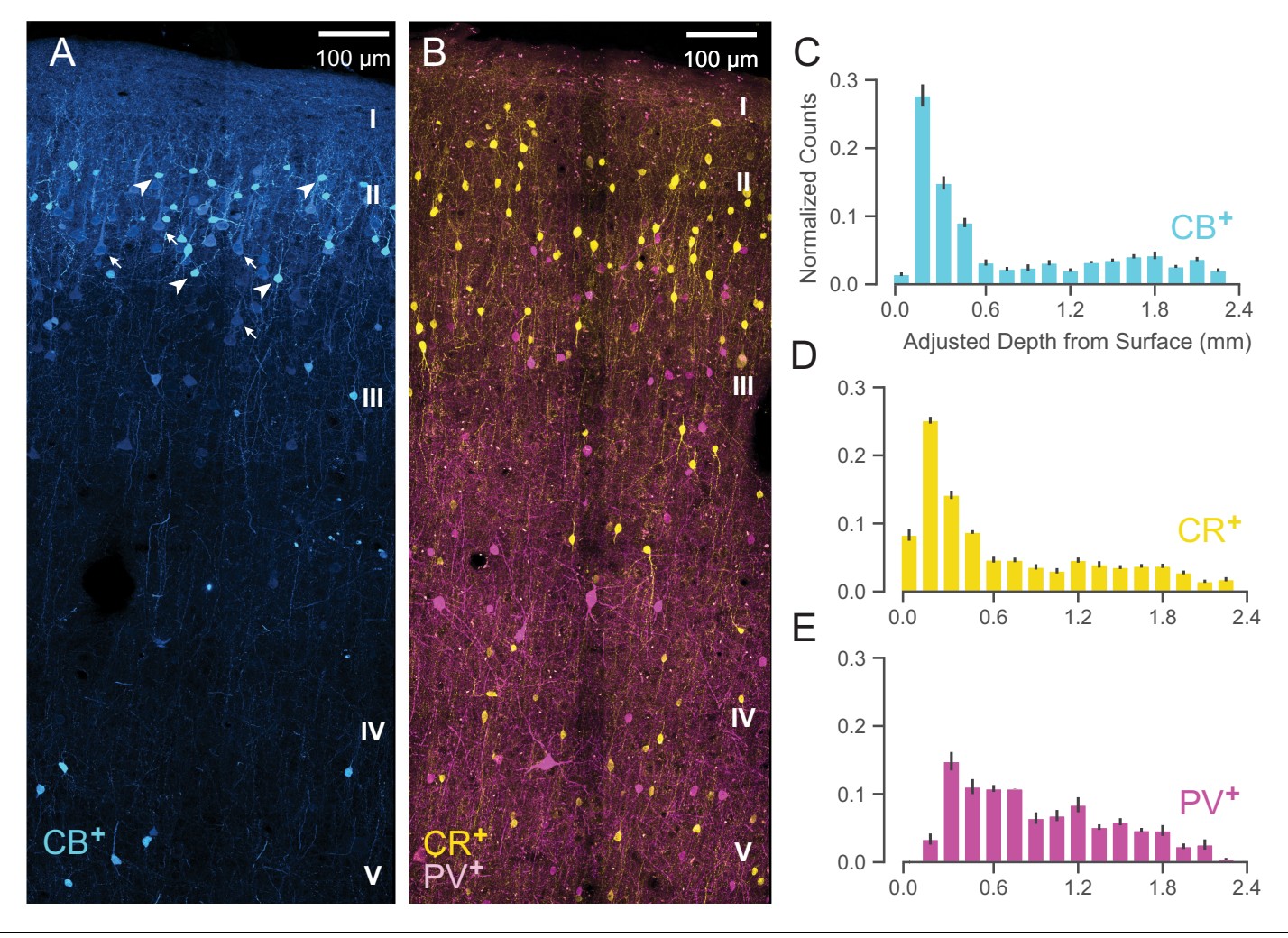

**Figure 9.** Anatomical labeling of three inhibitory interneuron types in PMd. (**A**) Sample maximum intensity projection of immunohistological (IHC) staining of rostral PMd calbindin-positive (CB+) interneurons in blue. Note the many weakly-positive excitatory pyramidal neurons (arrows) in contrast to the strongly-positive interneurons (arrowheads). Only the interneurons were considered in stereological counting. In addition, only around first 1.5 mm of tissue is shown (top of layer V) but the full tissue area was counted down to the 2.4 mm (approximately the top of white matter). Layer IV exists as a thin layer in this area. Layer divisions were estimated based on depth and referencing *Arikuni et al., 1988* (*Arikuni et al., 1988*). (**B**) Sample maximum intensity projection of IHC staining of PMd calretinin-positive (CR+) and parvalbumin-positive (PV+) interneurons in yellow and fuschia respectively. The same depth of tissue and layer delineations were used as in (**A**). (**C, D, E**) Stereological manual counts (*Schmitz et al., 2014*) (mean ± S.D.) of CB+, CR+, PV+ cells in PMd, respectively. Counts were collected from six specimens, each with all three IHC stains, and with counts normalized to each sample. Source files for this figure are available on Dryad (https://doi.org/10.5061/dryad.z612jm6cf).

6.6% of variance. Including both cortical depth and cluster identity in the regression explained 7.3% of variance in discrimination time.

In contrast, we found that GMM clusters regressed against discrimination time only explained 3.3% of variance and the inclusion of both GMM cluster and cortical depth only explained 4.6% of variance.

Thus, we find that *WaveMAP* clustering explains a much larger variance relative to cortical depth alone. This demonstrates that *WaveMAP* clusters come closer to cell types than previous efforts and are not artifacts of layer-dependent decision-related inputs. That is, both the cortical layer in which a cell type is found as well *WaveMAP* cluster membership contributes to the variability in decision-related responses. Furthermore, *WaveMAP* clusters outperform GMM clusters as regressors of a functional property associated with cell types. These results further highlight the power of *WaveMAP* to separate out putative cell types and help us better understand decision-making circuits.

## Discussion

Our goal in this study was to further understand the relationship between waveform shape and the physiology, function , and laminar distribution of cell populations in dorsal premotor cortex during perceptual decision-making. Our approach was to develop a new method, *WaveMAP*, that combines a recently developed non-linear dimensionality reduction technique (UMAP) with graph clustering (Louvain community detection) to uncover hidden diversity in extracellular waveforms. We found this approach not only replicated previous studies by distinguishing between narrow- and broad-spiking neurons, but did so in a way that (1) revealed additional diversity, and (2) obviated the need to examine particular waveform features. In this way, our results demonstrate how traditional feature-based methods obscure biological detail that is more faithfully revealed by our *WaveMAP* method. Furthermore, through interpretable machine learning, we show our approach not only leverages many of the features already established as important in the literature but expands upon them in a more nuanced manner—all with minimal supervision or stipulation of priors. Finally, we show that the candidate cell classes identified by *WaveMAP* have distinct physiological properties, decision-related dynamics, and laminar distribution. The properties of each *WaveMAP* cluster are summarized in *Figure 8—figure supplement 1A and B* for broad- and narrow-spiking clusters, respectively.

*WaveMAP* combines UMAP with high-dimensional graph clustering and interpretable machine learning to better identify candidate cell classes. Our approach might also be useful in other domains that employ non-linear dimensionality reduction such as computational ethology (*Ali et al., 2019*; *Hsu and Yttri, 2020*; *Bala et al., 2020*), analysis of multi-scale population structure (*Diaz-Papkovich et al., 2019*), and metascientific analyses of the literature (*Noichl, 2021*). We also note that while traditional uses of non-linear dimensionality reduction and UMAP has been to data lacking autoregressive properties, such as transcriptomic expression (*Becht et al., 2019*), this does not seem to be an issue for *WaveMAP*. Even though our waveforms have temporal autocorrelation, our method still is able to pick out interesting structure. Other work has found similar success in analyzing time series data with non-linear dimensionality reduction (*Sedaghat-Nejad et al., 2021*; *Dimitriadis et al., 2018*; *Jia et al., 2019*; *Gouwens et al., 2020*; *Ali et al., 2019*).

### Advantages of *WaveMAP* over traditional methods

At the core of *WaveMAP* is UMAP which has some advantages over other non-linear dimensionality reduction methods that have been applied in this context. Although most algorithms offer fast implementations that scale well to large input dimensionalities and volumes of data (*Linderman et al., 2019*; *Nolet et al., 2020*), UMAP also projects efficiently into arbitrary *output* dimensionalities while also returning an invertible transform. That is, we can efficiently project new data into any arbitrary dimensional projected space without having to recompute the mapping.

These properties provide three advantages over other non-linear dimensionality reduction approaches: First, our method is stable in the sense that it produces a consistent number of clusters and each cluster has the same members across random subsamples (*Figure 3—figure supplement 1B*). Clustering in the high-dimensional space rather than the projected space lends stability to our approach. Second, it allows exploration of any region of the projected space no matter the intuited latent dimensionality—this yields an intuitive understanding of how UMAP non-linearly transforms the data, which might be related to underlying biological phenomena. Thus, UMAP allows *WaveMAP* to go beyond a 'discriminative model' typical of other clustering techniques and function as a 'generative model' with which to make predictions. Third, it enables cross-validation of a classifier trained on cluster labels, impossible with methods that don't return an invertible transform. To cross-validate unsupervised methods, unprocessed test data must be passed into a transform computed *only* on training data and evaluated with some loss function (*Moscovich and Rosset, 2019*). This is only possible if an invertible transform is admitted by the method of dimensionality reduction as in UMAP.

A final advantage of UMAP is that it inherently allows for not just unsupervised but supervised and semi-supervised learning whereas some other methods do not (*Sainburg et al., 2020*). This key difference enables 'transductive inference' which is making predictions on unlabeled test points based upon information gleaned from labeled training points. This opens up a diverse number of novel applications in neuroscience through informing the manifold learning process with biological ground truths (in what is called 'metric learning') (*Bellet et al., 2013*; *Yang and Jin, 2006*).

Experimentalists could theoretically pass biological ground truths to *WaveMAP* as training labels and 'teach' *WaveMAP* to produce a manifold that more closely hews to true underlying diversity. For instance, if experimentalists 'opto-tag' neurons of a particular cell type (*Roux et al., 2014*; *Deubner et al., 2019*; *Jia et al., 2019*; *Cohen et al., 2012*; *Hangya et al., 2015*), this information can be passed along with the extracellular waveform to *WaveMAP* which would, in a semi-supervised manner, learn manifolds better aligned to biological truth.

A learned manifold could also be useful in future experiments to identify cell types in real-time without opto-tagging. This could be done by projecting the averaged waveforms found within an experiment into the learned *WaveMAP* manifold. This method would be especially useful in a scenario in which the number of electrodes exceeds the number of channels available to record from simultaneously and not all cell types are of equal interest to record (e.g. Neuropixels probes which have 960 electrodes but simultaneously record from only 384; *Trautmann et al., 2019*; *Jun et al., 2017*). We believe this is a rich area that can be explored in future work.

*WaveMAP* uses a fully unsupervised method for separating and clustering waveform classes associated with distinct laminar distributions and functional properties in a decision-making task. One concern with fully unsupervised methods is that the features used for separation are unclear. However, by applying interpretable machine learning (*Shapley, 1988*; *Lundberg and Lee, 2017*), we showed that our unsupervised methods utilized many of the same waveform features derived by hand in previous work but did so in a single unifying framework. Our interpretable machine learning approach shows how each waveform feature delineates certain waveform clusters at the expense of others and—more importantly—shows how they can be optimally recombined to reveal the full diversity of waveform shapes.

Our novel approach of using non-linear dimensionality reduction with graph clustering on the population of extracellular action potentials compared to specified waveform features has parallels with the evolution of new approaches for the analysis of neuronal firing rates in relevant brain areas (*Shenoy et al., 2013*; *Churchland et al., 2012*; *Mante et al., 2013*; *Remington et al., 2018*; *Wang et al., 2018*). Classically, the approach to analyzing firing rates involved in cognition was to develop simple metrics that separated neurons recorded in relevant brain areas. For instance, tuning is used to separate neurons in the motor (*Georgopoulos et al., 1986*) and visual cortex (*Hubel and Wiesel, 1959*). Similarly, visuomotor indices that categorize neurons along a visual to motor continuum are used to understand firing rates during various tasks in the frontal eye fields (*Bruce and Goldberg, 1985*) and premotor cortex (*Chandrasekaran et al., 2017*). However, these specified features quash other aspects of a firing rate profile in favor of focusing on only a few other aspects. New approaches to analyze firing rates use dimensionality reduction techniques such as principal component analysis (*Shenoy et al., 2013*; *Churchland et al., 2012*; *Cunningham and Yu, 2014*), tensor component analysis (*Williams et al., 2018*), demixed principal component analysis (*Kobak et al., 2016*), targeted dimensionality reduction (*Mante et al., 2013*), and autoencoder neural networks (*Pandarinath et al., 2018*). These methods have provided insight into heterogeneous neural activity patterns in many brain areas without the need for specified features like tuning or a visuomotor index. Our study strongly suggests that non-linear dimensionality reduction methods applied to the entire extracellular waveform are better than using hand-derived waveform features such as trough to peak duration, repolarization time, spike width and other metrics. This progression from user-defined features to data-driven methods follows similar trends in the field of machine learning.

## Waveform cluster shapes are unlikely to arise from electrode placement

It is a possibility that the diversity of waveforms we observe is just an artifact of electrode placement relative to the site of discharge. This supposes that waveform shape changes with respect to the distance between the neuron and the electrode. This is unlikely because both in vitro studies (*Deligkaris et al., 2016*) and computational simulations (*Gold et al., 2006*) show distance from the soma mostly induces changes in amplitude. There is a small widening in waveform width but this occurs at distances in which the amplitude has attenuated below even very low spike thresholds (*Gold et al., 2006*). We controlled for this cell-type-irrelevant variation in amplitude by normalizing spike troughs/peaks during preprocessing to be between $-1$ and $+1$. It should also be noted that without any normalization, all structure was lost in the UMAP projection which instead yielded one

large point cloud (*Figure 3—figure supplement 2E*). Intuitively, this can be understood as UMAP allocating most of the projected space to explaining amplitude differences rather than shape variation. This can be visualized by coloring each point by the log of the amplitude of each spike (log of difference in maximum vs. minimum values) and observing that it forms a smooth gradient in the projected space (*Figure 3—figure supplement 2F*).

It is possible that differences that we observe in waveform shape could be due to recording from different morphological structures (dendrites, soma, or axons) rather than different cell types. However, we believe that most of our waveforms are from the soma. While it is true that there are some cell structures associated with different waveform shapes (such as triphasic waveforms near neurites, especially axons *Barry, 2015*; *Deligkaris et al., 2016*; *Robbins et al., 2013*; *Sun et al., 2021*), highly controlled in vitro studies show that a large majority of EAP's are from somata (86%) (*Deligkaris et al., 2016*). In concordance with these results, we only observed one cluster (④, 6% of all EAP's) with a triphasic shape and these waveforms were only found in deep layers where myelination is prevalent. Thus, we believe that almost all of our waveforms come from somata, with the possible exclusion of ④. Finally, we observed distinct physiological properties (*Figure 6*), decision-related dynamics (*Figure 7*), and laminar distribution (*Figure 8*) for each *WaveMAP* cluster. This would not be the case if the waveforms were just obtained from different compartments of the same neurons.

Given that electrode location has little effect on waveform shape, we might then ask what about a neuron's waveform shape, in terms of cellular physiology, is captured by *WaveMAP*? We propose that the space found by UMAP-1 and UMAP-2 sensibly covaries according to documented properties of $K^+$ ion channel dynamics. As UMAP-1 increases, we observe a smooth transition of the inflection of the repolarization slope from negative to positive (slow to fast repolarization rate; *Figure 5A*). Said differently, the post-hyperpolarization peak becomes sharper as we increase in the UMAP-1 direction. These observations are consistent with the same gradual change in intracellular AP repolarization slope facilitated by the kinetics of the fast voltage-gated Kv3 potassium-channel in an activity-dependent manner (*Kaczmarek and Zhang, 2017*). These channels are necessary for sustained high-frequency firing (*Ding et al., 2011*). In the UMAP-2 direction, there is a smooth decrease in the width of the post-hyperpolarization peak and this direction roughly traverses from broad- to narrow-spiking to triphasic waveforms. This gradual change too has been noted as being associated with the kinetics of the Kv3 potassium-channel: blocking this channel in a dose-dependent manner with tetraethylammonium induces a gradual widening of post-hyperpolarization peak width (*Erisir et al., 1999*; *Bean, 2007*). Both of these changes in intracellular waveform shape likely have a strong effect on the shape of extracellular waveforms (*Henze et al., 2000*).

## Reliance on waveform features might obscure cell type diversity

Our results show a greater proportion of narrow- (putatively inhibitory) vs. broad-spiking (putatively excitatory) neurons (69% vs. 31%, respectively); this is inconsistent with anatomical studies (*Dombrowski et al., 2001*; *Zaitsev et al., 2009*; *Povysheva et al., 2013*). These studies demonstrate, through direct labeling of cell type, that in the macaque cortex, 65–80% of neurons are excitatory while 20–35% are inhibitory. We are not the only study to report this puzzling result: Onorato and colleagues (*Onorato et al., 2020*) also report greater numbers of narrow-spiking compared to broad-spiking neurons in monkey V1. Thus, care must be taken when attempting links between spike waveform shape and cell type (*Lemon et al., 2021*). A resolution to this discrepancy is to rethink equating narrow-spiking to inhibitory cells and broad-spiking to excitatory cells. Anatomical studies show that a substantial number of excitatory neurons in the monkey motor and visual cortices express the Kv3.1b potassium channel which is known to confer neurons with the ability to produce action potentials of narrow spike width and high firing rate (*Constantinople et al., 2009*; *Kelly et al., 2019*; *Kelly and Hawken, 2020*; *Ichinohe et al., 2004*; *Lemon et al., 2021*). Furthermore, researchers have used antidromic stimulation to show that narrow-spiking neurons can be excitatory in motor and premotor cortex (*Vigneswaran et al., 2011*; *Lemon et al., 2021*).

We therefore believe prior studies have underexplored the diversity of classes accessed by their physiological recordings. Evidence of this is that histograms of peak width (and other specified features) across literature are often not cleanly bimodal (*Krimer et al., 2005*; *Zhu et al., 2020*) especially in premotor cortices (*Merchant et al., 2012*). In addition, the relative proportions of narrow vs. broad is often dependent on the cutoff chosen which widely varies across studies

(*Vigneswaran et al., 2011*; *Merchant et al., 2012*). Analyses like ours which look at entire waveforms—rather than a few specified features—extract this diversity from extracellular recordings whereas specified features mix waveform classes.

## Better parcellation of waveform variability leads to biological insight

We find that many narrow-spiking subtypes in PMd signal choice earlier than broad-spiking neurons in our decision-making task (*Figure 7F*). These observations are consistent with another study of PMd in monkeys in reach target selection and movement production (*Song and McPeek, 2010*). In this study, narrow-spiking neurons signaled the selected target 25 ms earlier than broad-spiking neurons. Our results are also consistent with other studies of narrow- vs. broad-spiking neurons in the frontal eye fields (FEF) (*Ding and Gold, 2012*) and inferior temporal area (IT) (*Mruczek and Sheinberg, 2012*) during decision-making. In these studies, narrow-spiking neurons had higher firing rates before movement onset compared to broad-spiking neurons—a result consistent with our observations for some 'narrow-spiking' PMd neurons. Our analyses recapitulate these results and provide additional insights into how different narrow-spiking cell types correlate with decisions. We reproduce the result that narrow-spiking cells, as a whole, have a faster discrimination time than broad-spiking cells but in addition we show that certain narrow-spiking cells respond as slowly as broad-spiking cells (② and ④; *Figure 7F*). This lends further evidence to our theory that ② and ④ are likely narrow-spiking excitatory cells. In contrast, ③ and ① while both narrow-spiking, had distributions that more aligned with histologically-verified inhibitory types. In addition, ③ and ① had physiological properties more in line with inhibitory cell types.

*WaveMAP* suggests that narrow-spiking waveforms encompass many cell classes with distinct shape and laminar distribution. One of our narrow-spiking clusters (cluster ②) was restricted to more superficial layers (*Figure 8B*) and had certain functional properties—low baseline firing rate and longer discrimination times—which are thought to be more closely aligned to properties of excitatory neurons (*Song and McPeek, 2010*). Another narrow-spiking cluster, ④, exhibited physiological and functional properties similar to ② (all comparisons not significant in *Figure 6C,D and E* or *Figure 7E and F*) but with a distinct laminar distribution (*Figure 8B*) and highly triphasic waveform shape (*Figure 8—figure supplement 1B*). In contrast to ②, which was concentrated in layer III, ④ was restricted to deep layers. These tri-phasic neurons could either be large corticospinal excitatory pyramidal cells (*Ichinohe et al., 2004*; *Soares et al., 2017*; *Vigneswaran et al., 2011*), or axons (*Barry, 2015*; *Robbins et al., 2013*; *Sun et al., 2021*).

## High-density probes and optogenetics can provide better insight into cell classes in the primate

Our recordings here were performed with 16 channel U-probes which provided reasonable estimates of laminar organization for these different putative cell classes. Use of high-density electrophysiological methods providing higher electrode counts perpendicular to the cortical surface would provide further insight into the laminar organization of different cell types (*Jun et al., 2017*; *Dimitriadis et al., 2020*). High-density recordings would allow us to perform *WaveMAP* in an additional dimension (across multiple electrodes) to increase confidence in identified cell classes (*Mosher et al., 2020*) and localization of signal to somata (*Jia et al., 2019*). Sensitive electrodes providing spatial access to neural activity (*Jun et al., 2017*) can also improve our understanding of how these cell classes are organized both parallel and perpendicular to cortical surface (*Saleh et al., 2019*; *Mosher et al., 2020*) and across areas (*Dimitriadis et al., 2020*). Access to cell types with high-density recordings would also allow for the identification of 'me-types' through electromorphology (*Gouwens et al., 2020*; *Tasic et al., 2018*). This information could also help inform detailed models of cortical circuits that incorporate cell type information (*Gouwens et al., 2018*; *Billeh et al., 2020*; *Reimann et al., 2013*).

Another powerful tool that has been leveraged in the study of cell types during behavior is optogenetics (*Pinto and Dan, 2015*; *Lui et al., 2021*; *Kvitsiani et al., 2013*). Although in its infancy relative to its use in the mouse, optogenetics in monkeys offers direct interrogation of cell types. Future studies will allow us to more precisely link putative cell classes in vivo to function (*Courtin et al., 2014*). NHP optogenetics is slowly advancing and efforts in many research groups around the world are producing new methods for in vivo optogenetics (*Tremblay et al., 2020*). We expect future

experiments using the promising new mDlx (*De et al., 2020*) and h56d (*Mehta et al., 2019*) promoter sequences to selectively opto-tag inhibitory neurons or PV⁺ neurons directly (*Vormstein-Schneider et al., 2020*) will greatly benefit validation of these derived cell classes. Finally, *WaveMAP*'s ability to find clusters of putative biological relevance using waveform shape alone encourages its application in settings where ground truth evaluation is particularly difficult to obtain such as in the human brain (*Paulk et al., 2021*).

# Materials and methods

## Key resources table

| Reagent type (species) or resource | Designation | Source or reference | Identifiers | Additional information |
|---|---|---|---|---|
| Primary antibody | Rabbit anti-calbindin D-28k (polyclonal) | Swant | Cat#: CB38 RRID:AB_10000340 | 1:2000 dilution |
| Primary antibody | Rabbit anti-calretinin D-28k (polyclonal) | Swant | Cat#: 7697 RRID:AB_2619710 | 1:2000 dilution |
| Primary antibody | Guinea pig anti-parvalbumin (polyclonal) | Swant | Cat#: GP72 RRID:AB_2665495 | 1:2000 dilution |
| Secondary antibody | Donkey anti-rabbit Alexa 546 | ThermoFisher | Cat#: A10040 RRID:AB_2534016 | 1:200 dilution |
| Secondary antibody | Donkey anti-guinea pig Alexa 546 | Jackson | Cat#: 706-545-148 RRID:AB_2340472 | 1:200 dilution |

## Code and data availability

All figures and figure supplements can be generated from the code and data included with the manuscript and uploaded to Dryad/Zenodo (RRID:SCR_005910/RRID:SCR_004129) (https://doi.org/10.5061/dryad.z612jm6cf; *Lee et al., 2021*) and on Github (https://github.com/EricKenjiLee/WaveMAP_Paper). Pre-processing of raw averaged data was conducted in MATLAB (RRID:SCR_001622) using the files located in Preprocessing.zip (see contained README.md). *Figure 1* was generated using MATLAB whereas all other figures were generated in Python (RRID:SCR_008394) using the Jupyter/Google CoLab (RRID:SCR_018315/RRID:SCR_018009) notebook available with this manuscript. Please see the Readme.md file included in the zip file WaveMAP_Paper.zip for instructions on how to generate all manuscript figures and supplementary figures. Raw confocal fluorescence images with associated CellCounter annotations are also available (*Lee et al., 2021*). Further information about *WaveMAP* and updated notebooks can also be obtained from the Chandrasekaran lab website at Boston University (http://www.chandlab.org).

## Subjects and surgery

Our experiments were conducted using two adult male macaque monkeys (*Macaca mulatta*; monkey T, 7 years, 14 kg; O, 11 years, 15.5 kg) that were trained to reach to visual targets for a juice reward. Our monkeys were housed in a social vivarium with a normal day/night cycle. This study was performed in strict accordance with the recommendations in the Guide for the Care and Use of Laboratory Animals of the National Institutes of Health. All the procedures were approved by the Stanford Administrative Panel on Laboratory Animal Care (APLAC, Protocol Number 8856 entitled 'Cortical Processing of Arm Movements'). Surgical procedures were performed under anesthesia, and every effort was made to minimize suffering. Appropriate analgesia, pain relief, and antibiotics were administered to the animals when needed after surgical approval.

After initial training to come out of the cage and sit comfortably in a chair, monkeys underwent sterile surgery for implantation of head restraint holders (Crist Instruments, cylindrical head holder) and standard recording cylinders (Crist Instruments, Hagerstown, MD). We placed our cylinders over caudal PMd (+16, 15 stereotaxic coordinates) and surface normal to the cortex. We covered the skull within the cylinder with a thin layer of dental acrylic/PALACOS bone cement.

## Apparatus

Monkeys sat in a customized chair (Crist Instruments, Snyder Chair) with their head restrained via the surgical implant. The arm not used for reaching was loosely restrained using a tube and a cloth sling. Experiments were controlled and data were collected under a custom computer control system (xPC target and Psychtoolbox-3 [RRID:SCR_002881] *Kleiner et al., 2007*). Visual stimuli were displayed on an Acer HN2741 computer screen placed approximately 30 cm from the monkey and a photodetector (Thorlabs PD360A) was used to record the onset of the visual stimulus at a 1 ms resolution. Every session, we taped a small infrared reflective bead (11.5 mm, NDI Digital passive spheres) 1 cm from the tip of the middle digit of the right hand (left hand, monkey O). The position of this bead was tracked optically in the infrared (60 Hz, 0.35 mm root mean square accuracy; Polaris system; Northern Digital).

Eye position was tracked with an overhead infrared camera made by ISCAN along with associated software (estimated accuracy of 1°, ISCAN, Burlington, MA). To get a stable eye image for the overhead infrared camera, an infrared dichroic mirror was positioned at a 45° angle (facing upward) immediately in front of the nose. This mirror reflected the image of the eye in the infrared range while letting visible light pass through. A visor placed around the chair prevented the monkey from touching the infrared mirror, the juice tube, or bringing the bead to their mouth.

## Behavioral training

Our animals were trained using the following operant conditioning protocol. First, the animal was rewarded for arm movements toward the screen and learnt to take pieces of fruit on the screen. Once the animal acquired the association between reaching and reward, the animal was conditioned to reach and touch a target for a juice reward. The position, as well as the color of this target, was then randomized as the monkey learned to touch targets of various colors at different locations on the screen. We then used a design in which the monkey first held the central hold for a brief period, and then a checkerboard cue, which was nearly 100% red or 100% green, appeared for 400–600 ms and finally the two targets appeared. The monkey received a reward for making a reach to the color of the target that matched the predominant color of the checkerboard cue. Two-target 'Decision' blocks were interleaved with single target blocks to reinforce the association between checkerboard color and the correct target. After two weeks of training with this interleaved paradigm, the animal reliably reached to the target matching the color of the central checkerboard cue. We switched the paradigm around by adopting a design in which the targets appeared before the checkerboard cue onset. We initially trained on holding periods (where the monkeys view targets) from 300 to 1800 ms. We trained the animal to maintain the hold on the center until the checkerboard cue appeared by providing small amounts of juice at rough time intervals. When the animal reliably avoided breaking central hold during the hold period, we stopped providing the small amounts of juice for holding but maintained the juice reward for correct reaches. After the animal learned to stay still during the target viewing period, we introduced more difficult checkerboard cues (decreased color coherences) to the animal while reducing the maximal holding period to 900 ms. We then trained the animal to discriminate the checkerboard as accurately and as fast as possible while discouraging impulsivity by adopting timeouts.

## Electrophysiological recordings

To guide the stereotaxic coordinates for our eletrophysiological recordings we used known response-to-muscle palpation properties of PMd and M1. Our chambers were placed normal to the surface of cortex and aligned with the skull of the monkey. Recordings were performed perpendicular to the surface of the brain. Recordings were made anterior to the central sulcus, lateral to the spur of the arcuate sulcus, and lateral to the precentral dimple. For both monkeys, we were able to identify the upper and lower arm representation by repeated palpation at a large number of sites to identify muscle groups associated with the sites. Recordings were performed in the PMd and M1 contralateral to the arm used by the monkey. Monkey T used his right arm (O used his left arm) to perform tasks.

A subset of the electrophysiological recordings were performed using traditional single electrode recording techniques. Briefly, we made small burr holes through the PALACOS/acrylic using handheld drills. We then used a Narishige drive with a blunt guide tube placed in firm contact with the

dura. Recordings were obtained using FHC electrodes to penetrate the overlying dura (UEWLGC-SEEN1E, 110 mm long and 250 µm thick electrodes with a standard blunt tip and profile, epoxylite insulation, and an impedance of 5–7 MΩ) . Every effort was made to isolate single units during the recordings with FHC electrodes by online monitoring and seeking out well-isolated signals (see next section below).

We performed linear multi-contact electrode (U-probe) recordings in the same manner as single electrode recordings with some minor modifications. We used 180 µm thick 16-electrode U-probes (15 µm Pt/Ir electrode site diameter, 150 µm spacing, circular shape, polyimide insulation, and secured in medical-grade epoxy. Electrode contacts were ~100 KΩ in impedance). We used a slightly sharpened guide tube to provide more purchase on dura. We also periodically scraped away, under ketamine-dexmetotomidine anesthesia, any overlying tissue on the dura. Both these modifications greatly facilitated penetration of the U-probe. We typically penetrated the brain at very slow rates (~2–5 µm/s). Once we felt we had a reasonable sample population of neurons, potentially spanning different cortical layers, we stopped and waited for 45–60 min for the neuronal responses to stabilize. The experiments then progressed as usual.

We attempted to minimize the variability in U-probe placement on a session-by-session basis. Our approach was to place the U-probe so that the most superficial electrodes (electrodes 1, 2 on the 16 channel probe) were in layer I and able to record multi-unit spiking activity. Any further movement of the electrode upwards resulted in the disappearance of spiking activity and a change in the overall activity pattern of the electrode (suppression of overall LFP amplitudes). Similarly, driving the electrodes deeper resulted in multiphasic extracellular waveforms and also a change in auditory markers which were characterized by decreases in overall signal intensity and frequency content. Both markers suggested that the electrode entered white matter. Recording yields and electrode placement were in general much better in monkey T (average of ~16 units per session) than monkey O (average of ~nine units per session). We utilized these physiological markers as a guide to place electrodes and thus minimize variability in electrode placement on a session-by-session basis. Importantly, the variability in placement would act against our findings of depth-related differences shown in *Figure 8*.

## Identification of single neurons during recordings

Our procedure for identifying well-isolated single neurons was as follows: In the case of the single FHC tungsten electrode recordings, we moved the electrode and conservatively adjusted the threshold until we identified a well-demarcated set of waveforms. We took extra care to separate these waveforms from the noise and other smaller neurons. Our ability to isolate neurons was helped by the fact that these electrodes have a very small exposed area (hence high impedance) allowing for excellent isolation. Once a stable set of waveforms was identified, hoops from the Central software (Blackrock Microsystems) were used to demarcate the waveforms from noise and other potential single neurons. The electrode was allowed to settle for at least 15 min to ensure that the recording was stable. Once stability was confirmed, we began data collection. If we found that the recording was unstable, we discarded the neuron and moved the electrode to a newly isolated neuron and repeated the procedure. For a stable recording, we stored the waveform snippet and the time of the spike. Offline, we first visualized the waveforms in MATLAB by performing PCA. If we found that our online identification of the waveforms was inadequate, we either discarded the recording, identified it as a multi-unit (not used in this paper), or exported the data to Plexon Offline Sorter and redrew the cluster boundaries. We also took advantage of Plexon Offline Sorter's ability to visualize how PCA features changed with time to ensure the quality and stability of our isolation. Finally, after redrawing cluster boundaries, we exported the data back to our analysis pipeline.

For our 16-channel Plexon U-Probe recordings, we again lowered the electrode until we found a stable set of waveforms. The small contact area of these electrode sites again ensured excellent identification and low levels of background noise (~10–20 µV). We then waited at least 45 min until the recordings were very stable. Such an approach ensured that we minimized electrode drift. In our experience, the U-probes also have less drift than Neuropixel or other recording methods. We then again repeated the conservative thresholding and identification procedure outlined for the FHC electrodes. For U-probes, we did not move the electrodes once they had settled. Instead, we constantly monitored the recordings and any changes in a particular electrode over time led to the units

from that electrode being discarded and not included in further analysis. Finally, the same offline procedures used for FHC electrodes were repeated for the U-probe recordings.

## Preprocessing of single-unit recordings

We obtained 996 extracellularly recorded single units (778 units recorded with the U-probe) from PMd across two monkeys (450 from Monkey O and 546 from Monkey T). Of these, we identified 801 units whose ISI violations (refractory period $\leq$ 1.5 ms) $\leq$ 1.5% (*Chandrasekaran et al., 2017*). Our waveforms were filtered with a 4th-order 250 Hz high-pass Butterworth filter. The waveforms for each of the units were extracted for a duration of 1.6 ms with a pre-trough period of 0.4 ms, sampled at 30 kHz.

## Alignment and normalization of waveforms

In order to calculate the mean waveform for each single unit, we upsampled individual waveforms calculated over different trials by a factor of 10 and aligned them based on the method proposed in *Kaufman et al., 2013*. For each waveform, we calculated its upswing slope (slope between trough to peak) and the downswing slope (slope to the trough) and re-aligned to the midpoint of the slope that exceeded the other by a factor of 1.5. Following this alignment, we chose the best set of waveforms for calculating the mean as those that satisfied the criteria (1) less the two standard deviations (S.D.) from the mean at each point and (2) average deviation from the mean across time was less than 0.4 (*Kaufman et al., 2013*). The final set of waveforms for each unit was averaged and downsampled to 48 time points. Upon visual inspection, we then identified 761 units (625 single units with 490 U-probe recorded units) whose average waveforms qualified the criteria of exhibiting a minimum of two phases with trough occurring first. The remaining waveforms, unless stated otherwise here, were removed from the analysis. We excluded positive-spiking waveforms because of their association with axons (*Sun et al., 2021*). Finally, we normalized the waveforms by dividing the extreme value of the amplitude such that the maximum deviation is ±1 unit (*Snyder et al., 2016*).

It is important to note that the preprocessing we use, individual mean subtraction and ±1 unit normalization, operates independently of the data. Using another commonly used preprocessing normalization, normalization to trough depth (*Kaufman et al., 2010*), we obtained extremely similar results. We found ±1 unit trough to peak normalization had virtually the same number of clusters as normalization to trough ($8.29 \pm 0.84$ vs. $8.16 \pm 0.65$ clusters, mean ± S.D.; *Figure 3—figure supplement 2A and C*). Furthermore, both normalizations picked out the same structure (*Figure 3—figure supplement 2B and D*); the normalization to trough did have a 9th cluster splitting off of ⑤ but this was something also seen with ±one unit trough to peak normalization in certain data subsets as well.

## A step-by-step guide to UMAP and Louvain clustering in *WaveMAP*

To provide the reader with an intuitive overview of *WaveMAP*, we provide a step-by-step exposition of the different stages in the workflow shown in *Figure 2* beginning with UMAP followed by Louvain community detection. UMAP is a non-linear method that enables the capture of latent structures in high-dimensional data as a graph. This graph can then be used to visualize the latent structure in a low-dimensional embedding (*McInnes et al., 2018*). For a detailed description of the methods, please refer to the Supplemental Information or respective references for UMAP (*McInnes et al., 2018*) and Louvain community detection (*Blondel et al., 2008*).

### Figure 2A–i

We pass 625 normalized single-unit waveforms into UMAP. This normalization is crucial for exposing interesting structure in downstream analysis, although the particular normalization is less important (*Figure 3—figure supplement 2*). UMAP uses a five-step (ii.a to ii.e in *Figure 2A*) procedure to construct a weighted high-dimensional graph.

### Figure 2A–ii.a

In the first step, the data for each waveform is viewed in its original (sometimes called 'ambient') 48-dimensional space with each dimension corresponding to one of 48 time points along the waveform recording.

### Figure 2A–ii.b

A local metric is then assigned to each data point such that a unit ball (distance of one) surrounding it extends to the 1st-nearest neighbor. This ensures that every point is connected to at least one other point.

### Figure 2A–ii.c

Beyond this first connection, the distances to the next $(k-1)$-nearest neighbors increases according to an exponential distribution scaled by the local density. This is shown as a 'glow' around each of the unit balls in *Figure 2A–ii*.c.

### Figure 2A–ii.d

The distances from the local point to the $k-1$ data points beyond the unit ball are made to be probabilistic ('fuzzy') according to their distance (k = four in *Figure 2A–ii*.d with some low weight connections omitted for clarity). This also means that the metric around each data point has a different notion of 'how far' their neighbors are. Distances are shorter in dense regions (with respect to the ambient space) than are distances in sparser regions leading to a graph with asymmetric edge weights. If the notion of a probabilistic connection is confusing, this construction can just be understood as an asymmetric directed graph with edge weights between zero and one. One way to understand this is through the following real life example: to someone living in a dense city, traveling several miles may seem very far, while for a rural resident, this distance might be trivial even if the absolute distance is the same.

### Figure 2A–ii.e

The edge weights between any two data points, $a$ and $b$, are 'averaged together' according to the formula $a + b - a \cdot b$ known as probabilistic sum. This results in a graph that now has symmetric edge weights.

### Figure 2Biv

However, before we project this graph into lower dimension, we first apply clustering to the high-dimensional graph with a method known as Louvain community detection. This method proceeds in two steps per pass: modularity optimization and community aggregation (*Figure 2—figure supplement 1B*).

### Figure 2B-iv.a:

In the first step of Louvain, each node in the graph is assigned to its own 'community' which can be interpreted as its own cluster. Next, each node will join a neighboring node's community such as to maximize an objective function known as 'modularity score' (see Supplemental Information for the exact equation). This score is maximized when the sum of the weighted edges within a community is maximal relative to the sum of the weighted edges incident on the community from nodes outside the community. This procedure operates on all nodes until modularity can no longer be increased across the network; this concludes the modularity optimization step. In the next step, community aggregation, all nodes in a community are collapsed into a single node to create a new network. This completes the first pass of Louvain which then repeats modularity optimization and aggregation on this new graph until modularity is once again maximized. This continues until modularity no longer increases across hierarchies of graphs. Note that the resolution parameter is set to one in the work of *Blondel et al., 2008* but we use an implementation of Louvain that allows for changing of this parameter according to the definition of modularity given in *Lambiotte, 2007*. The resolution parameter gives the user some ability to specify how large of a community they expect as might be related to a phenomenon of interest and should be chosen empirically.

### Figure 2B-iv.b

The final Louvain community memberships are propagated back to the original nodes and assigned as labels for the associated data points which completes the classification step.

### Figure 2B–v

In the second step of UMAP, graph layout, the high-dimensional graph with symmetric edge weights from the previous step is projected down into some lower dimension (here it is two dimensions).

To initialize this process, the graph is first passed through a Laplacian eigenmap (*Belkin and Niyogi, 2002*) which helps regularize the initial embedding of the graph in low dimension (*Kobak and Linderman, 2019*).

### Figure 2B–v.a

We know the graph in high dimension but we have not yet found this graph in low dimension. From here a force directed graph layout procedure is used to align the initialized low-dimensional graph with the one found in high dimension. The force directed graph layout procedure minimizes the difference in distances in the instantiated graph compared to the high-dimensional one by alternatingly applying an attractive and a repulsive force according to the edge weights. These forces are chosen to minimize the cross-entropy between the graphs. This process ensures that points close in high dimension but far in low dimension are brought together (attraction) and those that are far in high dimension but close in low dimension are pushed apart (repulsion).

### Figure 2B–v.b

A final embedding of the data is found using stochastic gradient descent but by fixing the seed in our procedure, we enforce standard gradient descent. Although this requires more memory and is less performant, it guarantees that embeddings will look the same (even if this doesn't affect clustering).

### Figure 2B–vi

In the final step of our method, we combine the labels found through Louvain clustering with a low-dimensional embedding to arrive at our *WaveMAP* solution.

## WaveMAP parameter selection and validation

The normalized extracellular waveforms were passed into the Python package umap 0.4.0rc3 (RRID: SCR_018217) (*McInnes et al., 2018*) with the parameters shown in *Table 1*. The n_neighbors value was increased to 20 to induce more emphasis on global structure. UMAP utilizes a stochastic k-nearest neighbor search to establish the graph and stochastic gradient descent to arrive at the embedding thus it produces similar but different embeddings in the projected space. For reproducibility reasons, the random_state was fixed in the algorithm and in numpy. The choice of random seed only impacted the projection and not the clustering (*Figure 3—figure supplement 1A*). From here, the graph provided by umap.graph_ was passed into the Louvain community detection algorithm to generate the clustering seen in *Figure 3A*. For details of the UMAP algorithm, see Supplementary Information.

Graph networks are often hierarchical and it has been recommended that the Louvain resolution parameter be chosen to elicit the phenomenon of interest (*Porter et al., 2009*; *Lambiotte et al., 2008*). To select the resolution parameter $t$, we chose a value that best maximized modularity score (a measure of the ratio between connections within a cluster vs. incoming from outside of it; see Supplementary Information) while still returning an statistically analyzable number of clusters (n > 20). We selected a resolution parameter (green marker on *Figure 3B*) that maximized modularity score of Louvain clustering while still returning clusters of $n>20$ to allow for downstream statistical analyses. We note that this was also very close to the 'elbow' in terms of number of clusters; this verifies that we have reached near-optimality in a second sense of obtaining stable cluster number. These scores were calculated over 25 random UMAP instantiations of 80% of the full dataset in 5% intervals. For algorithmic details of Louvain clustering, see Supplementary Information.

To validate that our parameter selection was stable and produced the same number of clusters reliably, we used a bootstrap and applied the *WaveMAP* procedure to random subsets of the full dataset (*Figure 3—figure supplement 1B*, *Tibshirani and Walther, 2005*). We obtained 100 random samples from 10% to 90% of the full data set in 10% increments while simultaneously choosing a different random seed for the UMAP algorithm each time. We calculated both the number of Louvain clusters and the adjusted mutual information score (AMI) across these random samples and plot

it on the same graph *Figure 3—figure supplement 1B*, red and green. The AMI is a measure of how much 'information' is shared between a pair of clusterings with information specifically as Shannon entropy. The Shannon entropy of a given clustering (often called a 'partitioning'), $H(X)$, is defined as,

$$H(X) = -\sum_{i=1}^{n} P(x_i) \log P(x_i)$$

with the probability $P(x_i)$ pertaining to the probability that a certain data point $x_i$ will belong to a certain cluster. The clustering entropy can be understood as how 'unpredictable' the results of a random variable are (in this case the random variable is the particular clustering solution). For instance, a fair coin is less predictable (greater entropy) than a loaded coin (lower entropy).

Intuitively, AMI is how much information we receive about one variable given an observation of another and vice versa (*Timme and Lapish, 2018*) (or how much knowing one clustering allows us to predict another) corrected for random chance. Thus, it's bounded between 0 and 1 with 0 for two completely independent variables and one for completely identical variables. Formally, un-adjusted mutual information, $I(X,Y)$, is defined as,

$$I(X,Y) = H(X) - H(X|Y) = D_{KL}(P_{X,Y}(x,y) \parallel P_X(x) \cdot P_Y(y))$$

where $H(X|Y)$ is the conditional Shannon entropy, $P_{X,Y}(x,y)$ is the joint distribution of $X$ and $Y$, and $P_X(x)$ is a marginal distribution.

The value at 100% is omitted because it has the same cluster number as our dataset and zero variance since *WaveMAP* is invariant to random seed selection. Thus the variation in cluster number due to sampling and random seed are compounded and shown together in *Figure 3—figure supplement 1B*.

Ensemble clustering for graphs (ECG) (*Poulin and Théberge, 2018*; *Poulin and Théberge, 2019*) was used to validate the clusters found in *Figure 3A* (see *Figure 3—figure supplement 1C*). We added the algorithm (https://github.com/ftheberge/Ensemble-Clustering-for-Graphs; *Théberge, 2020*) into the python-igraph package (*Csardi and Nepusz, 2006*) and passed UMAP graphs into it directly. We set the number of partitions $k$ to be 10 to produce the plot in *Figure 3—figure supplement 1C*. This algorithm uses $k$ different randomized instantiations of the clusters in the graph followed by one round of Louvain clustering (*Figure 2—figure supplement 1B*). Each of these $k$ level-1 graphs (called level-1 partitions since one round of Louvain was performed) are then combined as a single graph such that when edges co-occur between nodes in one of the $k$ graphs, it is more heavily weighted. This ensembling of several graphs via the weight function $W_{\mathcal{P}}$ (see Supplemental Materials and methods section *Ensemble clustering for graphs (ECG)*) yields the final ECG graph.

## Gradient boosted decision tree classifier

We then trained a gradient boosted decision tree classifier in xgboost 1.0.2 (RRID:SCR_021361) (*Chen and Guestrin, 2016*). A 30–70% test-train split was used with the test set never seen by model training or hyperparameter tuning. A 5-fold cross-validation was applied to the training data and optimal hyperparameters were obtained after a grid search on the folds using scikit-learn's (RRID:SCR_002577) GridSearchCV function with final parameters in *Table 1*. The default multi-class objective function multi:softmax was also used. The percent accuracy for each cluster against all others is plotted as a confusion matrix in *Figure 3C* by applying the final classifier model to the unseen test set.

The same procedure was used when training on the GMM labels found in *Figure 4D* and for the eight cluster GMM labels in *Figure 4—figure supplement 2B*. Each of these classifiers also separately underwent hyperparameter tuning using scikit-learn's GridSearchCV function as well with final hyperparameters shown in 2.

It is important to note that cross-validation was done after the cluster labels were generated by looking at the entire dataset (both via the algorithm itself and our tuning of parameters). This results in data leakage (*Moscovich and Rosset, 2019*; *Kaufman et al., 2011*) which potentially hurts out-of-dataset performance. Thus, classifier performance is only used here to demonstrate UMAP's ability to sensibly separate waveforms within-dataset relative to traditional GMM methods (*Figure 4D*).

*Figure 3C* is not leveraged to provide firm insight into how such a classifier would perform out-of-dataset. It is also important to note that none of the parameters for *WaveMAP* (n_neighbors or resolution) were tuned to optimize for classifier performance and thus the direction of bias is not necessarily deleterious.

## Specified waveform shape features

To compute specified features for each normalized waveforms (*Figure 4A*), we first up-sampled the waveforms from 48 to 480 time points using a cubic spline interpolation method. We then used this up-sampled waveform to compute three separate features: trough to peak duration, AP width, and peak ratio. Trough to peak is the time from the bottom of the depolarization trough (global minimum) to the post-hyperpolarization peak (subsequent local maximum). AP width was calculated as the width of the depolarization trough at the full-width half-minimum point. Both these measures were found using the mwave function from the MLIB 1.7.0.0 toolbox (*Stuttgen, 2019*). Peak ratio was the ratio of heights (above baseline) between the pre-hyperpolarization (maximum before trough) and the post-hyperpolarization peak (maximum after trough).

## Gaussian mixture model clustering

Using the specified feature values (trough to peak, AP width, and peak ratio), the normalized waveforms were clustered in the three-dimensional feature space using a Gaussian mixture model (GMM) with hard-assignment (each data point belongs to one cluster) through MATLAB's fitgmdist function across 50 replicates (*Figure 4B*). Each replicate is a different random instantiation of the GMM algorithm and the model with the largest log likelihood is chosen.

The Bayesian information criterion (BIC) was used to determine the optimal cluster number and is defined as

$$\mathrm{BIC} = -2\ln P(X|\theta) + K\ln(n) \tag{1}$$

where the first term $-2\ln P(X|\theta)$ is a 'goodness of fit' term obtained from the negative log likelihood function, that is, the conditional probability of observing the sample $X$ given a vector of parameters $\theta$. In the particular case of GMM, the function $P(X|\theta)$ is the probability of observing the clusters given an underlying particular sum of multivariate Gaussians (the likelihood). The second term $K\ln(n)$ is a penalty on the number of parameters $n$ which approximates model complexity. Penalizing the number of model parameters (number of clusters $K$) scaled by the number of data points $n$ captures the idea that 'simplicity is better'. This criterion ultimately constrains the number of Gaussians used to fit the data.

Assuming we have $N_f$ features and $N_c$ clusters we can calculate $K$ using the following framework: For each Gaussian mixture model, the total number of parameters is $N_f$ means and $N_f(N_f + 1)/2$ covariance parameters. Another free parameter that is learned is the weight for each Gaussian that sums up to 1, leaving us with $N_c - 1$ unique weights. Thus the $K$ which is the effective number of parameters for a GMM is,

$$K = N_c(N_f + \frac{N_f(N_f + 1)}{2}) + N_c - 1 \tag{2}$$

The 'best' model in a BIC-sense will have the set of parameters $\theta$ maximizing the likelihood function (thus minimizing the negative log likelihood) for a given model or model family—a number of multivariate Gaussians in a three-dimensional feature space in this case. To arrive at the parameters best approximating the Gaussian distribution giving rise to the data (Maximum Likelihood Estimation or MLE), the Expectation-Maximization (EM) algorithm was used. The optimal cluster number was selected as the lowest number of clusters between 1 and 10 at which the change in BIC was minimized (at the 'elbow' in *Figure 4C*).

## Interpretable machine learning: UMAP inverse transform and SHAP

To facilitate interpretability, we used the invertibility of the UMAP transform (which itself is based on Delauney triangulation) to generate test waveforms tiling the projected space *Figure 5A*. 100 evenly-spaced test coordinates were generated spanning a portion of the embedded space and passed backwards through the UMAP transform using umap's built-in inverse_transform function.

The waveform generated at each test point is shown color-coded to the nearest cluster color or in gray if the distance exceeds 0.5 units in UMAP space.

Using the package shap (RRID:SCR_021362; *Lundberg et al., 2020*), SHAP values were calculated for the classifier trained on *WaveMAP* identified clusters. The trained XGBoost model was passed directly into the tree model-specific shap.TreeExplainer (*Lundberg et al., 2018*) which then calculated the mean absolute SHAP values (the average impact on model performance, postive or negative) for all waveform time points (features). TreeExplainer assigned SHAP values for every time point class-by-class and these were used to generate the class-specific SHAP plots (*Figure 5C*). The SHAP values for each time point, across classes, was summed to generate the overall SHAP values for each time point (*Figure 5B*).

## Choice-selective signal

We use an approach developed in *Meister et al., 2013* to estimate the choice-selective signal. We chose such an approach because decision-related activity of PMd neurons does not simply increase or decrease in firing rate and often shows considerable temporal modulation. We estimated for each neuron a choice-selective signal on a time point-by-time point basis as absolute value of the firing rate difference between left and right choice trials (|left - right|) or equivalently PREF-NONPREF. We use this choice-selective signal to understand choice-related dynamics and estimate discrimination time.

## Choice-related dynamics

To understand the dynamics of the choice-selectivity signal as a function of the unsigned checkerboard coherence, we performed the following analysis. As described above, we first estimated the choice-selectivity signal in spikes/s for each neuron and each checkerboard coherence as shown for example in *Figure 7A,B*. We then estimated the slope of this choice-selectivity signal in the 175–325 ms period after checkerboard onset. Repeating this analysis for each color coherence provided us with an estimate of the rate of change of the choice selectivity signal ($\eta$) for all the coherences in spikes/s/s. Averaging over neurons for each cluster provided us with the graphs in *Figure 7C,D*. We then estimated the dependence of $\eta$ on color coherence by regressing $\eta$ and color coherence to estimate how strongly choice-selectivity signals in a particular cluster were modulated by the stimulus input. This modulation is summarized in *Figure 7E* and measured as 'coherence slope' in units of spikes/s/s/% color coherence.

## Discrimination time

We identified the discrimination time, that is the time at which the neuron demonstrated significant choice selectivity, on a neuron-by-neuron basis. We compared the choice-selective signal at each point to the 95th percentile of the bootstrap estimates of baseline choice-selective signal (i.e. before checkerboard stimulus onset). We enforced the condition that the choice-selective signal should be significantly different from the baseline for at least 25 ms after this first identified time to be included as an estimate of a time of significant choice selectivity for that neuron. Using longer windows provided very similar results.

## Experimental subjects (anatomical data)

Archived tissues were harvested from six young rhesus macaques of both sexes (9 ± 1.13 years, *Macaca mulatta*). These subjects were close in age to the macaques used in the main study and were part of part of a larger program of studies on aging and cognition led by Dr. Douglas Rosene. These monkeys were obtained from the Yerkes National Primate Center (RRID:SCR_001914) and housed individually in the Laboratory Animal Science Center at the Boston University School of Medicine; these facilities are fully accredited by the Association for Assessment and Accreditation of Laboratory Animal Care (AAALAC). Research was conducted in strict accordance with the guidlines of the National Institutes of Health's Guide for the Care and Use of Laboratory Animals and Public Health Service Policy on the Humane Care and Use of Laboratory Animals.

## Perfusion and fixation

All brain tissue for histological studies was fixed and harvested using our well-established two-stage perfusion protocol as described (*Medalla and Luebke, 2015*). After sedation with ketamine hydrochloride (10 mg/ml) and deep anesthetization with sodium pentobarbital (to effect, 15 mg/kg i.v.), monkeys were perfused through the ascending aorta with ice-cold Krebs–Henseleit buffer containing (in mM): 6.4 $Na_2HPO_4$, 1.4 $Na_2PO_4$, 137.0 NaCl, 2.7 KCl, and 1.0 $MgCl_2$ at pH 7.4 (Sigma-Aldrich) followed by fixation with 4% paraformaldehyde in 0.1M phosphate buffer (PB, pH 7.4, 37°C). The fixed brain sample was blocked, in situ, in the coronal plane, removed from the skull and cryoprotected in a series of glycerol solutions, and flash frozen in 70°C isopentane (*Rosene et al., 1986*). The brain was cut on a freezing microtome in the coronal plate at 30 μm and were systematically collected into 10 matched series and stored in cryoprotectant (15% glycerol, in 0.1M PB, pH 7.4) at −80°C (*Estrada et al., 2017*).

## Immunohistochemistry

To assess the laminar distribution of interneurons, we batch processed 30 μm coronal sections through the rostral dorsal premotor cortex area (PMdr) from six specimens. Sections were immunolabelled for inhibitory neuronal subtypes based on their expression of calcium binding proteins, calbindin (CB), calretinin (CR), and parvalbumin (PV), which label non-overlapping populations in primates (*DeFelipe, 1997*). Free floating sections were first rinsed (3 x 10 min, 4°C) in 0.01M phosphate-buffered saline (PBS) and incubated in 50 mM glycine for 1 hr at 4°C. Sections were then rinsed in 0.01M PBS (3 x 10 min, 4°C), and antigen retrieval was performed with 10 mM sodium citrate (pH 8.5) in a 60–70°C water bath for 20 min. Sections were then rinsed in 0.01M PBS (3 x 10 min, 4°C) and incubated in pre-block (0.01M PBS, 5% bovine serum albumin [BSA], 5% normal donkey serum [NDS], 0.2% Triton X-100) to reduce any non-specific binding of secondary antibodies. Primary antibodies (*Figure 1*) were diluted in 0.1 M PB, 0.2% acetylated BSA (BSA-c), 1% NDS, 0.1% Triton X-100. To increase the penetration of the antibody, two microwave incubation sessions (2 × 10 min at 150 watts) using the Pelco Biowave Pro (Ted Pella), followed by a 48 hr incubation at 4°C with gentle agitation. After rinsing (3 x 10 min) in 0.01M PBS at 4°C, sections were co-incubated with secondary antibodies diluted in incubation buffer (see 1), microwaved 2 × 10 min at 150 W, and placed at 4°C for 24 hr with gentle agitation. Sections were then rinsed (3 x 10 min) in 0.1M PB, mounted onto slides and coverslipped with prolong anti-fade gold mounting medium (ThermoFisher) and cured at room temperature in the dark.

## Confocal microscopy and visualization of immunofluorescent labeling

Immunofluorescent labeling was imaged using a laser-scanning confocal microscope (Leica SPE) using 488 and 561 nm diode lasers. For each coronal section, two sets of tile scan images of a cortical column, ~200 μm wide and spanning, from pia to the white matter boundary, were obtained in the PMdr. This corresponded to the area 6FR in cytoarchitectural maps (*Barbas and Pandya, 1987*; *Morecraft et al., 2004*; *Morecraft et al., 2019*) and area F7 in several functional maps (*Matelli and Luppino, 1996*; *Rizzolatti et al., 1998*). The two columns were spaced 200 μm apart. All images were acquired using a plain apochromat 40x/1.3 NA oil-immersion objective at a resolution of 0.268 x 0.268 x 0.5 μm voxel size. The resulting image stacks were deconvolved and converted to 8-bit images using AutoQuant (Media Cybernetics; RRID:SCR_002465) to improve the signal to noise ratio (*Medalla and Luebke, 2015*).

## Stereological cell counting

Due to its demonstrated ability in producing minimally-biased results, 3D stereologic cell counting (*Schmitz et al., 2014*) was utilized to count parvalbumin- (PV[+]), calretinin- (CR[+]) and calbindin-positive (CB[+]) cells. Using the CellCounter plugin in Fiji (RRID:SCR_002285) (*Schindelin et al., 2012*) on each image stack after maximum intensity projection, the inhibitory cells were counted slice by slice, recognized by their round shape (as opposed to pyramids), lack of apical dendrite, and relatively high uniform intensity. Cells at the bottom slice of each image stack and touching the left image border were excluded to avoid double-counting.

## Statistics

All statistical tests (Kolmogorov-Smirnov, Kruskal-Wallis, and Mann-Whitney *U*) were conducted using the package scipy.stats (RRID:SCR_008058) (*SciPy 1.0 Contributors et al., 2020*). Multiple comparisons were corrected for using false detection-rate adjusted p-values (Benjamini-Hochberg); this was done using scipy.stats.multitest and scikit-posthocs (RRID:SCR_021363) (*Terpilowski, 2019*). Ordinary least squares regressions were conducted in the package statsmodels (RRID:SCR_016074) (*Seabold and Perktold, 2010*). Bootstrapped standard errors of the median were calculated by taking 5000 random samples with replacement (a bootstrap) of a dataset and then the standard deviation of each bootstrap was taken. Effect sizes were given as adjusted $R^2$ values or Cohen's $f^2$ (of a one-way ANOVA) using statsmodels.formula.api.ols and statsmodels.stats.oneway respectively.

## Acknowledgements

We thank M Noichl and L McInnes for their suggestions on graph community detection and UMAP visualization. We also thank Dr. Kathleen Rockland, Dr. Jennifer Luebke, and Dr. Jonathan Kao for detailed comments on the previous versions of the manuscript.

## Additional information

### Funding

| Funder | Grant reference number | Author |
|---|---|---|
| National Institute of Neurological Disorders and Stroke | R00NS092972 | Chandramouli Chandrasekaran |
| National Institute of Neurological Disorders and Stroke | K99NS092972 | Chandramouli Chandrasekaran |
| Howard Hughes Medical Institute | | Krishna V Shenoy |
| National Institute of Mental Health | R00MH101234 | Maria Medalla |
| National Institute of Mental Health | R01MH116008 | Maria Medalla |
| Whitehall Foundation | 2019-12-77 | Chandramouli Chandrasekaran |
| Brain and Behavior Research Foundation | 27923 | Chandramouli Chandrasekaran |
| NIH Office of the Director | DP1HD075623 | Krishna V Shenoy |
| National Institute on Deafness and Other Communication Disorders | DC014034 | Krishna V Shenoy |
| National Institute on Deafness and Other Communication Disorders | DC017844 | Krishna V Shenoy |
| National Institute of Neurological Disorders and Stroke | NS095548 | Krishna V Shenoy |
| National Institute of Neurological Disorders and Stroke | NS098968 | Krishna V Shenoy |
| Defense Advanced Research Projects Agency | N66001-10-C-2010 | Krishna V Shenoy |
| Defense Advanced Research Projects Agency | W911NF-14-2-0013 | Krishna V Shenoy |
| Simons Foundation | 325380 | Krishna V Shenoy |
| Simons Foundation | 543045 | Krishna V Shenoy |
| National Institute of Neurological Disorders and Stroke | NS122969 | Chandramouli Chandrasekaran |

| Office of Naval Research | N000141812158 | Krishna V Shenoy |
|---|---|---|
| Stanford University | | Krishna V Shenoy |
| Wu Tsai Neurosciences Institute, Stanford University | | Krishna V Shenoy |
| Stanford Engineering | | Krishna V Shenoy |

The funders had no role in study design, data collection and interpretation, or the decision to submit the work for publication.

## Author contributions

Eric Kenji Lee, Conceptualization, Software, Formal analysis, Validation, Investigation, Methodology, Writing - original draft, Project administration, Writing - review and editing, Developed the UMAP and Louvain Clustering method combination that ultimately became WaveMAP and finally the interpretable machine learning approach; Hymavathy Balasubramanian, Data curation, Formal analysis, Investigation, Methodology; Alexandra Tsolias, Data curation, Methodology, Writing - review and editing; Stephanie Udochukwu Anakwe, Data curation, Formal analysis; Maria Medalla, Formal analysis, Investigation, Writing - review and editing, Prof. Medalla led the immunohistochemistry portion of the work in close collaboration with Mr. Kenji Lee, and Ms. Alexandra Tsolias; Krishna V Shenoy, Resources, Funding acquisition, Writing - review and editing; Chandramouli Chandrasekaran, Conceptualization, Data curation, Supervision, Funding acquisition, Investigation, Methodology, Writing - original draft, Project administration, Writing - review and editing, Trained monkeys and recorded in PMd using multi-contact electrodes and provided advice on the development of the WaveMAP approach

## Author ORCIDs

Eric Kenji Lee (iD) https://orcid.org/0000-0002-7166-0909
Hymavathy Balasubramanian (iD) https://orcid.org/0000-0001-5371-2966
Alexandra Tsolias (iD) https://orcid.org/0000-0002-2956-3267
Stephanie Udochukwu Anakwe (iD) https://orcid.org/0000-0001-9236-6090
Maria Medalla (iD) https://orcid.org/0000-0003-4890-2532
Krishna V Shenoy (iD) https://orcid.org/0000-0003-1534-9240
Chandramouli Chandrasekaran (iD) https://orcid.org/0000-0002-1711-590X

## Ethics

Animal experimentation: This study was performed in strict accordance with the recommendations in the Guide for the Care and Use of Laboratory Animals of the National Institutes of Health. All of the procedures were approved were approved by the Stanford Administrative Panel on Laboratory Animal Care (APLAC, Protocol Number 8856, entitled "Cortical Processing of Arm Movements"). Surgical procedures were performed under anesthesia, and every effort was made to minimize suffering. Appropriate analgesia, pain relief, and antibiotics were administered to the animals when needed after surgical approval.

## Decision letter and Author response

Decision letter https://doi.org/10.7554/eLife.67490.sa1
Author response https://doi.org/10.7554/eLife.67490.sa2

# Additional files

## Supplementary files

- Source code 1. MATLAB code and Python notebook for replicating all figures and figure supplements in the manuscript.

- Transparent reporting form

## Data availability

Data generated or analysed during this study are included in the linked Dryad repository (https://doi.org/10.5061/dryad.z612jm6cf). Source data for all figures are also in this zip file.

The following dataset was generated:

| Author(s) | Year | Dataset title | Dataset URL | Database and Identifier |
|---|---|---|---|---|
| Lee EK, Balasubramanian H, Tsolias A, Anakwe S, Medalla M, Shenoy K, Chandrasekaran C | 2021 | WaveMAP analysis of extracellular waveforms from monkey premotor cortex during decision-making | http://doi.org/10.5061/dryad.z612jm6cf | Dryad Digital Repository, 10.5061/dryad.z612jm6cf |

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

# Appendix 1

## Supplementary Information

### Clustering in high-dimensionality and the curse of dimensionality

Clustering in high-dimensions is a difficult problem. In particular, "concentration of measure" results in Euclidean distances, used by k-Means clustering, becoming meaningless as a measure of distance for clustering (*Aggarwal et al., 2001*). Specifically, as dimensionality increases, the difference between distances of randomly chosen points all converge to the same constant distance (*Verleysen et al., 2003*; *Beyer et al., 1999*). UMAP counters this by using graph distances on a nearest neighbor graph which is not susceptible to the concentration of measure phenomenon (*McInnes, 2019*). A common strategy is to cluster in some low-dimensional projected space with a dimensionality reduction method that preserves important latent structure. These use clustering methods like a Gaussian mixture model (GMM) or k-Means but clusters in low-dimension are not necessarily Gaussian in shape (an assumption of both the GMM and k-Means). This violation is induced by the perturbations introduced by non-linear projections even if the distributions were Gaussian in the original high-dimensional space.

### UMAP dimensionality reduction

UMAP is among the class of non-linear dimensionality reductions known as manifold learning algorithms which also includes other well-known methods in neuroscience such as Isomap (*Tenenbaum et al., 2000*) and t-SNE (*Maaten and Hinton, 2008*) (see *Lee and Verleysen, 2007* for a review of methods). Key to this algorithm is the presumption that although data may not be uniformly spaced in the ambient space, it is uniform on some low-dimensional manifold embedded within the high-dimensional space. It is also assumed that the underlying manifold is locally connected (i.e. doesn't have any breaks or isolated points) which is satisfied by the local distance metric unit ball extending to of the 1st-nearest neighbor (*Figure 2A-ii.b*). This leads to the conclusion that the underlying notion of distance (Riemannian metric) changes in each region of the manifold: the notion of a unit distance "stretches" in areas of sparser density and "shortens" in areas of higher density. This is formalized beginning with defining how a local Riemannian metric should be constructed by Lemma 1 of *McInnes et al., 2018*:

### Lemma 1

Let $M$ be a Riemannian manifold equipped with a metric $g$ in ambient space $\mathbb{R}^n$. Let $p \in M$ be a point in this space. If $g$ is locally constant about the point $p$ in an open neighborhood $U$ such that $g$ is a constant diagonal matrix in diagonal coordinates, then in a ball $B \subseteq U$ centered at $p$ with a volume $\frac{\pi^{n/2}}{\Gamma(n/2+1)}$ with respect to $g$, the geodesic distance from $p$ to any point $q \in B$ is $\frac{1}{r}d\mathbb{R}^n(p,q)$, where $r$ is the radius of the ball in the ambient space and $\frac{1}{r}d\mathbb{R}^n(p,q)$ is the existing metric on the ambient space.

Using this definition of $g$, each ball $B$ of fixed volume (using the manifold's distance metric) should contain the same number of data points in $X$ regardless of where on the manifold $B$ is ocated. This also implies that a ball centered on data point $x_i$ should contain the k-nearest neighbors of $x_i$ in a fixed volume no matter which $x_i \in X$ is chosen. Thus the geodesic distance around each data point is normalized by its distance to its k-nearest neighbor and the assumption of uniform sampling on the manifold is enforced.

To compensate for the impact of certain nearest neighbors in the ball lying much further than those closer by (as in very sparse regions in ambient space), the normalizing distances are transformed by the exponential function (*Figure 2A-ii.c*),

$$\sum_{j=1}^{k} \exp\left(\frac{-\left|x_i - x_{i_j}\right|}{r_i}\right) = \log_2(k)$$

To unite these disparate metric spaces (each data point has a unique local notion of distance), category theory is used to show that an equivalent representation can be made from a fuzzy simplicial set via an adjunction that will not be defined here (see Section 3 of *McInnes et al., 2018* and the Nerve Theorem). In this way, the topological structure of the data can be represented as a metric

space or as the union of a set of fuzzy simplices (*Figure 2A-ii.e*). One large benefit of this construction is that while normalized distances in high-dimensional spaces suffer from the Curse of Dimensionality in the form of concentration of measure, normalized nearest neighbor distances do not (*McInnes, 2019*). The end result of this process is an approximation of the topology of the manifold by fuzzy simplicial sets in the form of a Čech complex.

With this fuzzy topological representation, the low-dimensional representation can be found through an optimization procedure that minimizes the cross-entropy of fuzzy simplicial sets containing the same objects and implemented with a force directed graph layout procedure (*Figure 2B-v. a*). Given two fuzzy sets with the same members A and separate membership strength functions μ and ν of Spivak's characteristic form *Spivak, 2009*, the cross-entropy $C((A, \mu), (A, \nu))$ is defined as,

$$C((A,\mu),(A,\nu)) \stackrel{\Delta}{=} \sum_{a \in A} \left( \mu(a) \log \frac{\mu(a)}{\nu(a)} + (1 - \mu(a)) \log \frac{1 - \mu(a)}{1 - \nu(a)} \right).$$

The first term $\mu(a) \log \frac{\mu(a)}{\nu(a)}$ captures the attractive force minimised if short edges in high-dimension correspond to short edges in low-dimension and $(1 - \mu(a)) \log \frac{1 - \mu(a)}{1 - \nu(a)}$ is the repulsive forces that are minimised if long edges in high-dimension correspond to short edges in low-dimension or vice versa.

From a computational perspective, this whole UMAP process proceeds in two steps: construction of a k-nearest neighbor graph (*Figure 2A*) and layout of the graph (*Figure 2B*) into a low-dimensional manifold. Note that after the first step, the *k*-nearest neighbor graph is passed to Louvain community detection (*Figure 2B-iv*) and thus the clustering is not dependent on the embedding of the graph, just on its construction and associated UMAP parameters such as n_neighbors and metric but not layout parameters such as min_dist. The embedding *is* however used for visualization and interpretability and for consistency, the random seed for UMAP's layout procedure is set (UMAP will use gradient descent vs. stochastic gradient descent to compute the force directed graph layout).

## Graph construction

Given a set of data points $X = \{x_1, \ldots, x_N\}$ and a metric $d$, the construction of an undirected weighted $k$-nearest neighbor graph (captured by an adjacency matrix capturing the connection weights between nodes) is conducted using a nearest neighbor descent algorithm (*Dong et al., 2011*). For each data point $x_i \in X$ and fixed nearest neighbor hyperparameter $k$, we have the set $\{x_{i_1} \ldots x_{i_k}\}$ the set of $k$-nearest neighbors of $x_i$ under the local Riemannian metric $d$. We define $\rho_i$ and $\sigma_i$ such that,

$$\rho_i = \min \{d(x_i, x_{i_j}) | 1 \leq j \leq k, d(x_i, x_{i_j}) > 0\}$$

and setting $\sigma_i$ such that,

$$\log_2(k) = \sum_{j=1}^{k} \exp \left( \frac{-\max(0, d(x_i, x_{i_j}) - \rho_i)}{\sigma_i} \right).$$

The weighted graph $\overline{G} = (V, E, w)$ is defined in terms of the vertices $V$, edges $E = \{(x_i, x_{i_j}) | 1 \leq j \leq k, 1 \leq i \leq N\}$, and weight function $w$ as,

$$w((x_i, x_{i_j})) = \sum_{j=1}^{k} \exp \left( \frac{-\max(0, d(x_i, x_{i_j}) - \rho_i)}{\sigma_i} \right)$$

. If $A$ is the weighted adjacency matrix of $\overline{G}$, we can get the undirected weighted graph $B$ by the relationship,

$$B = A + A^\top - A \circ A^\top$$

where o is the Hadamard product.

## Graph layout

The UMAP algorithm finds a low-dimensional projection (manifold) of the high-dimensional data by a force directed layout of the constructed graph. Before this is done though, the graph is spectrally embedded to aid in consistency and convergence of the algorithm through initialization (*Belkin and Niyogi, 2002*; *Bengio et al., 2003*; *Kobak and Linderman, 2019*). The symmetric normalized Laplacian $L$ of the graph is calculated for the 1-skeleton of the weighted graph which is analogous to the Laplace-Beltrami operator (divergence of the gradient, $\Delta f = \nabla^2 f$) on a manifold. If $D$ is the degree matrix (a diagonal matrix containing the degree of each vertex) of the adjacency matrix $A$, we compute the Laplacian matrix as,

$$L = D^{\frac{1}{2}}(D - A)D^{\frac{1}{2}}$$

with associated eigenvectors $y$ and eigenvalues $\lambda$,

$$Ly = \lambda Dy.$$

After the spectral embedding of the graph with Laplacian eigenmaps, the force directed graph layout iteratively applies attractive and repulsive forces on the edges and vertices. The attractive force between two vertices $i$ and $j$ at coordinates $\mathbf{y}_i$ and $\mathbf{y}_j$ with tunable hyperparameters $a$ and $b$ and is determined by,

$$\frac{-2ab\mathbf{y}_i - \mathbf{y}_j^{2(b-1)}2}{1 + \mathbf{y}_i - \mathbf{y}_j^2 2}w((x_i, x_j))(\mathbf{y}_i - \mathbf{y}_j)$$

and the repulsive forces with hyper-parameter $\epsilon = 0.001$ to prevent division by zero,

$$\frac{b}{(\epsilon + \mathbf{y}_i - \mathbf{y}_j^2 2)(1 + \mathbf{y}_i - \mathbf{y}_j^2 2)}(1 - w((x_i, x_j)))(\mathbf{y}_i - \mathbf{y}_j).$$

This optimization procedure is then completed using stochastic gradient descent to arrive at the final embedding.

## Louvain method for community detection

The Louvain method for community detection (here called clustering) (*Blondel et al., 2008*) operates on a weighted network and locates highly-interconnected nodes called a community. This 'connectedness' is measured by their modularity $Q$ (taking real values between $-1$ and 1 inclusive) with added resolution parameter $t$ (*Newman and Girvan, 2004*; *Lambiotte, 2007*) defined as,

$$Q_t = (1 - t) + \frac{1}{2m}\sum_{i,j}[tA_{i,j} - \frac{k_i k_j}{2m}]\delta(c_i, c_j)$$

where $t$ is a parameter controlling the 'characteristic scale' of the communities (*Lambiotte et al., 2008*). The larger the resolution parameter, the fewer the number of communities and the larger their size (*Figure 3—figure supplement 1A*). Smaller values of resolution parameter results in more communities smaller in size. Note that when $t = 1$, the simplified definition of modularity is given as described in *Blondel et al., 2008*. $A_{i,j}$ is an adjacency matrix with the weights of the edges between the nodes indexed by $i$ and $j$ is the sum of weights of the edges connected to the node $i$. $c_i$ is the community to which the node $i$ belongs to and the function $\delta(c_i, c_j)$ is an indicator function that is 1 if $c_i = c_j$ and 0 otherwise. The value $m = \frac{1}{2}\sum_{i,j}A_{i,j}$ which is the sum of all the weights of all the edges in the network. This equation also serves as an objective function for the iterative procedure in *Blondel et al., 2008* which proceeds in two steps: modularity optimization and community aggregation.

### Modularity Optimization

Each node is assigned to its own singleton community (in the initialization step for only the first pass) and then each node is moved into a community with a random neighbor and the change in modularity, $\Delta Q_t$, is calculated. The equation for this change in modularity is,

$$\Delta Q_t = [\frac{\sum_{\text{in}} + k_{i,\text{in}}}{2m} - (\frac{\sum_{\text{tot}} + k_i}{2m})^2] - [\frac{\sum_{\text{in}}}{2m} - (\frac{\sum_{\text{tot}}}{2m}) - (\frac{k_i}{2m})^2].$$

where $\sum_{\text{in}}$ is the sum of all the weights inside the community that the node $i$ is moving into. $\sum_{\text{tot}}$ is the sum of all the weights of the edges to nodes in the community $i$ is moving into. $k_i$ is the sum of all the weighted edges incident on $i$. $k_{i,\text{in}}$ is the sum of the weights from the edges of $i$ to nodes in the cluster. Once the change in modularity is caclulated for a node before and after joining a neighboring cluster, the neighbor joins (or stays with) the community with the largest positive increase in modularity; if no increase can be found, the node remains a part of its current community. Once there can be found no increase in modularity for any nodes, the algorithm proceeds to the second step.

## Community Aggregation

Every node in each community in the previous step is then collapsed into a single node and their edges summed to form a new graph. The process is then repeated from the previous step. This process repeats until the graph with maximum modularity is found and each original node is assigned to a final cluster membership.

The graph produced by UMAP was passed into this Louvain clustering algorithm (*Blondel et al., 2008*) using cylouvain 0.2.3 with parameters in *Table 1*. This clustering method requires no prior specification of the number of clusters that should be present but its number does depend on the resolution parameter. To choose this parameter, a sensitivity analysis was conducted across various values of the resolution parameter with the number of communities and total modularity compared *Figure 3B*. Each waveform was then plotted in UMAP space and color-coded to its associated cluster label found by Louvain Clustering *Figure 3A*.

## Ensemble clustering for graphs (ECG)

ECG is a consensus clustering method for graphs and was used to validate the Louvain clustering algorithm (*Poulin and Théberge, 2018*; *Poulin and Théberge, 2019*). ECG consists of two steps: generation and integration.

### Generation

This step instantiates the ECG algorithm by using Louvain clustering to produce a randomized set of $k$ level-1 partitions $\mathcal{P} \in \{\mathcal{P}_\infty, \ldots, \mathcal{P}_{\|}\}$ (the level-1 refers to only computing the first pass of Louvain). The randomization comes from the randomization of vertices in the initialization step of Louvain clustering.

### Integration

Once the $k$ randomized level-1 Louvain partitions are obtained, Louvain is run on a weighted version of the initial graph $G = (V, E)$. These weights $W_\mathcal{P}(u, v)$ are obtained via co-association of edges $e = (u, v) \in E$. These weights are defined as,

$$W_\mathcal{P}(u,v) \triangleq \begin{cases} w_\star + (1 - w_\star) \cdot (\frac{\sum_{i=1}^{k} v_{\mathcal{P}_i}(u,v)}{k}) & \text{, if } (u,v) \text{ is in the 2-core of } G \\ w_\star & \text{, otherwise} \end{cases}$$

where we have the minimum ECG weight $0 < w_\star < 1$ and the co-occurence of edges $u$ and $v$ as $v_{P_i} = \sum_{j=1}^{l_i} 1_{C_i^j}(u) \cdot 1_{C_i^j}(v)$ where $1_{C_i^j}(u)$ is an indicator function of if the edge $u$ occurs in the cluster of $P_i$ or not.

With this function, ECG combines these level-1 Louvain partitions as a single weighted graph which serves as the result.

## SHapley Additive exPlanations (SHAP)

SHAP values build off of Shapley values (**Shapley, 1988**) and provides interpretability to machine learning models by computing the contributions of each feature towards the overall model. These explanations of machine learning models are models in and of themselves and are referred to as 'additive feature attribution methods'. These explanation models use simplified inputs $x'$ which are mapped to original inputs through a function $x = h_x(x')$ and try to ensure that $g(z') \approx f(h_x(z'))$ whenever $z' \approx x'$ where $f$ is the machine learning model and $g$ is the explanation model. This yields the additive form which is a linear combination of binary variables:

$$g(z') = \phi_0 + \sum_{i=1}^{M} \phi_i z_i'$$

where $z' \in \{0,1\}^M$ is the binary value specifying the inclusion or exclusion of a number of simplified feature inputs $M$ is the effect of each feature.

Work in **Lundberg and Lee, 2017** devises such a model satisfying three important properties within this framework:

### Local accuracy/efficiency

the explanation's features with their effects $\phi_i$ must sum for each feature $i$ to the output $f(x)$.

$$f(x) = \phi_0(f) + \sum_{i=1}^{M} \phi_i(f,x)$$

where $\phi_0(f) = \mathbb{E}[f(z)] = f_x(\emptyset)$

### Consistency/Monotonicity

If a model changes so that the effect of a feature increases of stays the same regardless of other inputs, that input's attribution should not decrease. For any two models $f$ and $f'$ if,

$$f_x'(S) - f_x'(Si) \geq f_x(S) - f_x(Si)$$

where $S \in \mathcal{F}$ are subsets of all features and $Si$ is the setting of feature $i$ to zero (or some background reference value intended to be negligible) then $\phi_i(f',x) \geq \phi_i(f,x)$.

### Missingness

This is the idea that features with no effect on $f_x$ should have no assigned impact $\phi_i$. This is expressed as,

$$f_x(S \cup i) = f_x(S)$$

for all subsets of features $S \in \mathcal{F}$, then $\phi_i(f,x) = 0$.

The authors prove that the only possible additive feature attribution method that satisfies these three criteria is SHAP whose values are computed as the following,

$$\phi_i(f,x) = \sum_{R \in \mathbb{R}} \frac{1}{M!}[f_x(P_i^R \cup i) - f_x(P_i^R)]$$

where $R$ is the set of all feature orderings and $P_i^R$ is the set of all features that come before the $i^{\text{th}}$ one in ordering $R$ and $M$ is the number of input features.

Extending SHAP values to tree classifiers, the authors create shap.TreeExplainer (**Lundberg et al., 2020**) to calculate SHAP values by using path-dependent feature perturbation to yield the plots in **Figure 5B and C**.

# WaveMAP is stable over parameter choice with respect to random seed and data bootstrap

## Parameter Choice

Louvain clustering (*Blondel et al., 2008*) requires the specification of a resolution parameter, $t$, which controls the 'characteristic scale' by which network communities are identified; the larger this parameter, the fewer the number of clusters (communities) detected and vice versa (*Lambiotte, 2007*).

We selected a resolution parameter based on two factors. The most important factor was modularity score. Modularity (the 'connectedness' of a community, see Materials and methods) is a community-wise measure defined as the difference between the weights of the edges within a cluster and the edges incoming from any other node outside of the cluster. Maximizing this value over the whole graph finds communities with high amounts of intra-connectivity and low out-connectivity. The second factor we considered was the sizes of the resulting clusters after choosing a resolution parameter. We did not want clusters with too few members (n < 20 which would be statistically difficult to interpret). The regions with the highest modularity score were at a resolution parameter of 1 and decreased from there onwards. However, choosing a resolution parameter of 1 led to a large number of clusters which often had very few members (*Figure 3—figure supplement 1A*, leftmost column) making downstream statistical comparisons underpowered. We therefore chose $t$ to be 1.5 which resulted in the next best average modularity score of $0.761 \pm 0.004$ (mean $\pm$ S.D.) and an average of $8.29 \pm 0.84$ (mean $\pm$ S.D.) clusters across 25 random data permutations (*Figure 3B*). In this manner, we found a set of waveform clusters that balanced the diversity found by UMAP and statistical interpretability.

## Random seed

To show this hierarchy of clustering resolutions along the curve in *Figure 3C* and to demonstrate *WaveMAP*'s robustness to random seed initialization, we plotted three different plots for several different resolution parameters in *Figure 3—figure supplement 1A*. Each random seed produced the same clustering with only slight perturbations of scale and rotation. To validate these clusters were reliable and not an artifact of our particular data sample, we counted the number of clusters from 100 randomly permuted subsets of the full dataset at varying proportions (from 10% to 100% in 10% increments) and also set each with a different UMAP random seed (*Figure 3—figure supplement 1B*, in red). As the data portion increased, we found that the number of clusters increased then tapered off to around eight clusters at ~60% of the full dataset. In the same manner, we also calculated the adjusted mutual information score (AMI) (*Timme and Lapish, 2018*) between a subset of the data and the full dataset (*Figure 3—figure supplement 1B*, in green). This is a measure of how closely two sets of clusterings agree with each other intuitively interpreted as how 'informative' knowing one potential clustering is for predicting another. Just as with the number of clusters, AMI increases steeply until around 40% at a score of 0.8 of the dataset and then slowly increased. This analysis is reassuring and demonstrates that we have an adequate number of waveforms to yield consistent clustering with which to describe the diversity of cell types in monkey PMd.

## Stability

Louvain clustering is sometimes unstable. That is, results from successive runs on the same data can show considerable variation on some datasets (*Poulin and Théberge, 2019*). To test whether these eight clusters consistently contained the same constituent data points run-to-run, we used ensemble clustering for graphs (ECG) (*Poulin and Théberge, 2018*; *Poulin and Théberge, 2019*). ECG generates $k$ randomized level-1 (one round of Louvain clustering, *Figure 2—figure supplement 1B*) partitions and combines together their graph structure via the co-occurrence of edges between nodes across partitionings. Hence the 'ensemble' in the name (also called 'consensus clustering'). Performing ECG with $k = 10$, 100 times on UMAP graphs with different random seeds produced an average of $8.87 \pm 0.74$ (mean $\pm$ S.D.) clusters which was similar to that found by Louvain clustering with resolution parameter set to 1.5. In addition, the runs of ECG that yielded eight clusters had an almost exact structure to that produced by *WaveMAP*

(compare *Figure 3—figure supplement 1C* to *Figure 3A*). The runs of ECG with more than eight clusters contained small clusters (n ≤ 20) splitting off from which were too small to allow us to make rigorous conclusions statistically. We therefore chose the more conservative eight cluster solution that balanced maximizing cluster number while ensuring adequate cluster sizes.

## WaveMAP's robustness comes from clustering in high-dimension rather than the projected space

Another common approach to clustering high-dimensional data has been to cluster in the embedded space, that is, clustering in the lower-dimensional projected space of a dimensionality reduction method. While this approach is successful for similar applications such as spike sorting (*Dimitriadis et al., 2018*; *Mahallati et al., 2019*; *Harris et al., 2000*; *Chah et al., 2011*), we found it to be more unstable when used for our use case of classifying waveform types (*Figure 4—figure supplement 1*). Using the robustness analysis in *Figure 3—figure supplement 1B*, we compared *WaveMAP* against both DBSCAN (*Ester et al., 1996*) on the embedded 2-dimensional t-SNE (*Maaten and Hinton, 2008*) space (DBSCAN on t-SNE) and a GMM on the projected space formed by the first three principal components of the waveform data (94% variance explained). As before, we first applied each method to the full dataset to construct a 'reference' to compare against. Thus each method is compared only to its own reference as to facilitate a fair head-to-head comparison.

DBSCAN on t-SNE and the GMM on PCA formed what seemed to be reasonable clusterings of the data with DBSCAN on t-SNE forming clusters similar to *WaveMAP* (*Figure 4—figure supplement 1A*, top) and GMM on PCA forming four clusters just as in *Figure 4B* albeit in PCA- and not the feature-space (*Figure 4—figure supplement 1A*, bottom). When analyzing the AMI of clusterings across data bootstraps, *WaveMAP*'s AMI was moderate (~0.5) even at low percentages of the full dataset and quickly increased to high values (> 0.8) from about 50% of the dataset onwards (*Figure 4—figure supplement 1B*, in blue). However, when we conducted our robustness analysis on the other methods, both of them proved to be less stable (*Figure 4—figure supplement 1B*, green and red). With DBSCAN on t-SNE, although it was able to reach high AMI values when the data fraction increased past 75%, it was low to moderate before this point. In this regime of high data fraction, many of the points between random samples were shared and so much of this stability may be due to similarity in the data (when the fraction of the full dataset is high, most of the data is the same between random samples). Ideally, a method should be able to pick up on structure predictive of out-of-sample data at low data fractions i.e. results should generalize. With the GMM on PCA approach, AMI started out about as high as *WaveMAP* but failed to increase as more data was included. Examining the individual samples, we see that at 40% of the full dataset (*Figure 4—figure supplement 1C*, left), a continuum of AMI values are occupied from low to high. At 90% of the full dataset (*Figure 4—figure supplement 1C*, right), the AMI scores cluster around different locations and overall occupying the same range as at 40%. This suggests that the GMM on PCA, having nearly the full dataset, is converging to one of several solutions (local minima) with some of them being highly suboptimal.

