## [Decision Letter]

**Acceptance summary:**

This article will be of interest to neurophysiologists interested in identifying different neuronal types from extracellular recordings, which is a difficult computational task. It describes novel data-driven methods for functionally dissociating circuits in the primate brain: using a combination of unsupervised dimensionality reduction techniques and clustering methods to categorize extracellular spike waveform profiles (the signatures of different neuronal types), the authors identified cell types with distinct functional and neurophysiological properties in a dataset collected from the premotor cortex of two macaques. The authors went to great lengths to validate their results with clear and informative visualizations, and to outline the capabilities and limitations of this promising cell identification technique.

**Decision letter after peer review:**

Thank you for submitting your article "Non-linear Dimensionality Reduction on Extracellular Waveforms Reveals Cell Type Diversity in Premotor Cortex" for consideration by *eLife*. Your article has been reviewed by 2 peer reviewers, and the evaluation has been overseen by a Reviewing Editor and Michael Frank as the Senior Editor. The reviewers have opted to remain anonymous.

Essential Revisions:

1) One potential weakness is the technical presentation of the new technique in the Supplementary information, which although complete, does not allow the targeted audience to create an intuition of it, or be able to replicate the analysis. Describing the Uniform Manifold Approximation and Projection (UMAP) idea to an audience not versed in topology and category theory is not easy. Yet the manuscript would increase its already significant impact if a small effort was put to describe UMAP both technically (as it is nicely done) and more intuitively (maybe with the help of a schematic or two). Including specific computer code would also make these methods more accessible.

2) The reviewers noted a few additional analyses that would provide further validation of the methods and a better sense of how they would generalize to other datasets. Please address the individual commentaries listed in their additional recommendations.

3) Reviewer 1 also raised a few questions that would benefit from clarification and refinement (in additional recommendations). Although not critical, addressing them would benefit the presentation and should be straightforward.

*Reviewer #1:*

1. When discussing hyperparameter optimization and gridsearch, is the entire dataset used in deriving the best values? The authors acknowledge the data-leakage in conducting cross-validation, which seems sound given the questions pursued in this paper; however, the hyperparameters themselves would also be a potential source of leakage and should be addressed/discussed (primarily for UMAP and Louvain, random forests with lesser importance).

2. The sentence "This algorithm is also fairly deterministic (after fixing a random seed) …" was confusing with the context of fixing a random seed. If it is deterministic, then one wouldn't need to fix the random seed, correct?

3. When conducting classification using random forests, do the authors utilize the normalized waveforms (i.e., same input going into WaveMAP of 1.4ms at 30 kHz; 42 samples or features)? An explicit mention of vector length would be helpful to the reader.

4. Are there any special considerations when applying UMAP to time-series data? The canonical examples of UMAP do not contain the autoregressive aspects of spiking waveforms and it might be worthwhile to mention the implications, if any, on the effectiveness of the method.

5. It is not clear how many models are being trained/tested for the generation of confusion matrices (e.g. Figure 2C). Are the binary classifiers trained to distinguish between data points from 1 vs other 7 clusters or are there different models for classification between each possible pair of clusters?

6. The authors indicate that normalization was conducted as a preprocessing step. Are there any concerns on whether normalization might be removing a potential feature that may factor into further classification between spikes? Is it a possibility that adding amplitudes would aid the clustering procedure (e.g., by including the scaling parameter for each spike as an additional feature at the end of the feature vector) or is there a methodological constraint that renders non-normalized amplitudes useless?

7. The authors addressed the issue of stability of WaveMAP, as it pertains to applying different seeds on the entire data as well as the robustness of the 8-cluster solution on varying sizes of data. My understanding is that the authors did not confirm whether those 8-cluster solutions were indeed similar (i.e. spikes clustered with neighboring spikes from the 100% dataset model) when varying data size. A third test is needed that takes a subset of the dataset (e.g., 80%; a subset that would still on average produce an 8-cluster solution) and tests whether the spikes would consistently cluster with their neighbors in the 100% data model. While this test is posthoc and the full data are still incorporated in the remainder of analysis in the paper, this might help to demonstrate the stability of the solution.

*Reviewer #2:*

1. The results rest a lot on appropriate spike sorting. Given that the probe used will not allow any of the 'triangulation' spike-sorting based methods to work (too spaced out electrodes), it would be useful for the authors to provide a bit more information (maybe in their Methods section) as to how spike-sorting was achieved. Spike selection description I find adequate, but describing the spike-sorting method as "inspection of extracellular waveforms and subsequent offline spike sorting (Plexon Offline Sorter)" doesn't really allow the reader to form a picture of the likelihood of having 'single units' that are not really single. Particularly, nothing is mentioned about thresholding of spikes. I am not suggesting a thorough description of the spike sorting method but some information on the techniques used (within the Offline sorter and/or anything else) and a report on the authors' confidence of the single units produced (and how such confidence was evaluated). I understand that the literature utilising the types of probes used by the authors is also most of the times vague as to what spike-sorting really means, yet in most cases the assignment of spikes to single units does not bare the same significance to the subsequent results as in the case of this work, hence the above argument for some more clarity on the matter.

2. A purported advantage of UMAP is that, in principle, it can deal with a very large number of features (better than t-SNE, which struggles with > 50 features or so). However, the technique is still new and this belief is largely anecdotal. It would be beneficial, to both the neuroscientific community and in general to anyone considering using UMAP, if the authors could make a comparison between using the raw data set in UMAP and using a dimensionality reduced one through PCA giving only the top PCs to UMAP. Practically provide some evidence to support your choice of using the raw data set. Also providing an idea of the "complexity" of the data set would be useful (maybe by mentioning the variance explained by different number of principal components after a PCA, even if you do not end up using the primary components generated).

3. The choice of the resolution parameter for the clustering algorithm (line 873 and Figure 2B) has rather important consequences in the subsequent analysis. The choice should be better justified.

a. The authors say they chose the resolution parameter at the elbow of the number of clusters, but that is not the case. The elbow point is the immediate next one (resolution of 2 and number of clusters probably 7).

b. The "number of clusters" value that was the most stable over the resolution is 5 (given resolutions between 3 and 5 if I read the graph properly), which might be another logical assumption for picking the used resolution.

c. So although the argument that choosing a smaller resolution leads to too many clusters with too few samples in them is a valid one, nonetheless there seems to be no real argument against choosing a resolution that would lead to a smaller number of clusters. Since this clustering impacts the whole of the subsequent analysis, it would be important to understand the authors' reasoning.

d. It would be interesting for example to see how a smaller number of WaveMAP generated clusters would fare in the learnability test shown by the authors (starting at line 188).

4. Comparison to t-SNE. This comment has been prompted by the phrase "While providing visualization, these methods are difficult to cluster upon because they return a different mapping on every initialization." (line 546). The authors provide reference 155 that actually directly contradicts this statement. Although the other comments about t-SNE and UMAP are valid (reverse mapping, and use of labels in training), I believe the first comment requires either removal or better further exploration. It would be very informative if the authors did a t-SNE embedding (on the data after PCA) and then used a DBSCAN to classify the cells. This would be rather straight forward to do at their data set size, even multiple times for different initialisations (but see Dimitriadis et al., 2018a for a GPU accelerated t-SNE algorithm and also for t-SNE and DBSCAN used in spike-sorting). That result can then also be contrasted to the WaveMAP and GMM methods using their random forest classifier. Even if the results from the t-SNE / DBSCAN clustering are comparative to WaveMAP the reverse mapping capability of UMAP is definitely a strength that is very nicely utilised in this work and which any other non-linear dimensionality reduction technique doesn't have. As a final point, in the comparison section the authors could also add that t-SNE is way slower than UMAP and this speed difference starts to be very obvious (minutes vs days) in sample sizes in the hundreds of thousands to millions (i.e. the regime of spike collection that high density probes collect Dimitriadis et al. 2018b and Steinmetz et al., 2018).

5. Number of waveform features used to classify the cells. The authors compare their WaveMAP to a seemingly standard technique in the macaque literature (or maybe in the literature that uses sparse electrode probes), that uses 3 features of the waveform to classify the cells. This is not the way things are done in most of the rodent electrophysiology (or works that use denser probes or tetrodes). The standard technique is to PCA each spike form and use the first 3 components per electrode. In the case of this work that would lead to the use of the same number of features, but (probably) much larger retention of information. See Harris et al., 2000 (and references therein) for how the PCA of the waveforms is used in tetrode recordings to do spike-sorting. More recent spike-sorting algorithms use template matching of the waveform. For instance, Kilosort (Pachitariu et al., 2016) omits spike detection and PCA, which can miss useful information. Instead, it relies on identifying template waveforms and their timing properties, in order to assign spikes (in the entirety of their waveform) to different cells. Also comparing the WaveMAP process to the clustering results (following the used Generalised Gaussian Models method, or other peak-density detection methods) of this feature set would be both more informative and more valid for the larger electrophysiology community.

1. Harris, K. D., Henze, D. A., Csicsvari, J., Hirase, H., and Buzsaki, G. (2000). Accuracy of tetrode spike separation as determined by simultaneous intracellular and extracellular measurements. Journal of neurophysiology, 84(1), 401-414.

2. Pachitariu, M., Steinmetz, N., Kadir, S., and Carandini, M. (2016). Kilosort: realtime spike-sorting for extracellular electrophysiology with hundreds of channels. BioRxiv, 061481.

3. Dimitriadis, G., Neto, J. P., and Kampff, A. R. (2018). T-SNE Visualization of Large-Scale Neural Recordings. Neural Computation, 30(7), 1750-1774. https://doi.org/10.1162/neco_a_01097

4. Dimitriadis, G., Neto, J. P., Aarts, A., Alexandru, A., Ballini, M., Battaglia, F., Calcaterra, L., David, F., Fiáth, R., Frazão, J., Geerts, J. P., Gentet, L. J., Helleputte, N. V., Holzhammer, T., Hoof, C. van, Horváth, D., Lopes, G., Lopez, C. M., Maris, E., … Kampff, A. R. (2018). Why not record from every channel with a CMOS scanning probe? BioRxiv, 275818. https://doi.org/10.1101/275818

5. Steinmetz, N. A., Zatka-Haas, P., Carandini, M., and Harris, K. D. (2018). Distributed correlates of visually-guided behavior across the mouse brain. BioRxiv, 474437. https://doi.org/10.1101/474437

---

## [Author Response]

Essential Revisions:1) One potential weakness is the technical presentation of the new technique in the Supplementary information, which although complete, does not allow the targeted audience to create an intuition of it, or be able to replicate the analysis. Describing the Uniform Manifold Approximation and Projection (UMAP) idea to an audience not versed in topology and category theory is not easy. Yet the manuscript would increase its already significant impact if a small effort was put to describe UMAP both technically (as it is nicely done) and more intuitively (maybe with the help of a schematic or two). Including specific computer code would also make these methods more accessible.

We thank the editor and reviewers for this comment on how to improve the impact of our work. Our intent with Figure 2-Supplement 1 in the manuscript was to provide this sort of intuition, and we completely agree that this intuition can be missed for the reader who does not focus on the Supplementary Information. We also recognize that spending a large amount of time reading supplementary information before understanding the conclusions of the paper can be unappealing for many readers.

To address this potential weakness we have adopted a three step strategy. First, we have now included an explicit figure (Figure 2 in the manuscript) to schematize WaveMAP and also incorporate some additional text on the principles underlying UMAP and Louvain clustering. We have also added a full methods section (“A step-by-step guide to UMAP and Louvain clustering in WaveMAP") describing the intuition for WaveMAP in detail and expanding on the steps shown in the schematic. Finally, for the reader interested in the mathematical nuances of the UMAP and Louvain clustering approach, we have included the detailed mathematical descriptions in the Supplementary Information (“UMAP dimensionality reduction"). We are confident that these three pieces now included in the manuscript should allay the potential weakness in our exposition. The following text is now provided in the results:

“In WaveMAP (Figure 2), we use a three step strategy for the analysis of extracellular wave- Main Q1 forms: We first passed the normalized and trough-aligned waveforms (Figure 2A-i) into UMAP to obtain a high-dimensional graph (Figure 2A-ii) (McInnes et al., 2018). Second, we used this graph (Figure 2B-iii) and passed it into Louvain clustering (Figure 2B-iv, Blondel et al., 2008), to delineate high-dimensional clusters. Third, we used UMAP to project the high-dimensional graph into two dimensions (Figure 2B-v). We colored the data points in this projected space according to their Louvain cluster membership found in step two to arrive at our final WaveMAP clusters (Figure 2B-vi). We also analyzed the WaveMAP clusters using interpretable machine learning (Figure 2B-vii) and also an inverse transform of UMAP (Figure 2B-viii). A detailed explanation of the steps associated with WaveMAP is available in the methods, and further mathematical details of WaveMAP are available in the supplementary information.”

Code: In our original submission, we had included a Google Colab notebook for the readers and reviewers to try out our method. We also highlight this now in the methods section of the manuscript under the “Code and Data Availability Section" so that readers can follow along by executing code if they wish to replicate various figures. These will be provided with the paper if accepted for publication. We have added the following text to the code and data availability statement at the start of our Methods section:

“All figures and figure supplements can be generated from the code and data included

with the manuscript. Figure 1 is generated using MATLAB whereas all other figures were generated using the Jupyter/Google Colab notebook available with this manuscript. Please see the Readme.md file included in the zip file for further instructions. Note, we have not included the raw firing rates across conditions for the neurons because of the large size of the data. This data can be made available by emailing the authors. Further information about WaveMAP and updated notebooks can also be obtained from the Chandrasekaran lab website at Boston University (http://www.chandlab.org).”

Since the publication of our bioRxiv paper, a few researchers have approached us to use WaveMAP. We have shared some of this computer code and they have been able to get it working for their use case. In particular, Paulk et al., (2021) used WaveMAP to understand the types of neurons observed in human dorsolateral prefrontal cortex. We are therefore condent that the included code will be helpful for other researchers who want to use WaveMAP in their workows.

2) The reviewers noted a few additional analyses that would provide further validation of the methods and a better sense of how they would generalize to other datasets. Please address the individual commentaries listed in their additional recommendations.

We have now responded to each of the comments in detail. Please see our responses below.

3) Reviewer 1 also raised a few questions that would benefit from clarification and refinement (in additional recommendations). Although not critical, addressing them would benefit the presentation and should be straightforward.

We have now edited the manuscript and responded to these comments and made changes to the manuscript to improve clarity.

Reviewer #1:1. When discussing hyperparameter optimization and gridsearch, is the entire dataset used in deriving the best values? The authors acknowledge the data-leakage in conducting cross-validation, which seems sound given the questions pursued in this paper; however, the hyperparameters themselves would also be a potential source of leakage and should be addressed/discussed (primarily for UMAP and Louvain, random forests with lesser importance).

We apologize for any confusion but are a bit unsure of if this comment is referring to either (1) potential data leakage through the tuned hyperparameters of the classifier, (2) tuning of parameters in WaveMAP to maximize classification accuracy, or (3) the application of WaveMAP to the entire dataset, rather than a subset, resulting in potential overfitting. Below, we address all three of these potential points.

To address (1), we used a test-train split with k-fold cross-validation instead of just a simple test-train split when training the classifier. In the test-train split with k-fold cross-validation, a test set is put aside for final model evaluation (not tuning) and the training data is split further into another training set and a validation set. Unfortunately, the terminology does not differentiate between the training data the k-folds algorithm is applied to and the training data subset produced by the k-folds. The training-validation split is determined by k-fold splitting (five folds used in the manuscript thus an 80:20 ratio five times) and is used to train the model (training set) and tune hyperparameters (validation set). The test set is never seen by the training and hyperparameter optimization procedure and is only used after model tuning as a final evaluation of performance. We have revised the manuscript to clarify our procedure. We also corrected in the manuscript erroneous mentions of using a random forest classifier as we actually implemented a gradient boosted decision tree.

“To validate that WaveMAP finds a “real" representation of the data, we examined if a

very different method could learn the same representation. We trained a gradient boosted decision tree classifier (with a softmax multi-class objective) on the exact same waveform data (vectors of 48 time points, 1.6 ms time length) passed to WaveMAP and used a test-train split with k-fold cross-validation applied to the training data. Hyperparameters were tuned with a 5-fold cross-validated grid search on the training data and final parameters shown in Table 2. After training, the classification was evaluated against the held-out test set (which was never seen in model training/tuning) and the accuracy, averaged over clusters, was 91%.”

In regards to (2), we did not iteratively optimize or tune the parameters or hyperparameters of UMAP and Louvain to maximize classification accuracy in the boosted decision tree classifier. Because we did not parameter tune across a test-train split, we believe data leakage (at least in the conventional, deleterious sense) does not apply. The only parameter we changed relative to defaults was a slightly more conservative setting of 20 for the n neighbors parameter for UMAP. This change does affect the number of clusters since it affects the graph construction but we demonstrate that WaveMAP performs better than the GMM on features regardless of the value chosen for the n neighbor parameter (see Figure 4-Supplement 2). Tuning parameters that change cluster number, whether n neighbors or resolution, had little effect on classifier performance. WaveMAP yielded mappings that were more generalizable than a GMM on features across every number of clusters and both parameters investigated in the manuscript. Furthermore, WaveMAP maintained high separability in its clusters even across a wide range of n neighbors parameter values while the separability of clusters for the GMM on features method fell quickly.

We have included the following text in the manuscript .

“In fact, across all cluster numbers (n components from 2 to 16), a classifier tuned for the GMM performed more poorly on the GMM labels than a WaveMAP projection with the same number of clusters (Figure 4-Supplement 2E, in red). Tuning WaveMAP parameters that induce different cluster numbers, whether n neighbors (in dark blue) or resolution (in light blue), had little effect on classifier performance (Figure 4-Supplement 2E, in blues). WaveMAP yielded mappings that were more generalizable than a GMM on features across every number of clusters and both parameters investigated.”

In regards to (3), it could be that applying a transformation to the entire dataset before splitting the data could lead to potential overfitting. This is an excellent point getting at the intrinsic difficulty of applying data-driven methods to unlabeled data (where ground truth is not available). By construction, the very transformations themselves are defined by the data. Note, that using the full dataset is not unique to WaveMAP and is also a problem with GMM approaches as well.

In an ideal scenario, we would follow the recommended procedure of Moscovich and Rosset (2019) by (1) splitting the dataset, (2) applying a transformation learned from the training set to the test set, and (3) evaluating the discrepancy with a loss function. However, this procedure only works if there is some well-defined loss function incorporating ground truth. In our case, this would be “ground truth clusters". To mitigate this intrinsic difficulty, we used sufficiently large sample counts (number of neurons) to close the theoretical gap between validation error (discrepancy between train and test sets) and generalization error (discrepancy between data we do and don't have, see Figure 1 of Moscovich and Rosset, 2019). We demonstrate that sample counts are in a stable regime of cluster number presumably close to the ground truth number of clusters (Figure 3-Supplement 1B of the manuscript).

“WaveMAP was consistent in both cluster number (Figure 3-Supplement 1B, red) and cluster membership (which waveforms were frequently “co-members" of the same cluster; Figure 3-Supplement 1B, green).”

Finally, to address the notion that we are just simply finding a better parcellation of waveforms rather than anything having to do with real cell types, we also applied WaveMAP in a setting where we did know the ground truth cell types (mouse somatosensory cortex juxtacellular waveform data from Yu et al., (2019)). Alignment to ground truth was assessed through a measure called “homogeneity score" (Rosenberg and Hirschberg, 2007) in which clusterings with only one ground truth type in each cluster are given a score of 1.0 and clusterings with equal mixtures of each ground truth class is given 0.0.

In this way, we assess how aligned either method is to ground truth cell types and find that WaveMAP matches better with ground truth cell types than a GMM on features (0.67 vs. 0.41 homogeneity score). Homogeneity score is used (instead of AMI as later in our analyses) because it does not penalize a clustering for over-splitting a ground truth cluster. Homogeneity only penalizes a clustering if different ground truth types are incorporated into one cluster. WaveMAP tends to align its clusterings with the borders of ground truth cell types (Author response image 1) while GMM on features more often cuts through ground truth groups forming clusters with mixtures of types (Author response image 1). In the future, we hope to follow up on this promising result from mice in monkeys using optogenetics and in-vivo recordings.

**Author response image 1. respfig1:** WaveMAP clusters align closer to ground truth cell types than GMM on features. (**A**) At left, WaveMAP was applied to juxtacelluar waveforms of known cell type (Yu et al., 2019) withthe same parameters as in the manuscript. Cell types were identified through optogenetic-tagging and histochemical verification. At right, the ground truth cell types are displayed in the UMAP projected space. Homogeneity score for WaveMAP was 0.67. (**B**) At left, a Gaussian mixture model is applied in the feature space that appears in Figure 1D of (Yu et al., 2019); VIP-positive neurons were excluded for relatively lowcounts (n = 8) and some excitatory cells with very large peak-to-trough ratio were capped at a value of 10. At right, the ground truth cell types are shown in the feature space. Homogeneity score for the GMM on features was 0.41.

2. The sentence "This algorithm is also fairly deterministic (after fixing a random seed) …" was confusing with the context of fixing a random seed. If it is deterministic, then one wouldn't need to fix the random seed, correct?

We apologize for the confusing wording and have removed this sentence and revised our manuscript accordingly. The UMAP algorithm is non-deterministic by default but by setting a seed, it is made deterministic (this makes the algorithm implement gradient descent instead of stochastic gradient descent). More importantly, random seed only affects the particular layout of the points in the projected 2-dimensional space but is of no consequence to the clustering. Our procedure clusters on the network graph before this layout step, and is therefore not affected by the random seed. The fixing of the seed

is only used to reproduce the same “visualization" from run to run.

“We find that cluster memberships found by WaveMAP are stable with respect to random seed when resolution parameter and n neighbors parameter are fixed. This stability of WaveMAP clusters with respect to random seed is because much of the variability in UMAP layout is the result of the projection process (Figure 2B-v.a). Louvain clustering operates before this step on the high-dimensional graph generated by UMAP which is far less sensitive to the random seed. Thus, the actual layout of the projected clusters might differ subtly according to random seed, but the cluster memberships largely do not (see Supplementary Information and columns of Figure 3-Supplement 1A). Here, we fix the random seed purely for visual reproducibility purposes in the figure. Thus, across different random seeds and constant resolution, the clusters found by WaveMAP did not change because the graph construction was consistent across random seed at least on our dataset (Figure 3-Supplement 1A).”

3. When conducting classification using random forests, do the authors utilize the normalized waveforms (i.e., same input going into WaveMAP of 1.4ms at 30 kHz; 42 samples or features)? An explicit mention of vector length would be helpful to the reader.

Thank you for this comment. We used the same normalized waveforms for the classification as for the WaveMAP analysis. We have revised our manuscript to make this more explicit. We note that the waveforms are actually 1.6 ms in length and not 1.4 ms. As noted above, we also corrected in the manuscript erroneous mentions of using a random forest classifier as we actually implemented a gradient boosted decision tree.

“We trained a gradient boosted decision tree classifier (with a softmax multi-class objective) on the exact same waveform data (vectors of 48 time points, 1.6 ms time length) passed to WaveMAP and used a test-train split with k-fold cross-validation applied to the training data.”

4. Are there any special considerations when applying UMAP to time-series data? The canonical examples of UMAP do not contain the autoregressive aspects of spiking waveforms and it might be worthwhile to mention the implications, if any, on the effectiveness of the method.

This is correct: there is certainly additional autocorrelative structure in the data that is

perhaps not efficiently found by UMAP which treats each time point as independent. However, we don't note any deleterious effects on the analysis and a few other studies have deployed these methods on time-series data successfully (Ali et al., 2019; Gouwens et al., 2020; Jia et al., 2019; Sedaghat-Nejad et al., 2021). We have updated the Discussion of the manuscript to mention this:

“We also note that while traditional uses of non-linear dimensionality reduction have been applied to data lacking autoregressive properties, such as transcriptomic expression (Becht et al., 2018), this doesn't seem to be an issue for WaveMAP. Even though our waveforms have temporal autocorrelation, our method still is able to pick out interesting structure. Other work has found similar success in analyzing time series data with non-linear dimensionality reduction (Ali et al., 2019; Gouwens et al., 2020; Jia et al., 2019; Sedaghat-Nejad et al.,2021).”

5. It is not clear how many models are being trained/tested for the generation of confusion matrices (e.g. Figure 2C). Are the binary classifiers trained to distinguish between data points from 1 vs other 7 clusters or are there different models for classification between each possible pair of clusters?

We realize we were unclear in our exposition and have revised our manuscript accordingly.

We mistakenly stated that our classifier was using a logistic objective function for binary classification. The gradient boosted decision tree used was actually a single multi-class classifier with the softmax objective function (multivariate logistic regression, xgboost). XGBoost.XGBClassifier conducts the classification simultaneously: the classifier assigns a probability to each class for each data point. It then chooses the class with greatest probability and outputs this as the predicted class label. We have updated the manuscript accordingly in the Results and Supplementary Figure sections. This comment also helped us realize we used a gradient boosted decision tree rather than a gradient boosted random forest algorithm.

“To validate that WaveMAP finds a “real" representation of the data, we examined if a

very different method could learn the same representation. We trained a gradient boosted decision tree classifier (with a softmax multi-class objective) on the exact same waveform data (vectors of 48 time points, 1.6 ms time length) passed to WaveMAP and used a test-train split with k-fold cross-validation applied to the training data. Hyperparameters were tuned with a 5-fold cross-validated grid search on the training data and final parameters shown in Table 2.”

6. The authors indicate that normalization was conducted as a preprocessing step. Are there any concerns on whether normalization might be removing a potential feature that may factor into further classification between spikes? Is it a possibility that adding amplitudes would aid the clustering procedure (e.g., by including the scaling parameter for each spike as an additional feature at the end of the feature vector) or is there a methodological constraint that renders non-normalized amplitudes useless?

This is an interesting suggestion. We applied WaveMAP to unnormalized waveforms but

found that it failed to focus on any structure beyond spike amplitude. We intuit that this is similar to the reasons for normalizing prior to principal component analysis: if unnormalized data is passed into this method, most of the variance explained is “eaten up" by data points with large variance. Similarly, we think that UMAP tries to explain more of the variability induced by large amplitude spikes at the expense of other features. We evaluated the influence of amplitude by applying UMAP to unnormalized average spike waveforms, and we found that it led to a loss of interpretable structure in the appearance of clustered points in the projected space. Intuitively, UMAP is focused on explaining the large amount of variation due to spike amplitude to the detriment of other features more related to cell type. This is readily apparent when coloring the points associated with each waveform in the projected space by the spike amplitude .

Exclusion of spike amplitude is perhaps not problematic because spike amplitude in extracellular settings does not seem to be an important differentiator of most cell types. Amplitude in extracellular settings varies according to three dimensions orthogonal to cell type differences: (1) it attenuates with distance from the recording electrode (Gold et al., 2006); (2) it varies in different parts of the neuron and is probably greatest at the soma or initial segment (Gold et al., 2006); and (3) gradually decreases during sustained firing in a spike train (Quirk and Wilson, 1999). So although some cell types exhibit different amplitude spikes as a reliable differentiator (Betz cells have extremely large spike amplitudes), most of spike amplitude variation is unrelated to cell type differences. We have updated the appropriate section in the Discussion to make this point clearer.

This is now included in panels E and F of Figure 3-Supplement 2; the text below is now included in the discussion portion of the manuscript:

Waveform cluster shapes are unlikely to arise from electrode placement

It is a possibility that the diversity of waveforms we observe is just an artifact of electrode placement relative to the site of discharge. This supposes that waveform shape changes with respect to the distance between the neuron and the electrode. This is unlikely because both in vitro studies (Deligkaris et al., 2016) and computational simulations (Gold et al., 2006) show distance from the soma mostly induces changes in amplitude. There is a small widening in waveform width but this occurs at distances in which the amplitude has attenuated below even very low spike thresholds (Gold et al., 2006). We controlled for this cell type-irrelevant variation in amplitude by normalizing spikes during preprocessing. It should also be noted that without any normalization, all structure was lost in the UMAP projection which instead yielded one large point cloud (Figure 3-Supplement 2E). Intuitively, this can be understood as UMAP allocating most of the projected space to explaining amplitude differences rather than shape variation. This can be visualized by coloring each point by the log of the amplitude of each spike (log of difference in maximum vs. minimum values) and observing that it forms a smooth gradient in the projected space (Figure 3-Supplement 2F).

7. The authors addressed the issue of stability of WaveMAP, as it pertains to applying different seeds on the entire data as well as the robustness of the 8-cluster solution on varying sizes of data. My understanding is that the authors did not confirm whether those 8-cluster solutions were indeed similar (i.e. spikes clustered with neighboring spikes from the 100% dataset model) when varying data size. A third test is needed that takes a subset of the dataset (e.g., 80%; a subset that would still on average produce an 8-cluster solution) and tests whether the spikes would consistently cluster with their neighbors in the 100% data model. While this test is posthoc and the full data are still incorporated in the remainder of analysis in the paper, this might help to demonstrate the stability of the solution.

This is an excellent point and something not fully addressed in our original mansucript. We have updated Figure 3-Supplement 1B in the manuscript. We calculated the adjusted mutual information score (AMI) between the WaveMAP clusters on randomly sampled subsets vs. WaveMAP on the full dataset and across 100 random samplings. AMI is a measure of how correlated two variables are, adjusted for correlation due to random chance and is a measure of how much information we receive about one variable given observation of another and vice versa (see note at end of response for a detailed description of this metric). Since subsets don't contain all the data points in the dataset, we drop the points from the full dataset missing from the subset when we make our AMI calculation (with the appropriate normalizations given clusterings are of different number, by construction). We find that across a large range of data subsets the clusterings largely agree with the solution using the full dataset. This is shown through the relatively high AMI scores once half of the dataset is included in the analysis. In addition, AMI scores tend to level off at approximately the same time as does the number of Louvain communities (clusters) which suggests that WaveMAP is consistently clustering the same units into the same fixed number of clusters.

We also compared WaveMAP to other methods suggested by Reviewer 2 (DBSCAN in t-SNE projected space and a GMM on PCA-projected data) using the same data subsetting framework (Figure 4-figure supplement 1). Although both DBSCAN on t-SNE and GMM on PCA produce what appears to be sensible clusters (Figure 4-figure supplement 1A), they exhibit suboptimal properties when AMI is examined vs. data fraction. While WaveMAP starts at a moderate AMI and steadily increases, DBSCAN on t-SNE starts very low and only reaches high levels at large data fraction. GMM on PCA's AMI does not increase with increasing data fraction (Figure 4-figure supplement 1B). In addition, GMM on PCA's AMI scores contract to different values at high data fraction suggesting that this method is switching between different solutions (local minima, perhaps what you were referring to with your comment about WaveMAP; Figure 4-figure supplement 1C, right). Comparison of WaveMAP to these other methods|which cluster in the embedded space suggests that clustering on the high-dimensional graph, before dimensionality reduction for visualization, is key to WaveMAP's stability in terms of both consistent numbers of clusters and cluster membership. The text below is now included in the manuscript and is panel B of Figure 3-Supplement 1.

“We also found that WaveMAP was robust to data subsetting (randomly sampled subsets of the full dataset, see Supplementary Information Tibshirani and Walther, 2012), unlike other clustering approaches (Figure 3-Supplement 1B, green, Figure 4-Supplement 1). We applied WaveMAP to 100 random subsets each from 10% to 90% of the full dataset and compared this to a “reference" clustering produced by the procedure on the full dataset. WaveMAP was consistent in both cluster number (Figure 3-Supplement 1B, red) and cluster membership (which waveforms were frequently “co-members" of the same cluster; Figure 3-Supplement 1B, green).”

Reviewer #2:1. The results rest a lot on appropriate spike sorting. Given that the probe used will not allow any of the 'triangulation' spike-sorting based methods to work (too spaced out electrodes), it would be useful for the authors to provide a bit more information (maybe in their Methods section) as to how spike-sorting was achieved. Spike selection description I find adequate, but describing the spike-sorting method as "inspection of extracellular waveforms and subsequent offline spike sorting (Plexon Offline Sorter)" doesn't really allow the reader to form a picture of the likelihood of having 'single units' that are not really single. Particularly, nothing is mentioned about thresholding of spikes. I am not suggesting a thorough description of the spike sorting method but some information on the techniques used (within the Offline sorter and/or anything else) and a report on the authors' confidence of the single units produced (and how such confidence was evaluated). I understand that the literature utilising the types of probes used by the authors is also most of the times vague as to what spike-sorting really means, yet in most cases the assignment of spikes to single units does not bare the same significance to the subsequent results as in the case of this work, hence the above argument for some more clarity on the matter.

We thank the reviewer for this comment and agree that the definition of "spike-sorting" can be murky in many cases. We also recognize that we were perhaps too brief in our description of how the waveforms were selected. But for our experiments, it is straightforward. We have now expanded out the text in the methods to describe how exactly we identified our units. Please see the text below that has been included in the methods section of the manuscript. Our confidence in our waveforms comes from a

mixture of online vigilance and offline spike sorting.

In the methods section, we included the following subsection:

“Identification of single neurons during recordings Our procedure for identifying well-isolated single neurons was as follows: In the case of the single FHC tungsten electrode recordings, we moved the electrode and conservatively adjusted the threshold until we identified a well-demarcated set of waveforms. […] Finally, the same offline procedures used for FHC electrodes were repeated for the U-probe recordings.”

In the results, we included the following sentences:

“We restricted our analysis to well-isolated single neurons identified through a combination of careful online isolation combined with offline spike sorting (see Methods section: Identification of single neurons during recordings). Extracellular waveforms were isolated as single neurons by only accepting waveforms with minimal ISI violations (1.5% < 1.5 ms). This combination of online vigilance, combined with offline analysis, provides us the confidence to label these waveforms as single neurons.”

2. A purported advantage of UMAP is that, in principle, it can deal with a very large number of features (better than t-SNE, which struggles with > 50 features or so). However, the technique is still new and this belief is largely anecdotal. It would be beneficial, to both the neuroscientific community and in general to anyone considering using UMAP, if the authors could make a comparison between using the raw data set in UMAP and using a dimensionality reduced one through PCA giving only the top PCs to UMAP. Practically provide some evidence to support your choice of using the raw data set. Also providing an idea of the "complexity" of the data set would be useful (maybe by mentioning the variance explained by different number of principal components after a PCA, even if you do not end up using the primary components generated).

Thank you for this comment and helping us to better communicate the impact of our

method. We were not sure if the “advantages" referred to here had to do with either (or both) (1) UMAP being anecdotally more performant than t-SNE on high-dimensional data or (2) PCA being necessary pre-processing for UMAP to elicit interesting structure in the projected space. We address both potential questions below.

To the first point, PCA is often suggested for t-SNE as it reduces the dimensionality of the input data and is useful for speeding up the t-SNE algorithms. However, recent improvements in the algorithm, such as FIt-SNE (Linderman et al., 2019), put algorithm speed on par with UMAP even with high input dimensionalities. In our dataset, FIt-SNE computed faster than UMAP with the ambient 48-dimensions: 2.39 ± 0.03 s vs. 4.92 ± 0.12 s (mean ± S.D.) for FIt-SNE and UMAP respectively. Thus, if input dimensionality is specifically considered, both t-SNE and UMAP are equally performant. However, UMAP scales better with increasing output dimensionality (the n components parameter) whereas t-SNE, across its implementations, becomes exponentially slower (McInnes et al., 2018). Although not used in our manuscript, in other settings it might be interesting to explore UMAP dimensions beyond two (UMAP-3, -4, -5, etc). We have edited our manuscript to make clear that both algorithms are similarly fast with respect to input dimension but differ in how their performance scales with output dimension. The revised discussion now reads:

“At the core of WaveMAP is UMAP which has some advantages over other non-linear

dimensionality reduction methods that have been applied in this context. Although most

algorithms offer fast implementations that scale well to large input dimensionalities and

volumes of data (Linderman et al., 2019; Nolet et al., 2020), UMAP also projects efficiently into arbitrary output dimensionalities while also returning an invertible transform. That is, we can efficiently project new data into any arbitrary dimensional projected space without having to recompute the mapping.”

To the second point about UMAP's structure with and without pre-processing with PCA, we find that the structure is largely the same after being passed data reduced onto the first three principal components covering 94% of the variance. This is perhaps unsurprising given the low-dimensionality of our dataset which can be fully-captured in just a few components. We have added the following text and a supplementary figure (Figure 3-Supplement 3) to the manuscript to provide this information.

“In addition, common recommendations to apply PCA before non-linear dimensionality reduction were not as important for our waveform dataset, which was fairly low-dimensional (first three PC's explained 94% variance). Projecting waveforms into a 3-dimensional PC-space before WaveMAP produced a clustering very similar to data without this step (Figure 3-Supplement 3).”

3. The choice of the resolution parameter for the clustering algorithm (line 873 and Figure 2B) has rather important consequences in the subsequent analysis. The choice should be better justified.a. The authors say they chose the resolution parameter at the elbow of the number of clusters, but that is not the case. The elbow point is the immediate next one (resolution of 2 and number of clusters probably 7).b. The "number of clusters" value that was the most stable over the resolution is 5 (given resolutions between 3 and 5 if I read the graph properly), which might be another logical assumption for picking the used resolution.c. So although the argument that choosing a smaller resolution leads to too many clusters with too few samples in them is a valid one, nonetheless there seems to be no real argument against choosing a resolution that would lead to a smaller number of clusters. Since this clustering impacts the whole of the subsequent analysis, it would be important to understand the authors' reasoning.d. It would be interesting for example to see how a smaller number of WaveMAP generated clusters would fare in the learnability test shown by the authors (starting at line 188).

We thank the reviewer for this question. We realized that our justification for how we chose the number of clusters was very unclear.

1. Regarding the first point, unlike the Gaussian mixture model where the number of clusters is an actual parameter, number of clusters is not a parameter for WaveMAP. Instead, this is indirectly controlled by the resolution parameter. The number of clusters is an output after choosing a resolution parameter and is not the parameter optimized for.

2. The objective function for optimizing WaveMAP is the maximization of modularity score. Modularity (the “connectedness" of a community, see Methods) is a community-wise measure defined as the difference between the weights of the edges within a cluster and the edges incoming from any other node outside of the cluster. Maximizing this value over the whole graph finds communities with high amounts of intra-connectivity and low out-connectivity. We chose a resolution parameter that balanced our need for ensuring that we don't fractionate our dataset into clusters with very few samples while also maximizing the modularity score. A resolution parameter of 1.5 allowed us to balance these two goals. Choosing a resolution of 1 would have led to many clusters with small sample sizes and a resolution parameter of 2 would have meant that we were choosing a suboptimal solution in terms of modularity. This is our justification for choosing 1.5 in the manuscript. In the future, further knowledge of ground truth and a rough understanding of the number of candidate cell types identified by electrophysiology might also make it easier to use that as a prior for a resolution parameter.

3. We recognize that it might be difficult to choose a resolution parameter. For this particular case, we also offer the option of using Ensemble Clustering With Graphs (ECG). This clustering method obviates the need for choosing a resolution parameter. The results of ECG are nearly exact to what we found using a resolution parameter of 1.5. We have now updated the text to better justify our choice of 1.5 as the resolution parameter for this dataset.

4. Louvain clustering also acts hierarchically with respect to changing resolution parameter: clusters split or merge as the parameter decreases or increases respectively (Figure 3-Supplement 1A). The clusters and their constituent members do not shift dramatically as with other methods such as a GMM (Figure 4-Supplement 1C) when the number of clusters changes. Thus, even though the number of clusters change with resolution, the overall structure does not. We have now provided the following revised description in the text to better describe the justification for choosing 1.5 as our resolution parameter. We also added an explicit supplementary section which expands on the text in the results further.

“The number of clusters identified by WaveMAP is dependent on the resolution parameter for Louvain clustering. A principled way to choose this resolution parameter is to use the modularity score (a measure of how tightly interconnected the members of a cluster are) as the objective function to maximize. We chose a resolution parameter of 1.5 that maximized modularity score while ensuring that we did not overly fractionate the dataset (n < 20 within a cluster; Figure 3A, B, and columns of Figure 3-Supplement 1A). Additional details are available in the “Parameter Choice" section of the Supplementary Information.

Louvain clustering with this resolution parameter of 1.5 identified eight clusters in total

(Figure 3A). Note, using a slightly higher resolution parameter (2.0), a suboptimal solution in terms of modularity, led to seven clusters (Figure 3-Supplement 1A). The advantage of Louvain clustering is that it is hierarchical and choosing a slightly larger resolution parameter will only merge clusters rather than generating entirely new cluster solutions. Here, we found that the higher resolution parameter merged two of the broad-spiking clusters 6 and 7 while keeping the rest of the clusters largely intact and more importantly, did not lead to material changes in the conclusions of analyses of physiology, decision-related dynamics, or laminar distribution described below. Finally, an alternative ensembled version of the Louvain clustering algorithm (ensemble clustering for graphs [ECG]; Poulin and Theberge, 2018), which requires setting no resolution parameter, produced a clustering almost exactly the same as our results (Figure 3-Supplement 1C).

We also added additional details in the supplementary information to guide the reader further:

“Parameter Choice: Louvain clustering (Blondel et al., 2008) requires the specification

of a resolution parameter, t, which controls the “characteristic scale" by which network

communities are identified; the larger this parameter, the fewer the number of clusters

(communities) detected and vice versa (Lambiotte, 2007).

We selected a resolution parameter based on two factors. The most important factor was modularity score. Modularity (the “connectedness" of a community, see Methods) is a community-wise measure defined as the difference between the weights of the edges within a cluster and the edges incoming from any other node outside of the cluster. Maximizing this value over the whole graph finds communities with high amounts of intra-connectivity and low out-connectivity. The second factor we considered was the sizes of the resulting clusters after choosing a resolution parameter. We did not want clusters with too few members (n < 20) which would be statistically difficult to interpret. The regions with the highest modularity score were at a resolution parameter of 1 and decreased from there onwards. However, choosing a resolution parameter of 1 led to a large number of clusters which often had very few members (Figure 3-Supplement 1A, leftmost column) making downstream statistical comparisons underpowered. We therefore chose t to be 1.5 which resulted in the next best average modularity score of 0:761 ± 0:004 (mean ± S.D.) and an average of 8:29 ± 0:84 (mean ± S.D.) clusters across 25 random data permutations (Figure 3B). In this manner, we found a set of waveform clusters that balanced the diversity found by UMAP and statistical interpretability.”

We also agree it is important to see if our results only hold for a particular choice of resolution parameter. We find that across many values of resolution parameter UMAP is better than GMM. In the original manuscript, we performed this analysis for 4 and 8 clusters and found that WaveMAP was better than a GMM. Now, at your and R1's encouragement, we have performed it for all values from 2 to 16 and show that in every case our solution is as good if not better than GMMs (Figure 4-Supplement 2). We also changed both the n neighbors and resolution parameters from UMAP and Louvain respectively to yield a large range of cluster numbers; at the same time, we changed the number of components for a Gaussian mixture model (the n components parameter) across the same range. We find that, across every cluster number, a gradient boosted decision tree classifier trained on WaveMAP clusters had better performance than the clusters produced by a GMM applied to waveform features. We have now included a new supplementary figure (Figure 4-Supplement 2) and the text below to address your concern.

“In fact, across all cluster numbers (n components from 2 to 16), a classifier tuned for the GMM performed more poorly on the GMM labels than a WaveMAP projection with the same number of clusters (Figure 4-Supplement 2E, in red). Tuning WaveMAP parameters that induce different cluster numbers, whether n neighbors (in dark blue) or resolution (in light blue), had little effect on classifier performance (Figure 4-Supplement 2E, in blues). WaveMAP yielded mappings that were more generalizable than a GMM on features across every number of clusters and both parameters investigated.”

4. Comparison to t-SNE. This comment has been prompted by the phrase "While providing visualization, these methods are difficult to cluster upon because they return a different mapping on every initialization." (line 546). The authors provide reference 155 that actually directly contradicts this statement. Although the other comments about t-SNE and UMAP are valid (reverse mapping, and use of labels in training), I believe the first comment requires either removal or better further exploration. It would be very informative if the authors did a t-SNE embedding (on the data after PCA) and then used a DBSCAN to classify the cells. This would be rather straight forward to do at their data set size, even multiple times for different initialisations (but see Dimitriadis et al., 2018a for a GPU accelerated t-SNE algorithm and also for t-SNE and DBSCAN used in spike-sorting). That result can then also be contrasted to the WaveMAP and GMM methods using their random forest classifier. Even if the results from the t-SNE / DBSCAN clustering are comparative to WaveMAP the reverse mapping capability of UMAP is definitely a strength that is very nicely utilised in this work and which any other non-linear dimensionality reduction technique doesn't have. As a final point, in the comparison section the authors could also add that t-SNE is way slower than UMAP and this speed difference starts to be very obvious (minutes vs days) in sample sizes in the hundreds of thousands to millions (i.e. the regime of spike collection that high density probes collect Dimitriadis et al. 2018b and Steinmetz et al., 2018).

We thank the reviewer for pointing out this inaccuracy and we have removed this statement. We have now revised the Discussion to instead focus more on three advantages that UMAP has over other non-linear dimensionality reduction methods namely (1) it is highly-performant and approximately invertible with respect to output dimensionality, (2) it returns an invertible transform for cross-validation, and (3) it supports supervised and semi-supervised learning.

“At the core of WaveMAP is UMAP which has some advantages over other non-linear dimensionality reduction methods that have been applied in this context. Although most

algorithms offer fast implementations that scale well to large input dimensionalities and

volumes of data (Linderman et al., 2019; Nolet et al., 2020), UMAP also projects efficiently into arbitrary output dimensionalities while also returning an invertible transform. That is, we can efficiently project new data into any arbitrary dimensional projected space without having to recompute the mapping.

These properties provide three advantages over other non-linear dimensionality reduction approaches: First, our method is stable in the sense that it produces a consistent number of clusters and each cluster has the same members across random subsamples (Figure 3-Supplement 1B). Clustering in the high-dimensional space rather than in the projected space lends stability to our approach. Second, it allows exploration of any region of the projectedspace no matter the intuited latent dimensionality this yields an intuitive understanding of how UMAP non-linearly transforms the data, which might be related to underlying biological phenomena. Thus, UMAP allows WaveMAP to go beyond a “discriminative model" typical of other clustering techniques and function as a “generative model" with which to make predictions. Third, it enables cross-validation of a classifier trained on cluster labels, impossible with methods that don't return an invertible transform. To cross-validate unsupervised methods, unprocessed test data must be passed into a transform computed only on training data and evaluated with some loss function (Moscovich and Rosset, 2019). This is only possible if an invertible transform is admitted by the method of dimensionality reduction as in UMAP.”

We also agree that it would be useful to examine how our method compares against DBSCAN clustering applied to a t-SNE projection which has been useful for clustering neural activity (Dimitriadis et al., 2018; Mahallati et al., 2019). However, we would first like to draw a distinction between the clustering of single spikes for purposes of spike sorting and the clustering of averaged spikes from single units for putative type classification. In the former, spikes seem to clearly separate into clusters in the projected space of t-SNE or UMAP (see Dimitriadis et al., 2018 and Sedaghat-Nejad et al., 2021); in the latter (our work), averaged single unit spikes seem to form continuums that cannot be easily delineated as clusters in the projected space. This seems to be a key difference explaining how clustering in the projected space succeeds for spike sorting but fails (as we will show next) for clustering in the projected space for cell type classification.

Here, we compare WaveMAP against DBSCAN on a t-SNE projection. To make this comparison, we applied each method to 100 random subsamples, of various proportions of the full dataset, and calculated the adjusted mutual information score (AMI) against a “reference" clustering obtained from applying the respective method to the full dataset. For a detailed explanation of AMI, see note at the end of the document. We find that while DBSCAN on t-SNE produces parcellations of waveform structure very similar to WaveMAP when examining a single clustering (Figure 4—figure supplement 1A) of the full dataset, this method produces very variable clusterings on random subsets of the full dataset (Figure 4—figure supplement 1B).

WaveMAP begins with moderate AMI scores at low data fractions and increases steadily. DBSCAN on t-SNE begins at very low AMI scores, increases quickly with data fraction, then matches WaveMAP at high percentages of the full dataset. However, DBSCAN on t-SNE exhibits highly variable AMI scores as individual analyses are examined especially at lower data fractions (Figure 4—figure supplement 1C, left). So although DBSCAN on t-SNE's variability disappears at 90% of the dataset (Figure 4—figure supplement 1C, right), this suggests that this technique might generalize less well out-of-dataset or else requires more data to converge than is provided in our data fractions.

Based on these results, we have now added the following text to the manuscript to assuage readers who might have the same question:

“We also found that WaveMAP was robust to data subsetting (randomly sampled subsets of the full dataset, see Supplementary Information Tibshirani and Walther, 2012), unlike other clustering approaches (Figure 3-Supplement 1B, green, Figure 4-Supplement 1). We applied WaveMAP to 100 random subsets each from 10% to 90% of the full dataset and compared this to a “reference" clustering produced by the procedure on the full dataset. WaveMAP was consistent in both cluster number (Figure 3-Supplement 1B, red) and cluster membership (which waveforms were frequently “co-members" of the same cluster; Figure 3-Supplement 1B, green).”

Thus in conclusion, WaveMAP seems to perform better in our particular use case in parcellating somewhat continuous structure while methods like DBSCAN on t-SNE are successful for other use cases such as spike sorting where clearer clusterings exist after non-linear dimensionality reduction. We reiterate that we think this has more to do with clustering on the network graph vs. in the projected space and this doesn't have to do with the choice of particular dimensionality reduction whether that be UMAP or t-SNE.

To address the final point on comparing the algorithmic performance of t-SNE vs. UMAP, we've revised the manuscript to avoid comparisons. We actually found that, given the low-dimensionality and low number of data points in our dataset, that t-SNE (with FIt-SNE) was faster than UMAP. However, there also exists an incredibly fast GPU implementation of UMAP through RAPIDS's cuML library (Nolet et al., 2020) but benchmarks were generated with an enterprise GPU cluster (NVIDIA DGX-1) that most researchers would not have access to. Thus we eschew this point as the advantages of one method or the other depend intrinsically upon the properties of the dataset and the means of the investigators.

5. Number of waveform features used to classify the cells. The authors compare their WaveMAP to a seemingly standard technique in the macaque literature (or maybe in the literature that uses sparse electrode probes), that uses 3 features of the waveform to classify the cells. This is not the way things are done in most of the rodent electrophysiology (or works that use denser probes or tetrodes). The standard technique is to PCA each spike form and use the first 3 components per electrode. In the case of this work that would lead to the use of the same number of features, but (probably) much larger retention of information. See Harris et al., 2000 (and references therein) for how the PCA of the waveforms is used in tetrode recordings to do spike-sorting. More recent spike-sorting algorithms use template matching of the waveform. For instance, Kilosort (Pachitariu et al., 2016) omits spike detection and PCA, which can miss useful information. Instead, it relies on identifying template waveforms and their timing properties, in order to assign spikes (in the entirety of their waveform) to different cells. Also comparing the WaveMAP process to the clustering results (following the used Generalised Gaussian Models method, or other peak-density detection methods) of this feature set would be both more informative and more valid for the larger electrophysiology community.

We apologize for not making a clearer distinction but we only document how our method performs in the context of classifying waveforms into candidate cell types rather than the sorting of individual spikes to distinguish single units (spike sorting). For spike sorting, researchers definitely use the first 3 principal components. Indeed, we did the same when drawing our cluster boundaries to identify isolated single neurons. However, what WaveMAP is focused on is separating cell classes which is often done after spike sorting.

We know of many rodent papers that use the feature based method on average waveforms to drive inference about the role of putative excitatory and inhibitory neurons in neural circuits (see examples from Yu et al., (2019), Robbins et al., (2013), Bartho et al., (2004), Niell and Stryker (2008), Bruno and Simons (2002), Stringer et al., (2016), and review by Peyrache and Destexhe (2019)). We don't think using waveform features is something unique to the macaque literature and many of these papers often deploy tetrode based electrophysiological methods to record neural activity. Having said that, perhaps there is opportunity for future work to eschew separate methods for sorting and classification and combine them in a single step.

Nevertheless, we hear your comment about the usefulness of comparing WaveMAP to a GMM based on the first three PCs (instead of features). We compared WaveMAP directly against a GMM on PCA in terms of AMI (Figure S3). We set a “ground truth" by applying the method to the full dataset and generated an AMI for each data fraction by averaging the individual AMI scores for 100 random subsets at various data fractions. While the GMM on PCA formed what seems to be sensible clusters, these were fairly variable and this did not improve even with increasing data fraction. Examining each run individually, we can see that at 40% of the full dataset, the AMI has a large variance but at 90% of the full dataset, this variance is qualitatively different instead of forming a continuum of AMI values over a range, the AMI contracts towards different locations. This seems to suggest that even at high data fractions, GMM on PCA arrives at several local minima in the solution space rather than WaveMAP which aggregates at one specific AMI score.

We have included Figure S3 in the manuscript and included the following text:

We also found that WaveMAP was robust to data subsetting (randomly sampled subsets of the full dataset, see Supplementary Information (Tibshirani and Walther, 2012)), unlike other clustering approaches (Figure 3-Supplement 1B, green, Figure 4-Supplement 1). We applied WaveMAP to 100 random subsets each from 10% to 90% of the full dataset and compared this to a “reference" clustering produced by the procedure on the full dataset. WaveMAP was consistent in both cluster number (Figure 3-Supplement 1B, red) and cluster membership (which waveforms were frequently “co-members" of the same cluster; Figure 3-Supplement 1B, green).”

To answer your second comment, as to whether WaveMAP would do better than template based methods, we again have to emphasize that this paper is more focused on the identification of cell classes after spike sorting. While we have not explicitly addressed the use of WaveMAP for spike-sorting, evidence suggests the method of UMAP in conjunction with graph clustering would also be effective in separating out units where the PCA approach is insufficient. We would like to point the reviewer to Sedaghat-Nejad et al., (2021) in which they use UMAP in their P-sort spike sorter and find that it outperforms template sorters (Kilosort2) or peak-density centroid methods (SpyKING CIRCUS) especially when more complicated waveforms are present as in cerebellar recordings. We should also mention that this group is interested in incorporating the innovations from our WaveMAP paper into their workow (personal communication).

ReferencesAli M, Jones MW, Xie X, Williams M (2019) TimeCluster: dimension reduction applied to

temporal data for visual analytics. The Visual Computer 35:1013{1026.

Bartho P, Hirase H, Monconduit L, Zugaro M, Harris KD, Buzsaki G (2004) Characterization of Neocortical Principal Cells and Interneurons by Network Interactions and Extracellular Features. Journal of Neurophysiology 92:600{608.

Becht E, McInnes L, Healy J, Dutertre CA, Kwok IWH, Ng LG, Ginhoux F, Newell EW (2018)

Dimensionality reduction for visualizing single-cell data using UMAP. Nature

Biotechnology 37:38{44.

Blondel VD, Guillaume JL, Lambiotte R, Lefebvre E (2008) Fast unfolding of communities in large networks. Journal of Statistical Mechanics: Theory and Experiment 2008:P10008.

Bruno RM, Simons DJ (2002) Feedforward Mechanisms of Excitatory and Inhibitory Cortical Receptive Fields. Journal of Neuroscience 22:10966{10975.

Deligkaris K, Bullmann T, Frey U (2016) Extracellularly Recorded Somatic and Neuritic Signal Shapes and Classification Algorithms for High-Density Microelectrode Array Electrophysiology. Frontiers in Neuroscience 10:421.

Dimitriadis G, Neto JP, Aarts A, Alexandru A, Ballini M, Battaglia F, Calcaterra L, Chen S,

David F, Fiath R, Fraz~ao J, Geerts JP, Gentet LJ, Helleputte NV, Holzhammer T, Hoof Cv,

Horvath D, Lopes G, Lopez CM, Maris E, Marques-Smith A, Marton G, McNaughton BL,

Meszena D, Mitra S, Musa S, Neves H, Nogueira J, Orban GA, Pothof F, Putzeys J, Raducanu BC, Ruther P, Schroeder T, Singer W, Steinmetz NA, Tiesinga P, Ulbert I, Wang S, Welkenhuysen M, Kampff AR (2020) Why not record from every electrode with a CMOS scanning probe? bioRxiv p. 275818.

Dimitriadis G, Neto JP, Kampff AR (2018) t-SNE Visualization of Large-Scale Neural

Recordings. Neural Computation 30:1750{1774.

Gold C, Henze DA, Koch C, Buzsaki G (2006) On the Origin of the Extracellular Action

Potential Waveform: A Modeling Study. Journal of Neurophysiology 95:3113{3128.

Gouwens NW, Sorensen SA, Baftizadeh F, Budzillo A, Lee BR, Jarsky T, Alfiler L, Baker K,

Barkan E, Berry K, Bertagnolli D, Bickley K, Bomben J, Braun T, Brouner K, Casper T,

Crichton K, Daigle TL, Dalley R, de Frates RA, Dee N, Desta T, Lee SD, Dotson N, Egdorf T,

Ellingwood L, Enstrom R, Esposito L, Farrell C, Feng D, Fong O, Gala R, Gamlin C, Gary A,

Glandon A, Goldy J, Gorham M, Graybuck L, Gu H, Hadley K, Hawrylycz MJ, Henry AM,

Hill D, Hupp M, Kebede S, Kim TK, Kim L, Kroll M, Lee C, Link KE, Mallory M, Mann R,

Maxwell M, McGraw M, McMillen D, Mukora A, Ng L, Ng L, Ngo K, Nicovich PR, Oldre A,

Park D, Peng H, Penn O, Pham T, Pom A, Popovic Z, Potekhina L, Rajanbabu R, Ransford S, Reid D, Rimorin C, Robertson M, Ronellenfitch K, Ruiz A, Sandman D, Smith K, Sulc J,

Sunkin SM, Szafer A, Tieu M, Torkelson A, Trinh J, Tung H, Wakeman W, Ward K, Williams G, Zhou Z, Ting JT, Arkhipov A, Sumbul U, Lein ES, Koch C, Yao Z, Tasic B, Berg J, Murphy GJ, Zeng H (2020) Integrated Morphoelectric and Transcriptomic Classification of Cortical GABAergic Cells. Cell 183:935{953.e19.

Harris KD, Henze DA, Csicsvari J, Hirase H, Buzsaki G (2000) Accuracy of Tetrode Spike

Separation as Determined by Simultaneous Intracellular and Extracellular Measurements. Journal of Neurophysiology 84:401{414.

Jia X, Siegle JH, Bennett C, Gale SD, Denman DJ, Koch C, Olsen SR (2019) High-density

extracellular probes reveal dendritic backpropagation and facilitate neuron classification. Journal of Neurophysiology 121:1831{1847.

Lambiotte R (2007) Finding communities at different resolutions in large networks.

Linderman GC, Rachh M, Hoskins JG, Steinerberger S, Kluger Y (2019) Fast interpolation-based t-SNE for improved visualization of single-cell RNA-seq data. Nature Methods 16:243{245.

Lundberg S, Lee SI (2017) A Unified Approach to Interpreting Model Predictions. arXiv.

Mahallati S, Bezdek JC, Popovic MR, Valiante TA (2019) Cluster tendency assessment in

neuronal spike data. PLOS ONE 14:e0224547.

McInnes L, Healy J, Melville J (2018) Umap: Uniform manifold approximation and projection for dimension reduction. arXiv.

Moscovich A, Rosset S (2019) On the cross-validation bias due to unsupervised pre-processing. arXiv.

Niell CM, Stryker MP (2008) Highly Selective Receptive Fields in Mouse Visual Cortex. The Journal of Neuroscience 28:7520{7536.

Nolet CJ, Lafargue V, Raff E, Nanditale T, Oates T, Zedlewski J, Patterson J (2020) Bringing UMAP Closer to the Speed of Light with GPU Acceleration. arXiv.

Pachitariu M, Steinmetz N, Kadir S, Carandini M, D. HK (2016) Kilosort: realtime spike-sorting for extracellular electrophysiology with hundreds of channels. bioRxiv p. 061481.

Paulk AC, Kffir Y, Khanna A, Mustroph M, Trautmann EM, Soper DJ, Stavisky SD,

Welkenhuysen M, Dutta B, Shenoy KV, Hochberg LR, Richardson M, Williams ZM, Cash SS (2021) Large-scale neural recordings with single-cell resolution in human cortex using high-density neuropixels probes. bioRxiv.

Peyrache A, Destexhe A (2019) Electrophysiological monitoring of inhibition in mammalian species, from rodents to humans. Neurobiology of Disease 130:104500.

Poulin V, Theberge F (2018) Ensemble Clustering for Graphs. arXiv.

Quirk MC, Wilson MA (1999) Interaction between spike waveform classification and temporal sequence detection. Journal of Neuroscience Methods 94:41{52.

Robbins AA, Fox SE, Holmes GL, Scott RC, Barry JM (2013) Short duration waveforms

recorded extracellularly from freely moving rats are representative of axonal activity. Frontiers in Neural Circuits 7:181.

Romano S, Vinh NX, Bailey J, Verspoor K (2016) Adjusting for chance clustering comparison measures. The Journal of Machine Learning Research 17:4635{4666.

12-02-2021-RA-*eLife*-67490 27 July 23, 2021

Rosenberg A, Hirschberg J (2007) V-measure: A conditional entropy-based external cluster evaluation measure In Proceedings of the 2007 Joint Conference on Empirical Methods in Natural Language Processing and Computational Natural Language Learning (EMNLP-CoNLL), pp. 410{420, Prague, Czech Republic. Association for Computational Linguistics.

Sedaghat-Nejad E, Fakharian MA, π J, Hage P, Kojima Y, Soetedjo R, Ohmae S, Medina JF,

Shadmehr R (2021) P-sort: an open-source software for cerebellar neurophysiology. bioRxiv.

Steinmetz N, Zatka-Haas P, Carandini M, Harris K (2018) Distributed correlates of

visually-guided behavior across the mouse brain. bioRxiv p. 474437.

Stringer C, Pachitariu M, Steinmetz NA, Okun M, Bartho P, Harris KD, Sahani M, Lesica NA (2016) Inhibitory control of correlated intrinsic variability in cortical networks. *eLife* 5:e19695.

Tibshirani R, Walther G (2012) Cluster Validation by Prediction Strength. Journal of

Computational and Graphical Statistics 14:511{528.

Timme NM, Lapish C (2018) A Tutorial for Information Theory in Neuroscience.

eNeuro 5:ENEURO.0052{18.2018.

Yu J, Hu H, Agmon A, Svoboda K (2019) Recruitment of GABAergic Interneurons in the Barrel Cortex during Active Tactile Behavior. Neuron 104:412{427.e4.